# Distinct neural mechanisms for heading retrieval and context recognition in the hippocampus during spatial reorientation

Celia M. Gagliardi [1], Marc E. Normandin [1], Alexandra T. Keinath [2], Joshua B. Julian[3], Matthew R. Lopez[1], Manuel-Miguel Ramos-Alvarez[4], Russell A. Epstein[5] & Isabel A. Muzzio [1] ✉

Reorientation, the process of regaining one's bearings after becoming lost, requires identification of a spatial context (context recognition) and recovery of facing direction within that context (heading retrieval). We previously showed that these processes rely on the use of features and geometry, respectively. Here, we examine reorientation behavior in a task that creates contextual ambiguity over a long timescale to demonstrate that male mice learn to combine both featural and geometric cues to recover heading. At the neural level, most CA1 neurons persistently align to geometry, and this alignment predicts heading behavior. However, a small subset of cells remaps coherently in a context-sensitive manner, which serves to predict context. Efficient heading retrieval and context recognition correlate with rate changes reflecting integration of featural and geometric information in the active ensemble. These data illustrate how context recognition and heading retrieval are coded in CA1 and how these processes change with experience.

During disorientation, when navigators lose their bearings, the internal sense of direction becomes unreliable. Lost navigators must then rely on external cues to identify their surroundings and re-establish their sense of direction. This process, known as spatial reorientation, has been extensively studied in the fields of animal behavior and developmental psychology (for review, see ref. 1). However, despite some exceptions[2,3], the neural mechanisms underlying reorientation are largely unknown. The primary objective of the present study is to address this gap in knowledge, with a specific focus on elucidating how various types of external cues are utilized in spatial reorientation and how they may contribute to different stages of this process.

Previous behavioral research on reorientation across various species has consistently revealed a critical role of environmental geometry, the shape of a layout defined by the relative positioning of walls and boundaries. The significance of geometry in reorientation becomes evident through the challenge faced by disoriented

navigators in distinguishing between geometrically identical locations within symmetrical environments, such as the opposite corners of a rectangular chamber. This difficulty persists even when non-geometric features are present to disambiguate them[4–7]. Non-geometric features include cues such as odors, sounds, landmarks, or other sensory information that navigators use for orientation that do not rely on geometric relationships or spatial configurations of objects.

Several theories of reorientation have attempted to explain the cognitive mechanisms underlying reorientation behavior. The view-matching theory suggests that disoriented navigators reach their targets by matching the current panoramic view with a stored view of the goal location[8–10], a strategy that relies on visual recognition rather than retrieval of a cognitive map. This theory has gained support from research with ants and chicks[11–13], but has been challenged by findings showing that children can reorient using geometry regardless of viewpoint[14] and that visually impaired subjects can reorient using

[1]Department of Psychological & Brain Sciences, University of Iowa, Iowa City, IA 52245, USA. [2]Department of Psychology, University of Illinois Chicago, Chicago, IL 60607, USA. [3]Princeton Neuroscience Institute, Princeton University, Princeton, NJ 08544, USA. [4]Psychology Department, University of Jaen, Campus Las Lagunillas, Jaen 23071, Spain. [5]Department of Psychology, University of Pennsylvania, Philadelphia, PA 19104, USA. ✉e-mail: isabel-muzzio@uiowa.edu

geometry[15–17]. The geometric modular theory, on the other hand, proposes that navigators reorient based solely on a cognitive representation of environmental geometry that is impermeable to non-geometric information[18–21]. Supporters of this view propose that the brain possesses self-contained (modular) representations of spatial layouts, which hold significance for survival as they reflect stable aspects of the environment[18]. This theory gained substantial support from behavioral studies in various animal models[20,22], as well as neurophysiological recordings in rodents[2,3]. However, it has been challenged by studies showing that navigators sometimes use non-geometric featural information when reorienting[23–25], which indicates that non-geometric information is not completely ignored, though its use may depend on situational factors[5,21,26,27].

To account for the use of non-geometric information during reorientation, supporters of the geometric module view have suggested that reorientation relies on two independent processes, one relying on a modular representation of geometry and a second one involving associations of features and goal locations through reinforcement learning[21,28]. This two-process theory has, in turn, been questioned by critics who argued that the use of featural information can occur without a second process of associative learning when features are salient or have strong predictive directional value[29]. These ideas were developed in two more recent views: (1) The adaptive cue combination theory, which proposes that navigators use probabilistic inference to evaluate all available cues (geometric and non-geometric), based on their salience and reliability[26,30] and (2) the associative learning model, which suggests that available cues compete with each other by gaining or losing associative strength through learning[31,32]. According to these theories, reorientation is mediated by a single process that uses geometry, features, or a combination of both depending on which cue or combination is the most informative.

One approach to resolve these theoretical disputes is to use paradigms that are more similar to natural reorientation situations. Previous studies have mostly tested reorientation in single experimental chambers, whose identity never varies. In these situations, reorientation only requires recovering facing direction within a single context (heading retrieval). without the need to recognize the context itself (context recognition). In a previous behavioral study, we employed a two-context reorientation paradigm, revealing that disoriented mice rely on geometry for determining their heading, while simultaneously using non-geometric features to recognize the context[33]. These results suggest that heading retrieval and context recogntion are cognitively dissociable; moreover, they raise the intriguing possibility that geometry and non-geometric features may differentially affect the neural representations underlying these processes. In this study, we aim to reconcile theoretical disagreements regarding the cognitive mechanisms involved in reorientation by investigating the neural representations underlying heading retrieval and context recognition.

Previously, we began to uncover the neural foundation underlying heading retrieval by analyzing the firing patterns of hippocampal place cells as disoriented mice reoriented within a traditional single-context paradigm[2]. Hippocampal place cells fire in specific locations as animals navigate[34]. In oriented animals, place cell firing is influenced by both geometry[35] and features[36]. When animals explore environments with distinct geometric and featural cues, changes in firing rate (rate remapping[37]) and/or firing location (location remapping[36,38]) allow place cells to represent both the navigator's location and identity of the environment. We found that place fields of disoriented animals exhibited two possible orientations, consistent with the two orders of rotational symmetry of a rectangular chamber, indicating that the hippocampal map aligned to geometry. These place field rotations occurred despite the presence of a directional, disambiguating featural cue on one of the short walls of the context. Moreover, geometry-based alignment predicted reorientation behavior, indicating that

hippocampal alignment supported heading retrieval. Similar results have been observed in entorhinal head direction and grid cells[3]. However, since these studies used a single experimental chamber, they failed to tackle the crucial question of how neural systems concurrently encode both context recognition and heading retrieval in situations of contextual ambiguity. Furthermore, they did not explore the possibility that neural representations could evolve as animals learn the directional significance of non-geometric cues.

In the current study, we attempted to close this knowledge gap by conducting electrophysiological and calcium imaging recordings from hippocampal area CA1 while mice performed an extended version of the two-context reorientation task. We focused on CA1 because distinct place cell subpopulations within this region have been shown to detect distinct aspects of the same environment during navigation[39], a characteristic that would be essential to differentiate contribution of geometry and non-geometric features. First, we examined whether geometry-based heading retrieval persisted over time when processing of features was required for context recognition. Second, we explored the hippocampal mechanisms behind context recognition and heading retrieval, considering whether they involved separate neural processes, as suggested by the behavior, and how these processes evolved with task familiarity. Our results provide strong support for geometry-based reorientation and the existence of a separate cognitive process underlying feature/reward associations; however, they also demonstrate that the cognitive map guiding reorientation is not modular. Instead, it integrates experience-dependent firing rate changes as navigators adapt their strategies in response to the learned directional value of available cues.

## Results
### Reorienting behavior in a two-context paradigm
To investigate heading retrieval and context recognition, we examined the behavior of 14 male mice in a two-context reorientation paradigm. Animals were trained to find a reward hidden in one of four cups embedded in each corner of two chambers of identical rectangular shape. These contexts were distinguished by distinct polarizing features and reward locations (Fig. 1a). Throughout this paper, we will use the term feature exclusively to refer to the visual cues placed on the walls of each chamber. Before each trial, mice were disoriented to disrupt their internal sense of direction. Mice were placed in a plastic tube with a removable base, which was then placed on a rotating platform that alternated between clockwise and counterclockwise rotations (Fig. 1b). After being disoriented, the tube and its base were moved to one of the two rectangular chambers. Then, the mice were released by removing the base, allowing them to enter the chamber from random facing directions. The trials concluded either when the mice found the reward or after they had spent 3 min in the chamber. Detailed description of control measures implemented to prevent the use of external cues are described in the Methods.

Figure 1c represents the average percentage of first digs in each of the four cup locations in each context (C: correct rewarded location, G: geometric equivalent location sharing the same geometric properties as C, N: near location adjacent to the same short wall as C, F: far location adjacent to the same short wall as G). On day 1, animals searched more often in the rewarded geometric axis (C and G) than unrewarded geometric one (N and F) in each chamber (Fig. 1c, left panel), displaying a dissociable pattern of digging behavior between the two contexts (Supplementary Table 1). Consistent with our previous behavioral findings[33], these results indicate that mice use geometry for heading retrieval, as shown by the similar number of digs in geometrically equivalent cup locations. At the same time, they use features to disambiguate the contexts, evidenced by the greater number of digs in locations along opposite diagonals in each chamber. However, with additional training mice display more digs in the rewarded location than the geometric equivalent one (Fig. 1c, central

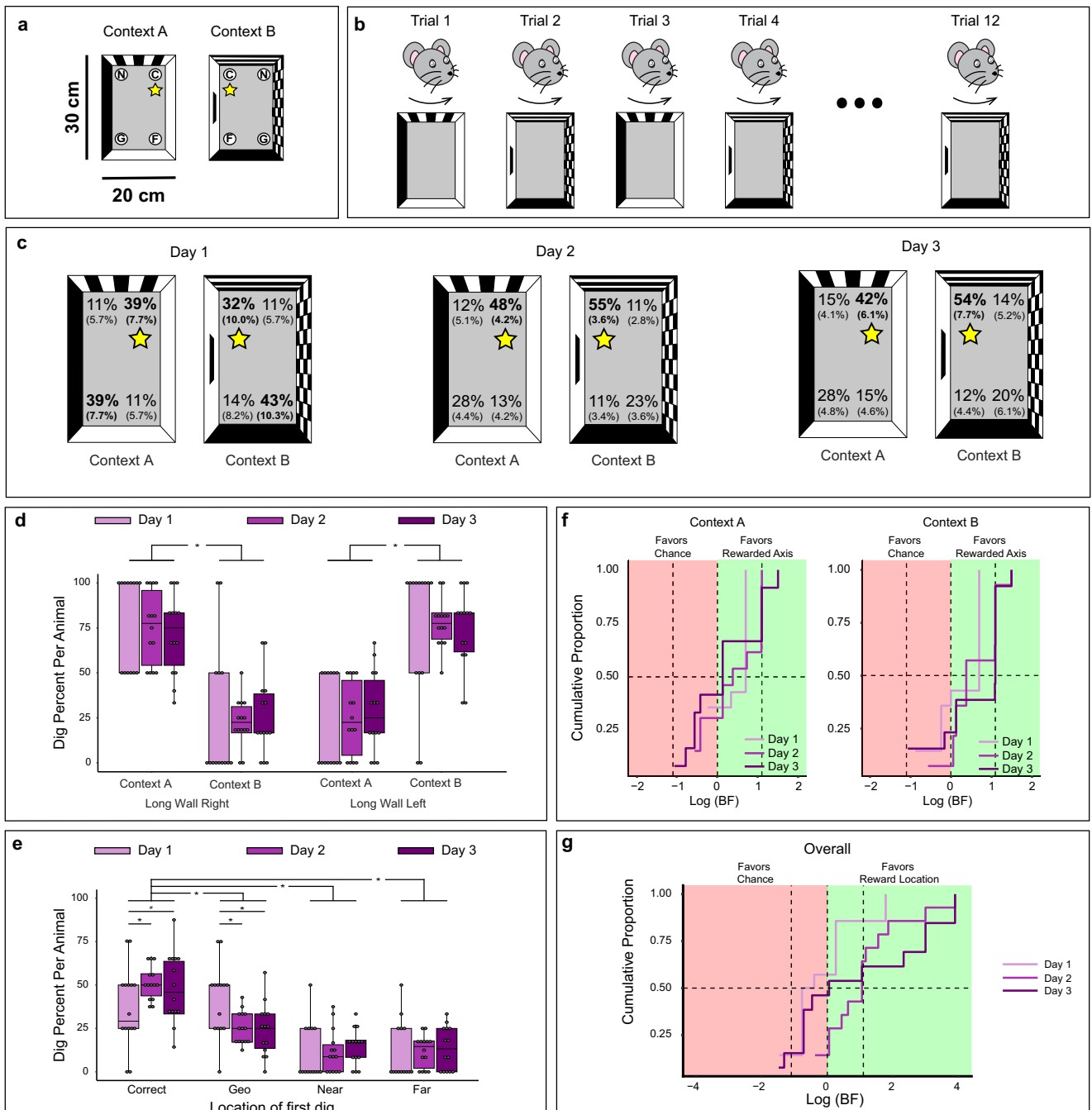

**Fig. 1 | Reorientation behavior in a two-context paradigm. a** Experimental chambers showing reward location (yellow star) in Context A (left) and Context B (right). **b** Session structure of two-context paradigm. **c** Percentage of digs in each cup location on test trials on day 1 (left), day 2 (center), and day 3 (right). Standard error of the mean is shown in parentheses below the mean percentage values. **d** Boxplots showing distribution of digs in geometrically correct vs. incorrect axes in each context. **e** Boxplots showing distribution of digs in each cup location combining both contexts. In all boxplot graphs, the boxes indicate the upper and lower quartiles of the data and the whiskers (extending lines) the minimum and maximum outside the quartiles. The horizontal line indicates the median. Dots represent individual data points and asterisks (*) indicate $p \leq 0.05$, the value at which the significance level was set. **f** Cumulative proportion of Bayes Factor (BF) on days 1 to 3 per context evaluating the alternative model ($M_{Alt}$) that animals

preferentially dug on a distinct rewarded axis in each context vs. the null model ($M_{null}$) that animals dug by chance. The point at which the cumulative proportion intercepts the dotted line at 0 shows that even though there are some subtle differences in the rate of learning on day 2, these differences decrease on day 3 (Percentages on negative values: Day 2: Context A: 28.57%, Context B: 7.14%; Day 3: Context A: 35.71%, Context B: 21.43%). **g** Cumulative proportion of individual Bayes Factors (BF) across days 1 to 3 combining contexts evaluating the alternative model ($M_{Alt}$) that animals increased digging in C locations with experience vs. the null model ($M_{null}$) that animals dug by chance. Conventional values showing the border marking credibility for $M_{alt}$ ($\log(BF) > \log(1/3) = 1.1$) and $M_{null}$ ($\log(BF) < \log(1/3) = -1.1$) are indicated by vertical dashed lines. The value of half (0.5) of the sample is marked by a horizontal dashed line. $N = 14$ animals. Source data are provided as a Source Data file.

and right panels). This suggests that with experience, the animals incorporate features to reorient.

To confirm that mice differentiated between the contexts by showing distinct digging behavior in each chamber, we computed the proportion of digs made along the geometrically rewarded (C and G) and unrewarded (N and F) axes in each context. Subsequently, we performed a 3-way repeated measures ANOVA, utilizing axis, context (A or B), and day of testing (day 1, 2, or 3) as within-subject factors (Fig. 1d). Each axis was defined by the diagonal that connects two corners with geometrically equivalent properties and are described as absolute corner location. In context A, cups C and G were positioned where the long wall was on the right, while cups N and F were placed where the long wall was on the left, relative to the animal's heading toward the cups on each trial. Thus, the rewarded geometric axis in Context A was called "long wall right" and in Context B "long wall left". The results showed a significant interaction between axis and context [$F(1.00, 13.00) = 96.55$, $p < 0.0001$], while the rest of the effects were not significant ($p > 0.05$). Post hoc analyses using Rom's multiple comparisons revealed that percentage of digs on the long wall right axis were higher than digs along the long wall left axis in Context A (long wall right: 75.3% vs. long wall left: 24.7%, $p < 0.05$), but the opposite occurred in Context B (long wall right: 24.4% vs. long wall left: 75.6%, $p < 0.05$). These findings demonstrate that animals discriminate the contexts using their unique featural cues and reward locations, which is evident in distinct digging patterns in each chamber.

The digging data shown in Fig. 1c also suggested that prolonged training in the task led to the use of features for heading retrieval, which was evident in more digs in the rewarded corner (C) and fewer digs in the geometric corner (G) on days 2 and 3 (Fig. 1e). To confirm the shift in reorientation strategies (i.e., the incorporation of features to recover heading), we conducted a repeated measures ANOVA on the combined dig locations in both contexts, with day of testing (day 1 to 3) and digging locations (C, G, N, and F) as within subjects' factors (Fig. 1e, note that digging locations are different in the two contexts but are combined in this analysis). Results revealed a main effect of dig location [$F(2.18, 28.36) = 29.30$, $p < 0.0001$], as well as an interaction between day and dig location [$F(4.35, 56.53) = 2.60$, $p = 0.042$], but no effect of day ($p > 0.05$). Post hoc analyses using Rom's multiple comparisons revealed a difference in first dig location across days. On day 1, digs in the correct and geometrically equivalent corners (C and G) were higher than in the incorrect ones (N and F, $p < 0.05$), with no difference between C and G ($p > 0.05$), or N and F ($p > 0.05$). However, digs in C were significantly higher than digs in other locations on days 2 and 3 (G, N, F, $p < 0.05$). Moreover, digs in the G location decreased on days 2 and 3 in comparison to day 1 ($p < 0.05$). These behavioral results confirmed that animals learned to use features to distinguish geometrically equivalent corners with experience, in addition to using these same cues to disambiguate the contexts.

Notably, despite extensive training, animals occasionally made errors. Therefore, we analyzed the second digging choice to assess if animals could correct their behavior after an error. Our results showed that mice frequently corrected errors on second digs, demonstrating their ability to learn task contingencies even after an initial mistake (Supplementary Fig. 1). Together, our data indicate that animals incorporate features to recover heading direction with extensive training.

We validated these results using hierarchical Bayesian analysis, which assesses individual animal performance. First, we tested the alternative model ($M_{Alt}$) that animals preferentially dug on the rewarded axis in each context (long wall right in Context A and long wall left in Context B) vs. the null model ($M_{null}$) that animals dug by chance. The global Bayes Factor (BF) in each context on days 1, 2, and 3 exceeded the critical value commonly used to evaluate the credibility of the alternative model in each context [$\log(BF) > 1.1$], indicating credibility for learning [$\log(BF) > 1.1$ provides credibility for $M_{Alt}$, $\log(BF) < -1.1$

provides credibility for $M_{null}$][40]. This supports the notion that animals distinguish between contexts by digging along unique axes within each chamber [Context A: group BFs (days 1, 2, and 3): 4.16, 7.23 and 6.94; Context B: group BFs (days 1, 2, and 3): 4.18, 8.60, and 8.56; Fig. 1f, Supplementary Table 2].

We next tested if BF corroborated that featural cues were incorporated to recover heading with extensive training. We evaluated the alternative model ($M_{Alt}$) that animals dug in the rewarded cup location vs. the null model ($M_{null}$) that animals dug by chance. For this analysis we combined data from both contexts. The cumulative distributions of individual BF and global BF analysis provided credibility for $M_{Alt}$ only on days 2 and 3 ($\log(BF) > 1.1$), indicating that animals dug in C more than the other locations (N, G, or F) [group BFs: Day 1: $\log(BF) = -2.61$; Day 2: $\log(BF) = 15.06$; Day 3: $\log(BF) = 17.67$, Fig. 1g, for BF individual data see Supplementary Table 3]. In summary, animals initially rely on geometry for heading retrieval and features for context recognition, but also incorporate features to recover heading with experience and disambiguate geometrically equivalent locations.

## Place field alignment to spatial geometry persists over days and predicts digging behavior

We next sought to investigate the neural correlates during reorientation. Given the potential impact of uneven exploration on place cell firing properties, we first assessed the extent of chamber and cup sampling in mice generating neural data. Our findings revealed that mice displayed proper exploration of both the environment and the cups in most trials (Supplementary Fig. 2), indicating that the tethers connecting the mice or short trial length did not disrupt proper exploration. This assurance provides confidence that any observed differences in neurological data are not attributable to uneven sampling.

Our previous work in single chambers showed that the hippocampal map aligns to geometry in disoriented mice[2]. Here, we tested if this alignment persisted over time when animals performed a two-context reorientation task, which required attention to non-geometric features for context recognition. In this section we analyzed the relationship between the hippocampal map and geometry without consideration of possible differences between the contexts (i.e., treating both contexts as if they were the same). We analyze potential contextual contributions in later sections.

To address this question, we conducted electrophysiological and calcium imaging recordings during the execution of the two-context task (electrophysiological recordings: 7 mice, 86 single-units on day 1, 6 mice, 82 single-units on day 2 and 6 mice, 69 single-units on day 3, calcium imaging: 5 mice, 839 cells on day 1, 5 mice, 833 cells on day 2, and 5 mice, 630 cells on day 3). Hippocampal geometric alignment reflecting place fields rotations at 0° and 180° was observed using both electrophysiological and calcium imaging recording methods (Fig. 2a). To further validate geometric alignment, we conducted a best match rotation (BMR) analysis[2]. In this analysis, each cell's place map was compressed to a square and compared across trials (66 pairwise comparisons for 12 trials). For each pairwise comparison, the map of Trial A was rotated 0°, 90°, 180°, and 270° and the pixel-to-pixel cross-correlation to the non-rotated map of Trial B was calculated to evaluate the similarity between the maps. The rotation that yielded the highest pixel-to-pixel cross correlation between trials was selected as the BMR (Fig. 2b). The proportion of BMR for all cells was averaged for each animal on each day. Since rectangular chambers have 180° rotational symmetry, high BMR values at 0° and 180° indicate place field alignment to geometry.

We analyzed the proportion of BMR using $3 \times 4$ repeated measures ANOVAs with day of testing (1–3) and rotation (0°, 90°, 180°, and 270°) as within subjects' factors on maps generated using single-units (Fig. 2c, left) or calcium-transients (Fig. 2c, right). Since we analyzed data across days, one electrophysiology animal was excluded from this

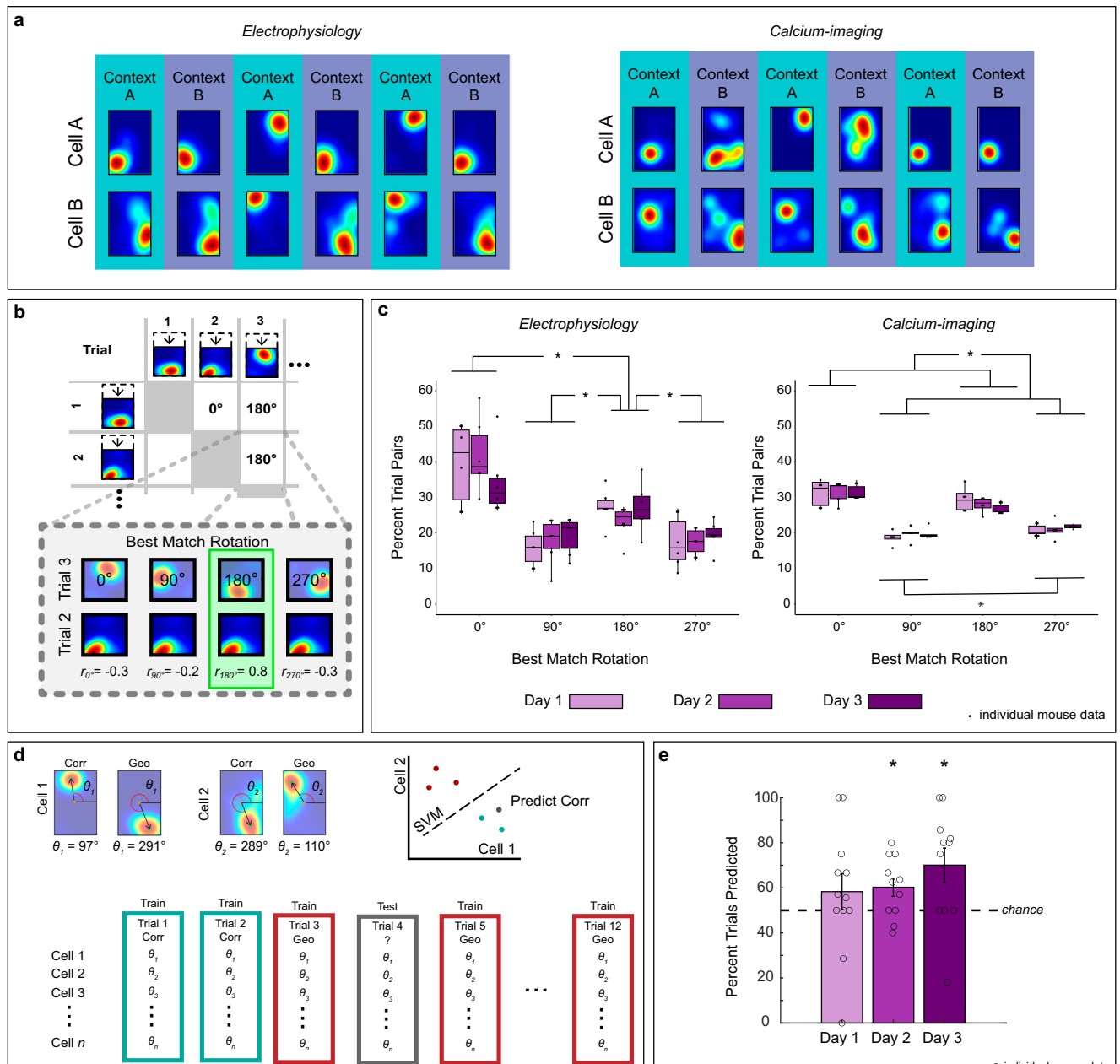

**Fig. 2 | Place field alignment to spatial geometry persists over days. a** Example place cell maps from two simultaneously recorded cells on day 3 from electrophysiology (left) and calcium-imaging (right) recordings. **b** Schematic showing best match rotation (BMR) between trials. **c** Boxplots showing distribution of BMRs across days using data from electrophysiological ($N = 6$) and calcium-imaging ($N = 5$ animals, right) recordings. The BMRs were computed as the proportion of pairwise trial comparisons for which each rotation yielded the best match, averaged per animal. Using 3 × 4 repeated measures ANOVA a significant effect of rotations was found for electrophysiological (left) and calcium-imaging (right) groups. Post hoc Rom's tests revealed that rotations at 0° and 180° were significantly higher than others (electrophysiology: 0° vs 90°: $p < 0.0303$; 0° vs 270°: $p < 0.0303$; 180° vs 90°: $p < 0.0332$; 180° vs 270°: $p < 0.0332$; calcium imaging: 0° vs 90°: $p < 0.0023$; 0° vs 270°: $p < 0.0013$; 180° vs 90°: $p < 0.0005$; 180° vs 270°: $p < 0.0023$), indicating that

the place fields aligned to geometry. Additionally, 0° vs 180° was significant only in the electrophysiology group ($p < 0.0303$) and 90° vs 270° was significant only in the calcium imaging group ($p < 0.0273$). Boxplots are composed of a horizontal line (median), a box (upper and lower quartiles), whiskers (minimum and maximum values), and dots (individual data points). **d** Schematic of heading prediction method using the center-out measure. **e** Heading prediction accuracy. The orientation of the hippocampal map predicted heading on days 2 and 3 but not on day 1 (robust one sample *t*-test relative to chance; Day 1: $p = 0.1680$; Day 2: $p < 0.0300$; Day 3: $p < 0.0280$). Bar charts represent mean ± standard error of the mean (SEM), circles represent individual animal points (Day 1 $N = 12$, Day 2 and 3 $N = 11$ animals). Black dashed line represents chance level (50%). Asterisks (*) indicate a significance value set at 0.05. Source data are provided as a Source Data file.

analysis because it only yielded data on day 1. The results from both datasets were consistent. There were no main effects of day ($p > 0.05$), or interactions between day and rotation ($p > 0.05$), but significant main effects of place cell map rotations in data from electrophysiological [$F_{(1.1, 5.5)} = 20.78$, $p < 0.0040$] and calcium imaging [$F_{(1.18, 4.74)} = 71.48$, $p < 0.0001$] recordings. Post hoc Rom's tests on

the main effect of rotation (days grouped together for each rotation) indicated that the BMR at 0° and 180° occurred more often than other rotations in both datasets ($p > 0.04$), indicating that the hippocampal map aligns to the geometry of the layout (statistical details of all comparisons are in Fig. 2c). We corroborated the geometric alignment using an alternative procedure that calculated the angle from the

center of the rectangular map to the center-of-mass of the place field (center-out measure). This analysis also yielded bimodal alignment -0° and 180° between all pairwise comparisons (Supplementary Fig. 3), indicating that geometric alignment was and did not depend on the quantification method. These data demonstrate that geometric alignment persists even after animals incorporate non-geometric features to recover heading.

We then asked if the orientation of the hippocampal map, as defined by the ensemble of orientations across individual cells, could predict search location, as previously demonstrated in single contexts[2]. We defined the orientation of the place field as the cell's center-out angle and used the orientation of C and G trials to train a support vector machine (SVM) for binary classification while excluding the trial to be predicted (Fig. 2d). Prediction was assessed for all C and G trials and averaged per animal for each day. Using a robust one-sample $t$-test with respect to chance level we found that the orientation of the hippocampal map predicted heading on days 2 and 3, but not day 1 [Fig. 2e, Day 1: $\xi = 0.53$, $p = 0.1680$; Day 2: $\xi = 0.65$, $p < 0.0300$; Day 3: $\xi = 0.85$, $p < 0.0280$]. The lack of statistical significance on Day 1 could reflect that there were fewer trials per animal, as visible reward training trials were not included. Alternatively, animals might require more experience to effectively use the cognitive map for heading retrieval. Nevertheless, the significant predictions on Days 2 and 3 strongly suggest that the orientation of the map plays a crucial role in heading retrieval.

It is worth noting that although place cells display a dominant geometric alignment in disoriented mice, there are some instances in which rotations of the hippocampal map between trials do not coincide with the rotational symmetry of the rectangle. These non-geometric rotations (90° and 270°) were also observed in the traditional, one-context reorientation task[2]. To investigate if non-geometric rotations coincided with errors, we trained a SVM using the same procedure described above, except we trained and predicted N and F trials (error trials). Error prediction was not significant on any day [Robust one-sample $t$-test relative to chance: Day 1: $\xi = 0.34$, $p = 0.2800$; Day 2: $\xi = 0.001$, $p = 0.9800$; Day 3: $\xi = 0.15$, $p = 0.6080$; Supplementary Fig. 4a]. However, since low prediction accuracy could occur because error trials are less common, we plotted the pair of digs corresponding to each BMR angle to determine what behaviors coincided with 90° and 270° (Supplementary Fig. 4b-d). We found that geometric behavioral responses (C/G and G/C), were prevalent at all BMR angles, suggesting that non-geometric rotations do not arise from error trials.

**Different CA1 cells display distinct context sensitivity**

Our behavioral results demonstrate that animals successfully use the non-geometric features to discriminate the two contexts beginning on day 1, as evidenced by the distinct digging patterns observed in each chamber during test trials (Fig. 1d). However, it is unclear if context recognition is represented in the hippocampus through location remapping to support this behavior. One possibility is that cells display geometric alignment within each context (i.e., rotating 180° along the same axis on a trial-by-trial basis), but display location remapping across contexts (e.g., rotating on different axes in each context; Fig. 3a, left panel). A second possibility is that one subpopulation of cells exhibits context-insensitive geometric rotations (e.g., same configuration in both contexts), while another subpopulation displays context-sensitive location remapping (Fig. 3a, right panel). Note that geometric alignment refers to 180° rotations that occur within context, whereas we refer to remapping to describe the non-geometric shift in the preferred firing location across contexts.

To test these possibilities, we assessed place field similarity across contexts after we aligned each cell's maps relative to the geometric axis that coincided with the rotational symmetry of the rectangle (0° or 180°), which the previous BMR analysis reflected as the dominant

rotational pattern of most CA1 cells during reorientation. In other words, place field maps for each cell were aligned to the same orientation within each context by rotating (or not rotating) them to maximize similarity across all trials (Fig. 3b). Then, an average aligned map was computed for each context and the two average maps were aligned relative to each other (e.g., best match between 0° or 180° rotations). Context similarity was defined as the highest pixel-to-pixel cross-correlation between the rotated or non-rotated average aligned maps in each context. Context similarity was only defined for cells that had place fields in both contexts (2291 cells from calcium-imaging and 235 single-units across 12 animals and 3 days; $n = 2526$). The distribution of these context similarity values displayed a strong leftward skewness (Fig. 3c). This indicated that while most cells displayed stability across contexts (i.e., place fields were similar after they were aligned to the same orientation), some cells remapped across contexts (i.e., maps were dissimilar across contexts even after alignment).

To further investigate if the context similarity distribution shown in Fig. 3c reflects the presence of distinct cell types, we employed a modeling approach. Non-linear modeling has been applied to characterize the dynamics of other processes, such as learning curves[41], psychophysics[42], and biological processes[43]. Importantly, modeling approaches have also been used to characterize the behavioral relevance of distinct neuronal cell types[44]. Here, we divided the context similarity values into deciles and created plots comparing the within context vs. across context correlations for each cell within each decile (Supplementary Fig. 5a). Only cells that had place fields in at least two trials per context were included to compute a within context comparison (11 cells were excluded for not having a minimum of 2 trials in each context, $n = 2515$ cells). Next, we calculated the Pearson $R$ value between the within and across context correlation for each decile (Supplementary Fig. 5b). Since this relationship was not linear, we tested several models and found that the most suitable one was a non-linear asymptotic regression model. Decile 1, comprising cells with context similarity values $\leq 0.3$ emerged as the most suitable parameter to separate distinct subpopulations because it coincided with the function's half-life (point at which the function reaches half of its maximum), the function's root (value that turns the function to zero), and the point that generated the largest function growth rate among all deciles. This indicated that cells in Decile 1 exhibited unique remapping properties distinct from the other deciles (for details about the model see Supplementary Fig. 5-6 and Supplementary Table 4). This conclusion was further validated by additional analyses described in the following sections. Notably, this threshold has also been used in previous studies involving mice to differentiate between stable and unstable place cells[45,46].

Using the cutoff value of 0.3, we divided the cells shown in the context similarity distribution (Fig. 3c) and categorized neurons as "feature-sensitive" (FS) if similarity across contexts after alignment was $\leq 0.3$ (i.e., cells displayed location remapping across contexts), or as "feature-insensitive" (FI) if similarity across contexts was $>0.3$ (i.e., cells displayed stability of place fields across contexts). It is worth noting that these definitions only account for changes in the location of firing, not changes in the rate of firing, which we address later in the study. Using this classification, we found that the average context similarity was $0.654 \pm 0.003$ for FI cells and $0.160 \pm 0.006$ for FS cells. Additionally, proportions of FS and FI cells were similar in data from electrophysiology and calcium-imaging recordings (Fig. 3d and Supplementary Fig. 7). Examples of average aligned Context A and Context B maps recorded from FI and FS cells are shown in Fig. 3e. It is important to acknowledge that if cells are stable in each context (all maps provide maximal correlation at 0°), but all their maps rotate 180° across context, our alignment method might mistakenly classify these cells as FI when they should be classified as FS cells. To address this potential misclassification, we examined cells exhibiting high stability in each context (those not requiring alignment). Among all recorded

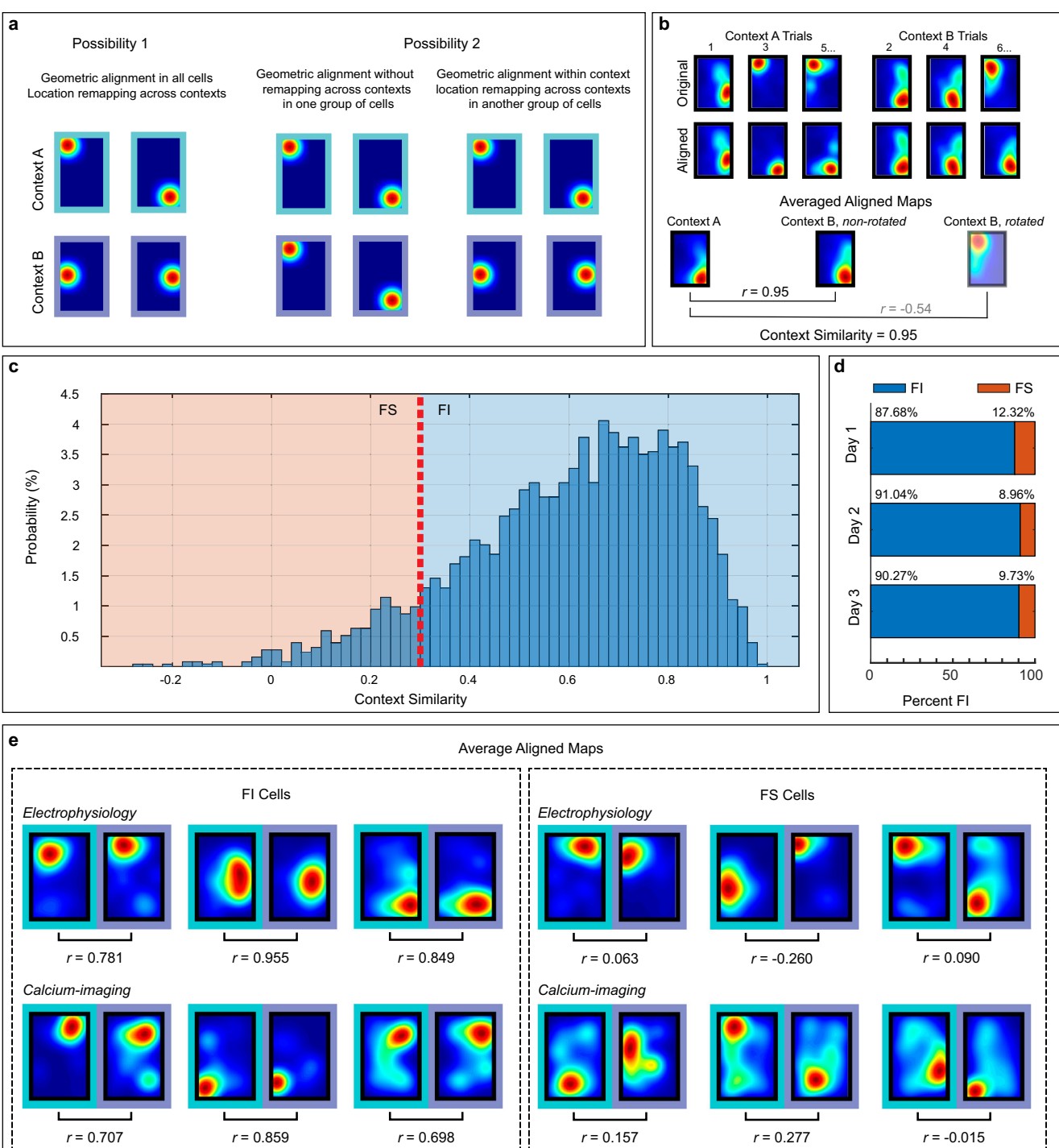

**Fig. 3 | Different CA1 cells display distinct context sensitivity. a** Hypothetical neural representations of context recognition. **b** Schematic of map alignment procedure. **c** Distribution of context similarity scores across context for cells with at least one place field in both contexts ($n = 2526$ cells). Red dashed line represents threshold to separate cells as feature-sensitive (FS) or feature-insensitive (FI).

**d** Proportion of FI and FS cells recorded on day 1 ($n = 925$ cells), day 2 ($n = 915$ cells), and day 3 ($n = 699$ cells). **e** Examples of average aligned maps of FI (left) and FS (right) cells from electrophysiology (top) and calcium-imaging (bottom) recordings, along with the corresponding context similarity measures. Source data are provided as a Source Data file.

cells, only one cell displayed the described pattern on day 1, but this pattern did not recur on subsequent days (Supplementary Fig. 8). Consequently, we attributed this occurrence to chance and concluded that this cell was not representative of our dataset.

We next examined whether differences in place field stability across contexts reflected distinct spatial properties in these cells. Due to the low throughput of electrophysiological techniques, we limited

cell type analyses to data recorded using calcium imaging. We computed spatial information content, a parameter that describes the information about animal's position contained in a cell's activity pattern[47]. Our analysis revealed no discernible differences between FI and FS cells (Supplementary Fig. 9a) or across deciles (Supplementary Fig. 9b) in spatial information content. Additionally, both FI and FS cells displayed similar geometric alignment within each context

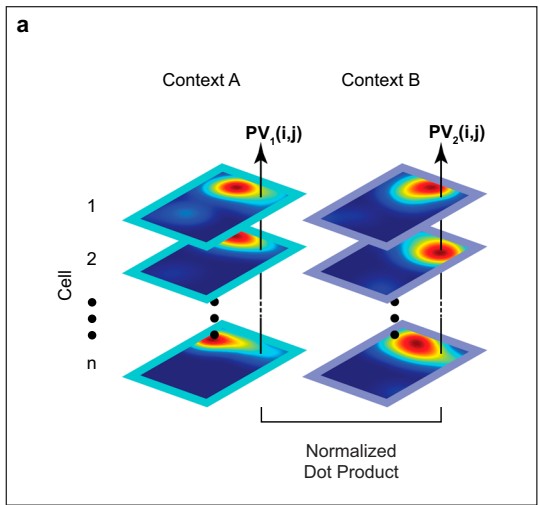
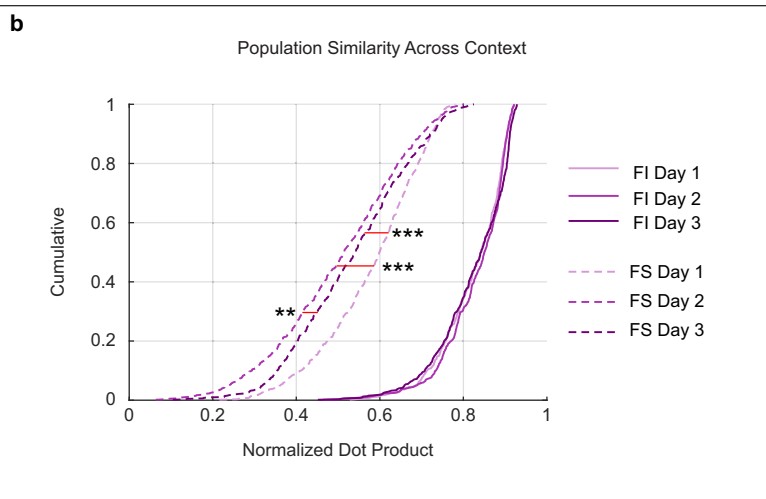
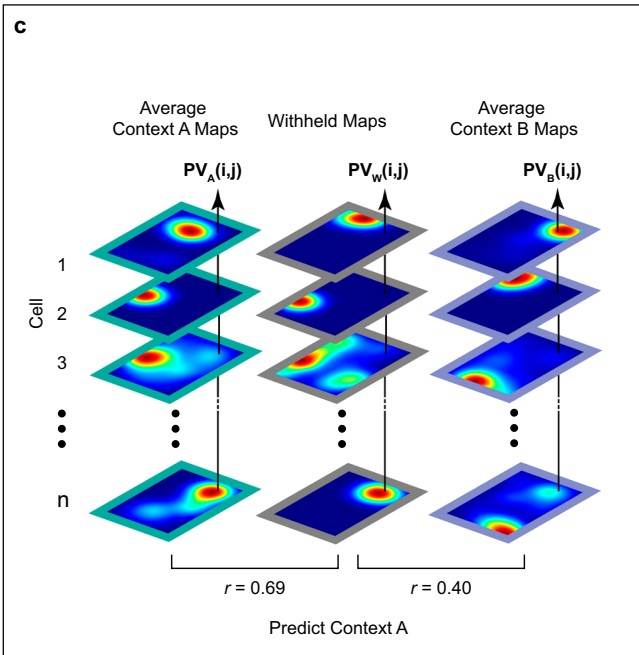
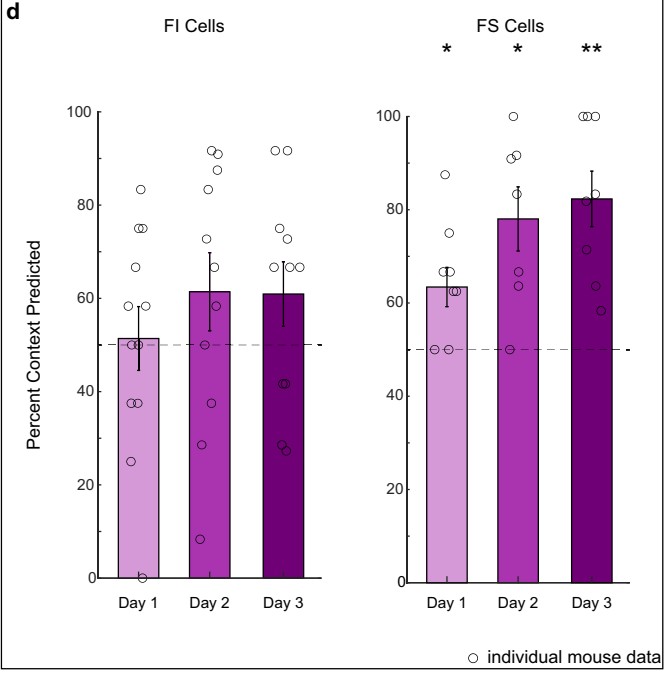

**Fig. 4 | FS cells display lower context similarity than FI cells and predict context across days. a** Schematic illustrating method for measuring population similarity across contexts. **b** Distributions of normalized dot product of population data as cumulative proportions collected from FI (solid) or FS (dotted) cells across days. A two-way Ranked ANOVA and post hoc Rom's tests revealed that FI population spatial representations were equally similar between contexts across days (Day 1 vs 2: $p = 0.2250$; Day 1 vs 3: $p = 0.6750$; Day 2 vs 3: $p > 0.999$). Conversely, FS population representations had a lower dot product on days 2 and 3 than day 1, indicating more dissimilarity in the spatial maps at the population level across time

(Day 1 vs 2 and Day 1 vs 3: $p = 0.0001$; Day 2 vs 3: $p > 0.0030$). **c** Schematic of *p*opulation vector (PV) analysis for predicting context. **d** Context prediction accuracy using FI cells (left panel) or FS cells (right panel). FS cells reliably predict context across days [FS (right; panel): Day 1 ($N = 9$): $p < 0.0001$; Day 2 ($N = 7$): $p < 0.0001$; Day 3 ($N = 8$): $p < 0.0001$], whereas FI do not [FI (left panel): Day 1 ($N = 12$): $p = 0.1700$; Day 2 ($N = 11$): $p = 0.1460$; Day 3 ($N = 11$): $p = 0.1880$]. *P* levels are obtained using robust one-sample t-tests. Bar charts represent mean ± SEM, circles represent individual data points. Asterisks represent: $p \leq 0.05$ (*), $p \leq 0.01$ (**), $p \leq 0.001$ (***). Source data are provided as a Source Data file.

(Supplementary Fig. 10 and 11 and Supplementary Table 5), indicating that the only difference between these cells was their pattern of remapping across contexts.

## FS cells display lower population context similarity than FI cells and predict context across days

The previous analyses illustrate differences in place field stability across contexts in FI and FS cells. To determine if this characteristic was also observed at the population level, we evaluated the context similarity of FI and FS place field maps across testing days using normalized dot products. This measure computed the dot product of population vectors at corresponding locations in the stacked average

aligned maps in Context A and B across cells (Fig. 4a). A larger dot product indicates greater similarity between population spatial representations[48]. As expected from single cell data, the population similarity in spatial representations across contexts was lower in FS than FI cells across days (Fig. 4b). Additionally, this analysis also uncovered an effect of learning, in which population similarity of FS cells decreases with training. These observations were corroborated with a two-way Ranked ANOVA with day of testing (day 1, 2, and 3) and cell type (FI and FS) as between factors. The main effects of cell type and day as well as the interaction between these variables were significant [context sensitivity: $V(1) = 7688.92$, $p < 0.0001$; day: $V(2) = 27.12$, $p < 0.0001$; interaction: $V(2) = 57.68$, $p < 0.0001$]. Post hoc

Rom tests indicated that FI population spatial representations were equally similar between contexts across days [Day 1 vs 2: V(1) = 1.78, $p = 0.2250$; Day 1 vs 3: V(1) = 1.21, $p = 0.6750$; Day 2 vs 3: V(1) = 0.30, $p > 0.999$]. Conversely, FS population spatial representations across contexts were more dissimilar on days 2 and 3 than on day 1 [Day 1 vs 2: V(1) = 9.11, $p < 0.0001$; Day 1 vs 3: V(1) = 6.10, $p < 0.0001$; Day 2 vs 3: V(1) = 3.25, $p < 0.0030$]. Finally, to further validate the properties of FS cells, we plotted the cumulative distributions of normalized dot products dividing the data into deciles according to context similarity values and found that the cumulative distribution of FS cells (represented by Decile 1) was markedly lower from the rest of the population (Supplementary Fig. 12a). In summary, these results indicate that differences in spatial representations across contexts in FS and FI cells are also observed at the population level—the population of FS cells shows pronounced remapping between contexts, whereas the population of FI cells displays high stability. Furthermore, the population similarity of FS cells decreases with training, whereas no such modulation was observed in the population similarity of FI cells.

Given that FS cells display different spatial representations across contexts at the single cell and population level, we hypothesized that the firing location of FS cells could be used to predict context. To test this, we created an average aligned Context A map and an averaged aligned Context B map for each cell, withholding one trial at a time, and calculated the average normalized dot product between the withheld map and the average aligned map as shown in Fig. 4c. The higher dot product comparison predicted context of the withheld trial. As expected, given the high stability across contexts in the FI cells, these neurons failed to predict context above chance on any day as evidenced by a robust one-sample $t$-test [Fig. 4d, left; Day 1: $\xi = 0.17$, $p = 0.6480$; Day 2: $\xi = 0.46$, $p = 0.1460$; Day 3: $\xi = 0.49$, $p = 0.1880$]. However, FS cells predicted context on days 1 to 3 [Fig. 4d, right; Day 1: $\xi = 0.76$, $p < 0.0001$; Day 2: $\xi = 1.38$, $p < 0.0001$; Day 3: $\xi = 1.35$, $p < 0.0001$]. These results were also validated by calculating context prediction as function of context similarity (Supplementary Fig. 12b), which further supported the role of FS in context discrimination. In summary, these data indicate that FS cells respond to the unique features in each chamber, which could serve as a substrate for context recognition.

### FI and FS cells display coherent patterns of alignment within and across contexts

Our data display place field rotations across trials in both FI and FS cells. However, it is unclear whether these rotations occur coherently in all cells, reflecting population coherence, or independently. If the orientation of the map is critical for heading retrieval, then FI cells, which primarily align to geometry, should consistently rotate across contexts and days displaying coherency. However, the location remapping across contexts in FS cells could reflect two remapping possibilities: (1) Global re-organization of place fields across contexts (random remapping), or (2) Coherent rotation of place fields across contexts (coherent remapping). As shown in the schematic of Fig. 5a, if there were random remapping, we would expect that each cell remaps independently of each other; however, if there were coherent remapping, we would anticipate the majority of place fields to undergo similar remapping, demonstrating consistent rotations across trial pairs.

To test these possibilities, we assessed the proportion of BMR angles (0°, 90°, 180°, and 270°) across trial pairs for each cell type (FI and FS) within and across context using compressed squared place maps (Fig. 5b). This analysis yielded 4 proportions based on the 4 angles used for analysis ranked by frequency, which are shown in Fig. 5c, d. Coherent remapping in the population would be indicated by a predominant proportion of cells sharing the same BMR angle between trial pairs. Global remapping would be represented by a distribution of BMR proportions around chance level (25%). We plotted

the proportion of cells displaying the most frequent to the least frequent BMR angle and found that the 1st BMR was significantly above chance for FI and FS cells (Fig. 5d). Interestingly, the 2nd BMR was also above chance, especially on days 1 and 2. A robust one-sample, one-tailed $t$-test with respect to chance level revealed that the 1st and 2nd BMR were significant across days ($p < 0.05$), except for FS cells on day 3, where only the 1st BMR reached significance ($p < 0.05$). The 3rd and 4th BMR were below the chance level across conditions and days ($p > 0.05$, Supplementary Table 6) and were therefore excluded from further analysis.

To investigate the dynamic relationship of the 1st and 2nd BMR, we conducted a 4-way ANOVA with day of testing (1 to 3), cell type (FI and FS), and contextual condition (within and across contexts) as between factors, and BMR (1 and 2) as a within factor. The main effects of the four factors as well as the interaction between cell type, day, and BMR were significant ($p < 0.05$; Supplementary Table 7a for complete details of ANOVA). Examination of this interaction was performed using a trend analysis (Supplementary Table 7b). For FI cells, there was no significant change over days in the proportion of 1st or 2nd BMRs ($p > 0.05$). For FS cells, the proportion of cells in the 1st BMR increased, while the proportion of cells in the 2nd BMR decreased linearly across days (Fig. 5e, $p < 0.0001$). These results were consistent within and across contexts.

To examine the angular relationship between the 1st and 2nd BMR, we plotted the joint distribution of best match rotations, which was calculated for each trial pair and averaged per animal (Fig. 5f). We hypothesized that if geometry played a role in controlling population coherency, the 1st and 2nd BMR would align along the same geometric axis (e.g., if cells displayed a 180° angular rotation in the 1st BMR, the 2nd BMR would remain stable at 0°). Conversely, if there were no relationship between the 1st and 2nd BMR, these rotations would occur independently, showing no pattern. A Wilcoxon one-sample, one-tailed test revealed that the relationship between 1st and 2nd BMR in FI cells was controlled by geometry within and across contexts, which is evident in higher average values along the same geometric axis (0° and 180°). On the other hand, FS cells displayed a less clear relationship between 1st and 2nd BMR across days (Supplementary Table 7c). This coincides with the fact that the 1st BMR increases with learning, becoming the only coherent rotation above chance by day 3 in FS cells. These results support previous findings that demonstrate multiple remapping coherencies in changing environments, where distinct coherent subpopulations represent different aspects of the task[49].

Qualitative examination of the marginal probabilities of the 1st BMR indicates that FI cells display rotations at 0° and 180° for both within- and across-context comparisons (Fig. 5f). Conversely, the 1st BMR in FS cells displays alignment to opposite geometric axis within and across contexts. In summary, the consistent coherency between 1st and 2nd BMR within and across context in FI cells corroborates their involvement in stable representations of geometry. Conversely, the learning-modulated relationship between the 1st and 2nd BMR along with the contrasting pattern of rotations of the 1st BMR within and across context in FS cells, may reflect these cells' involvement in the formation of feature/reward associations, which facilitates context recognition and improves heading retrieval over time.

### FI cells can predict heading across time

An advantage of calcium imaging techniques is that it allows tracking of the same neurons over time, which is achieved by superimposing the spatial footprints of identified cells within the calcium-imaging field of view across days (Fig. 6a). Therefore, we tracked FI and FS cells across days to determine if their remapping characteristics across contexts remained stable. For this analysis we excluded cells that did not have place fields in both contexts (registered cells: Day 1 to Day 2 $n = 289$; Day 1 to Day 3 $n = 236$; Day 2 to Day 3 $n = 272$). We anticipated

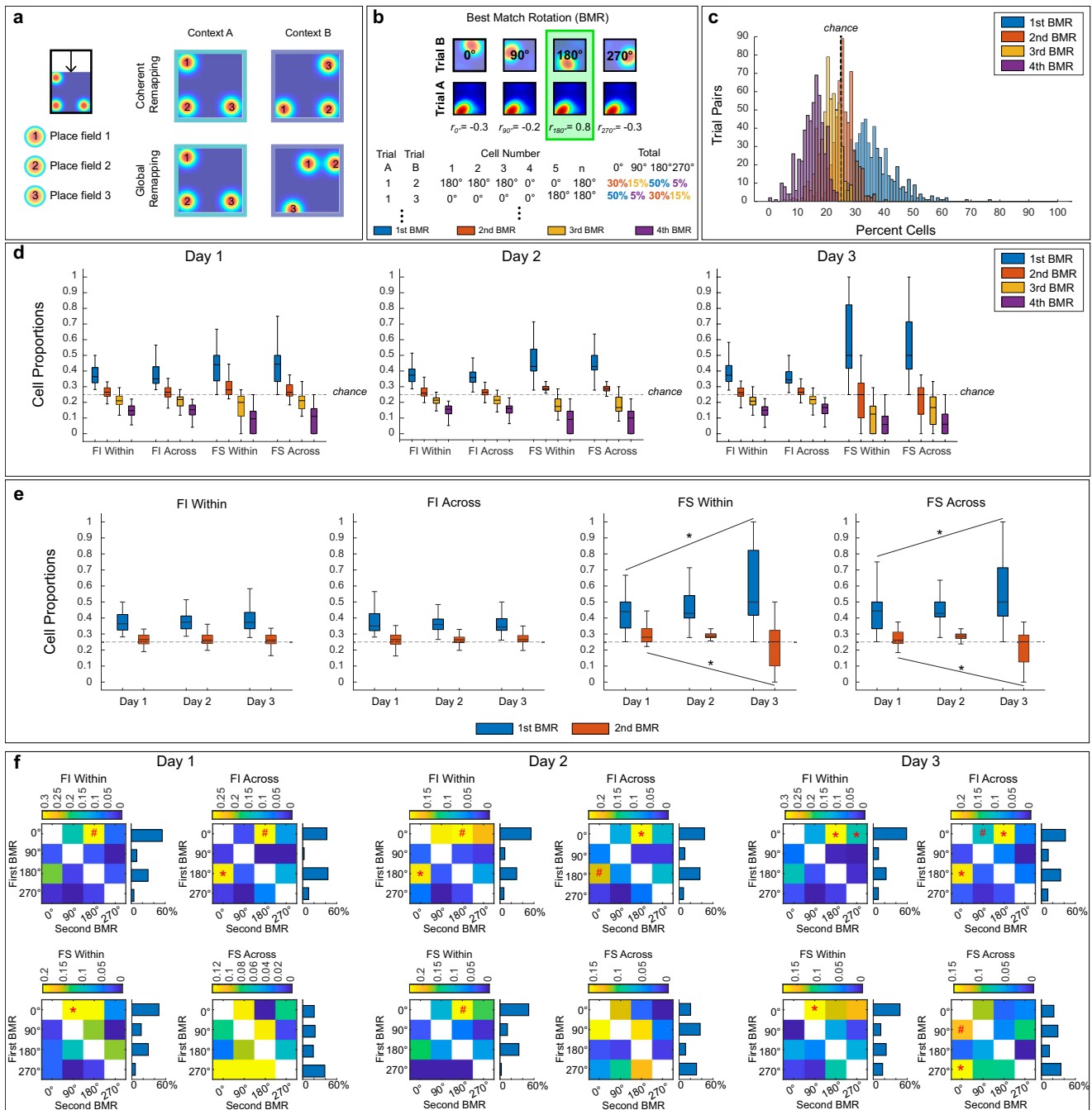

**Fig. 5 | FI and FS cells display coherent patterns of alignment within and across contexts. a** Schematic of remapping possibilities across trials. Top: Coherent remapping showing all cells remapping consistently relative to each other. Bottom: Global remapping showing cells remapping independently of each other.
**b** Schematic of coherency analysis using proportion of best match rotations (BMR). Ranked proportions are shown in blue (1st), red (2nd), yellow (3rd), and 4th (purple). **c** Histogram of BMR cell percentages. Black line at 25% indicates proportion of BMRs expected from chance. **d** Boxplots showing proportion of cells within each BMR per cell type (FI or FS) and trial comparison (within or across) for day 1 (left), day 2 (center), and day 3 (right) [FI trial pairs within: day 1: $N = 51$, day 2: $N = 69$, day 3: $N = 124$; FI across: day 1: $N = 67$, day 2: $N = 89$, day 3: $N = 150$; FS within day 1: $N = 51$, day 2: $N = 69$, day 3: $N = 124$; FS across day 1: $N = 67$, day 2: $N = 89$, day 3: $N = 150$].

Black dashed line indicates chance level. **e** Boxplots showing 1st and 2nd BMRs (same data as **d**). The 1st and 2nd BMRs in FI cells are consistently above chance within and across contexts. In FS cells, the 1st BMR increases and the 2nd BMR decreases across days (4-way ANOVA; $p < 0.05$, statistical details in Supplementary Table 7a, b). Boxplots display the median (horizontal line), upper and lower quartiles (boxes), and minimum and maximum values (whiskers). **f** Joint probabilities of the 1st and 2nd BMRs shown as matrix heatmaps for each cell type (FI and FS), and trial comparison (within and across contexts) for day 1 (left), day 2 (center), and day 3 (right). Rows indicate 1st BMR and columns indicate 2nd BMR. Wilcoxon one-sample, one-tailed tests details in Supplementary Table 7c. Marginal probability of 1st BMR is shown at the right of each matrix. Asterisks represent a significant value set at 0.05, and # indicates a trend approaching this level.

that since the firing location of FI cells is constant and responds to the geometry of the layout, these cells would preserve their identity across time. On the other hand, if FS cells are indeed involved in the formation of feature-reward associations, they would exhibit dynamic activity patterns, as previously demonstrated in hippocampal subpopulations during different learning paradigms[50]. In agreement with this idea, we found that the majority of FI cells maintained their identity, whereas FS cells did not (Fig. 6b). To rule out the possibility that differences in

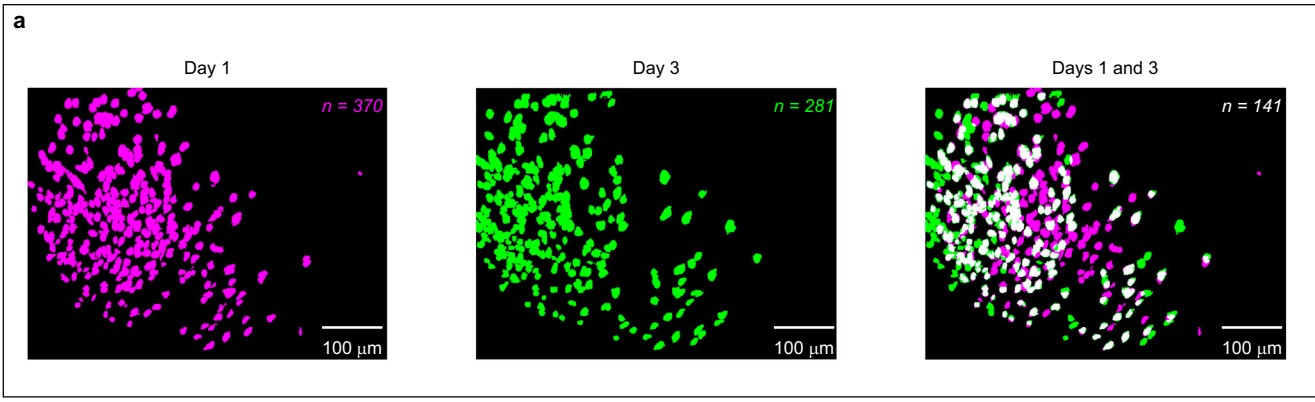

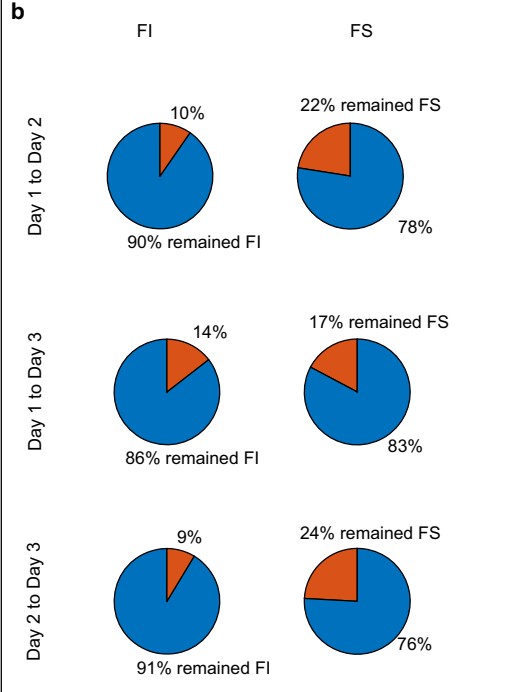

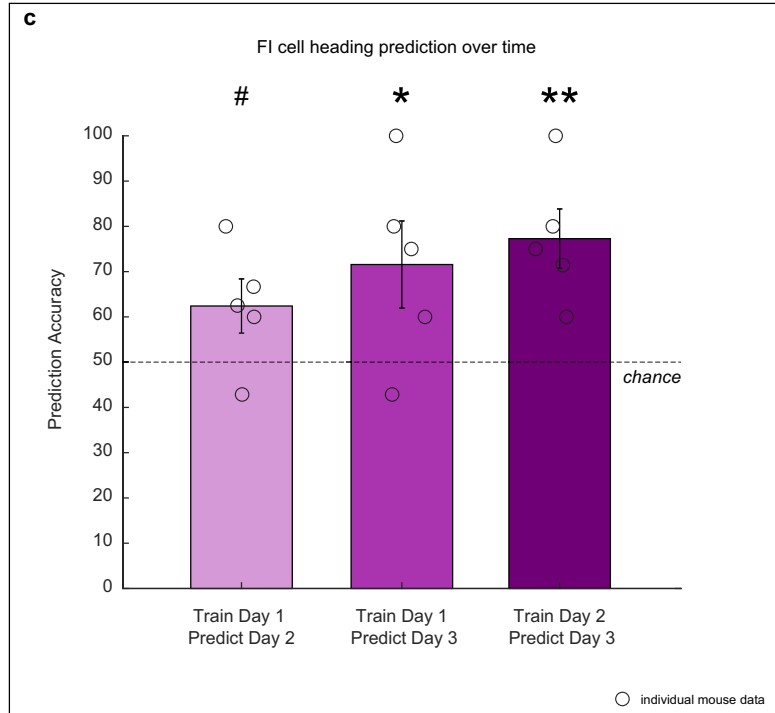

**Fig. 6 | FI cells can predict heading across time. a** Spatial footprints of cells identified from a representative animal on day 1 (magenta, left), day 3 (green, center), and cells identified on both days 1 and 3 (white, right) illustrating method to track cells across time. **b** Pie charts showing stability of cell identity across time. Top panel shows FI and bottom panel FS percentages of registered cells that displayed stable identity (same pattern of stability across days) or unstable identity (distinct identity across days). Note that the identity of FI remains stable across days, whereas only a small percentage of FS cells retain their identity. **c** Heading prediction accuracy using registered cells' center out angles across days using robust one-sample t-test relative to chance. Left bar trained a classifier with neural data from day 1 to predict heading on day 2 ($N = 5$ animals), center bar trained with data on day 2 to predict heading on day 3 ($N = 5$ animals), and right bar trained with data on day 1 to predict heading on day 3 ($N = 5$ animals). The circles indicate individual animal performance and error bars represent ± standard error of the mean (SEM). Asterisks represent: $p \le 0.05$ (*), $p \le 0.01$ (**), $p \le 0.001$ (***), # indicates a trend approaching significance ($p = 0.08$). Source data are provided as a Source Data file.

identity stability were due to distinct registration quality of FI and FS cells, we divided the cells by deciles according to their context similarity values (Fig. 3c). We found no differences in the registration scores across deciles (Supplementary Fig. 13a, $p > 0.05$). Additionally, since FS cells are defined by a more constrained range of correlations than FI cells, similar increases in stability may artificially increase the likelihood that FS cells shift identity across days. To rule out this possibility, we calculated the absolute difference in context similarity values across days by separating the cells into deciles as described above. Our findings show that FS cells indeed show greater changes in context similarity values than cells in other deciles across time (Supplementary Fig. 13b), supporting the idea that FS cells have inherent dynamic properties. These data show that changes in FS identity were not due to poor registration quality or their constrained correlation

values. In summary, information about environmental geometry is represented by a consistently active ensemble of cells, reflecting the constant properties of geometry. Conversely, the dynamic activity patterns of FS ensembles likely reflect the formation of feature-reward associations that develop over time.

Since FI cells preserve their remapping properties across days, we then tested if the orientation of place cell maps of these cells on day 1 could predict heading (C or G) on subsequent days. To this end, we used the place fields' center-out angle from maps obtained on day 1 to train a SVM and predicted heading behavior based on trial maps on days 2 or 3, or we trained the SVM with maps from day 2 to predict day 3. Robust one-sample *t*-test relative to chance showed that training the classifier with data on days 1 or 2 significantly predicts day 3 [Day 3 from Day 1: $\xi = 1.36$, $p < 0.0480$; Day 3 from Day 2: $\xi = 3.81$, $p < 0.0001$;

Fig. 6c], whereas training on day 1 and predicting on day 2 reveals a trend [Day 2 from Day 1: $\xi = 2.50$, $p = 0.0800$; Fig. 6c]. These results corroborate that the hippocampal map alignment underlies heading retrieval. Moreover, these data suggest that the representation of geometry forms immediately, but the effective use of this map improves with experience.

## Firing rate predicts context and reorientation behavior by integrating featural and geometric information

Our behavioral results indicated that heading retrieval and context recognition are driven by dissociable cues on day 1, but animals learn to integrate features and geometry on days 2 and 3 in order to distinguish between geometrically equivalent locations within context. How does this integration happen at the neural level? Previous studies have found that trial-dependent features are often encoded through rate remapping[39,51]. Furthermore, spatial and goal-directed information is integrated through rate changes in CA1[52], suggesting that rate is strongly influenced by learning. Therefore, we tested if spatial geometry and featural information could be integrated in the active ensemble through rate changes in our reorientation task. These analyses were conducted using electrophysiological data because calcium traces only provide an indirect measure of activity, which in our experiments was not sufficient to detect rate changes (Supplementary Fig. 14). Since we previously found that disoriented mice sample the environment in a stereotyped manner (i.e., sampling edges around the environment)[16], here we report the mean firing rate, which has been previously shown to be more sensitive to stereotyped exploration[53,54]. However, similar results were obtained with peak firing rate (Supplementary Fig. 15).

We plotted the absolute difference in mean firing rate for each cell between trials as a rate matrix (Fig. 7a). The across-context rate differences increased over days, but the within-context rate differences did not (Fig. 7b, c). To quantify this effect, the average rate differences for within- and across-context comparisons were calculated for each cell. A two-way robust repeated measures ANOVA using context comparison (within vs. across) as a within factor, day of testing (1, 2, and 3) as a between factor, and absolute rate difference per cell (rate remapping) as a dependent measure showed that there was no effect of day [$F_w(2) = 0.52$, $p = 0.6294$], but a significant effect of context comparison [$F_w(1) = 18.60$, $p < 0.0001$] and an interaction between day of testing and context comparison [$F_w(2) = 2.90$, $p < 0.0351$]. Post hoc Rom's tests showed that within and across comparisons were not different on day 1 ($p = 0.4075$) but were significantly different on days 2 ($p < 0.0039$) and 3 ($p < 0.0022$). Similar findings were obtained using different occupancy thresholds (Supplementary Fig. 15 and Supplementary Table 8), providing further confidence that the short length of our trials did not influence these results. Critically, we observed similar firing rate changes in both FI and FS cells, suggesting that rate remapping occurred in the entire active ensemble (Supplementary Fig. 16a, b). Together, these findings indicate that rate remapping increases across context with training.

To further establish the relationship between the rate code and the use of features for context recognition and heading retrieval, we performed two classification analyses in our electrophysiological group. To explore the temporal progression at which featural information is integrated into the rate code for context recognition, we initially examined whether the firing rate on each trial could predict context-related behavior. To this end, we used the cells' mean firing rates in trials in which animals correctly identified the reorientation context as demonstrated by digging in the correct axis in each chamber (i.e., digs in C or G), to train a SVM for context prediction using a leave-one-out procedure (Fig. 7d). Prediction accuracy was assessed per session (Day 1: 7 animals, Day 2 and 3: 6 animals). A robust one-sample *t*-test with respect to chance demonstrated that the firing rate had no predictive power for context on day 1. However, the

prediction became significant on both days 2 and 3 [Fig. 7e, Day 1: $\xi = 0.89$, $p = 0.0640$; Day 2: $\xi = 1.46$, $p < 0.0001$; Day 3: $\xi = 2.45$, $p < 0.0001$].

Then, we assessed if the rate code also had heading predictive power. We again used C and G trials combining contexts to train a SVM classifier to determine if mean firing rate could predict when animals dug in the rewarded (C) vs. unrewarded side of the chamber (G). A robust one-sample *t*-test with respect to chance indicated that heading prediction showed significance on days 2 and 3 [Fig. 7f, Day 1: $\xi = 1.32$, $p = 0.1720$; Day 2: $\xi = 2.44$, $p < 0.0001$; Day 3: $\xi = 1.91$, $p < 0.0001$]. These results indicate that the rate code incorporates information about context and heading as animals gain experience with the task. Notably, there were no differences in prediction of heading or context by cell type (Supplementary Fig. 16c, d). These findings support the idea that rate remapping integrates featural and geometric information for efficient reorientation in the entire active ensemble.

## Discussion

In this study, we aimed to uncover the cognitive and neural mechanisms underlying heading retrieval and context recognition. To this end, we employed a two-context reorientation task that required attention to specific features and reward locations to distinguish between two contexts. Our behavioral findings on day 1 align with our prior research[33], showing that initially mice rely solely on geometric cues to determine their heading direction and features to differentiate between contexts. However, on days 2 and 3 animals learn the directional value of featural cues and incorporate these cues to distinguish between symmetrical locations. At the cellular level, we observe that the hippocampal map aligns to the shape of the chamber in both contexts, showing the same failure to resolve geometric ambiguities that is seen in behavior. This geometric alignment predicts the animals' digging behavior, implicating the orientation of the map in heading retrieval. Notably, the majority of cells, termed feature insensitive (FI), demonstrate no sensitivity to context-specific features, having place fields that align along the same geometric axis in both contexts. However, a minority of cells, termed feature sensitive (FS), exhibit place fields with contrasting patterns of rotation within and across contexts, indicating their responsiveness to featural information. Location remapping across contexts enables FS cells to predict context on a trial-by-trial basis, suggesting their involvement in context recognition. Notably, the FS remapping also shows increased coherence with learning, implicating these cells in the formation of feature/reward associations. In contrast, FI cells maintain consistent activity patterns across days, a crucial characteristic for encoding static elements of the environment, like geometry. Lastly, changes in firing rate in the whole active ensemble across contexts gradually emerge, aiding in predicting both context and dig location. These findings, which are summarized in Fig. 8, suggest that rate changes integrate geometric and featural information across the entire ensemble through learning, thus facilitating efficient reorientation.

The perception of geometry involves incidental learning independent of reinforcement (e.g., when animals are placed in an environment, they immediately perceive its shape). In our study, this is illustrated by the swift alignment of FI cells to geometry. This activity pattern is stable over time (i.e., a cell showing stability across contexts on day 1 continues displaying the same characteristic on day 3). Remarkably, this stability allows prediction of heading behavior across days (i.e., the orientation of the map on day 1 can be used to predict whether mice dig in C or G cups on day 3). This further indicates that geometric representations are enduring, providing a substrate for heading retrieval. At first glance, these results may suggest that geometric representations involved in heading retrieval are hardwired. However, it is worth noting that there is evidence showing that animals raised in circular environments do not use geometry to reorient. For

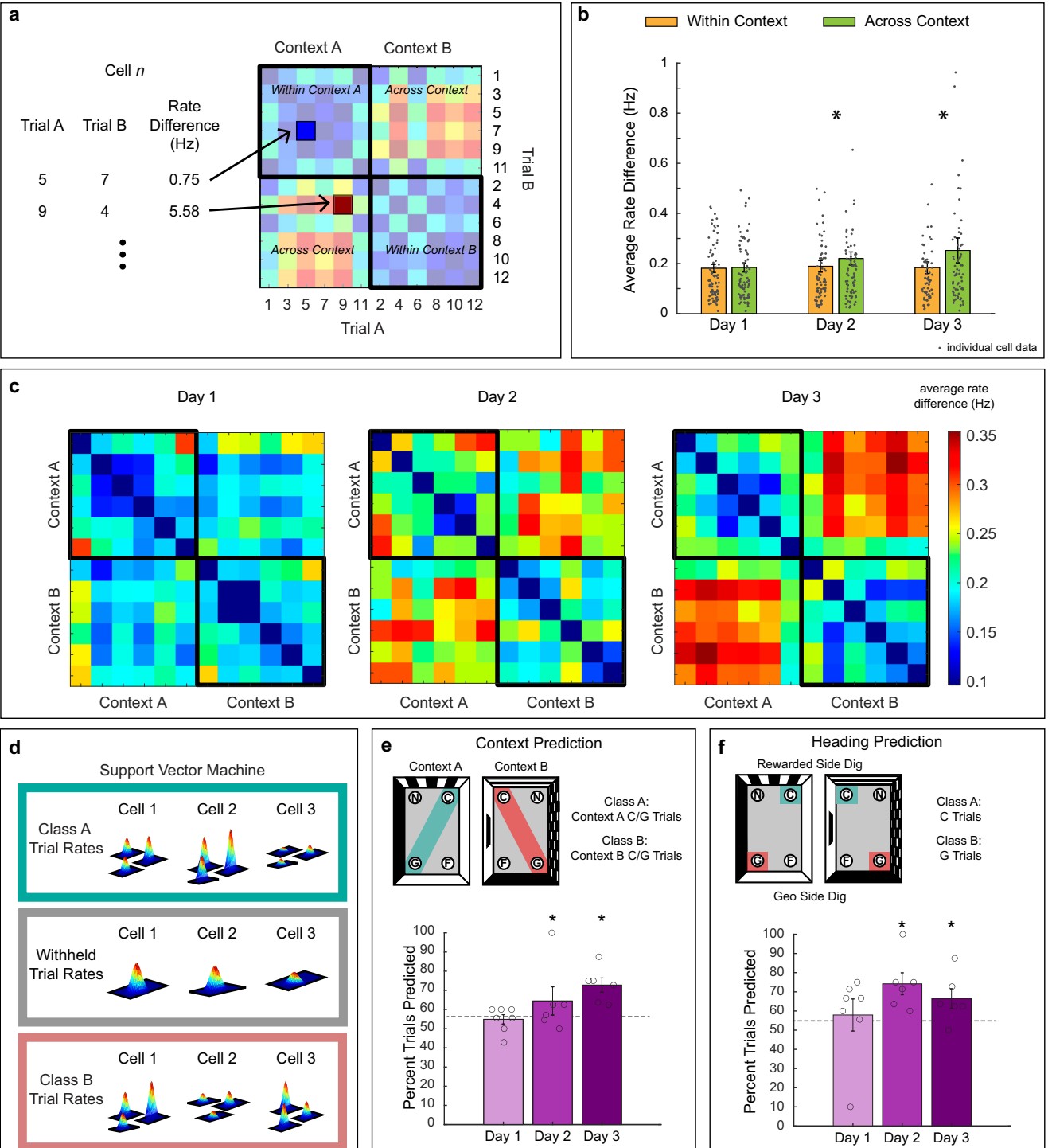

**Fig. 7 | Firing rate predicts context and reorientation behavior by integrating featural and geometric information. a** Schematic of method used to calculate heatmap rate matrices based on the absolute rate difference between trials. Quadrants with black contour depict rate differences within context (upper quadrant Context A and bottom quadrant Context B). Quadrants without contour depict rate differences across contexts. Rate matrices are computed for each cell and averaged per animal. **b** Bar chart showing average rate difference within and across contexts trial comparisons on days 1 ($N = 84$ cells), day 2 ($N = 82$ cells), and day 3 ($N = 69$ cells). Dots indicate individual cell data. Robust two-way repeated measures ANOVA with post hoc Rom's test showed that comparisons within and across context were not different on day 1 ($p = 0.4075$) but were significantly different on days 2 ($p < 0.0039$) and 3 ($p < 0.0022$). **c** Same as (**b**),

represented as heatmap matrices of average rate difference across animals recorded on day 1 (left), 2 (center), and 3 (right). **d** Schematic of prediction method. **e** *Top:* Illustration of trials used to predict context. *Bottom:* SVM prediction accuracy on day 1 ($N = 7$ animals; $p = 0.078$), day 2 ($N = 6$ animals; $p < 0.031$), and day 3 ($N = 6$ animals; $p < .016$). **f** *Top:* Illustration of trials used to predict heading. *Bottom:* SVM prediction accuracy on day 1 ($N = 7$ animals; $p = 0.148$), day 2 ($N = 6$ animals; $p < 0.016$), and day 3 ($N = 6$ animals; $p < 0.031$). $P$ values reflect one-sample robust t-tests relative to chance. Bar charts show prediction for each animal ± standard error of the mean (SEM) and circles indicate individual animals. C correct, G geometric error. Asterisks (*) represent $p \leq 0.05$. Source data are provided as a Source Data file.

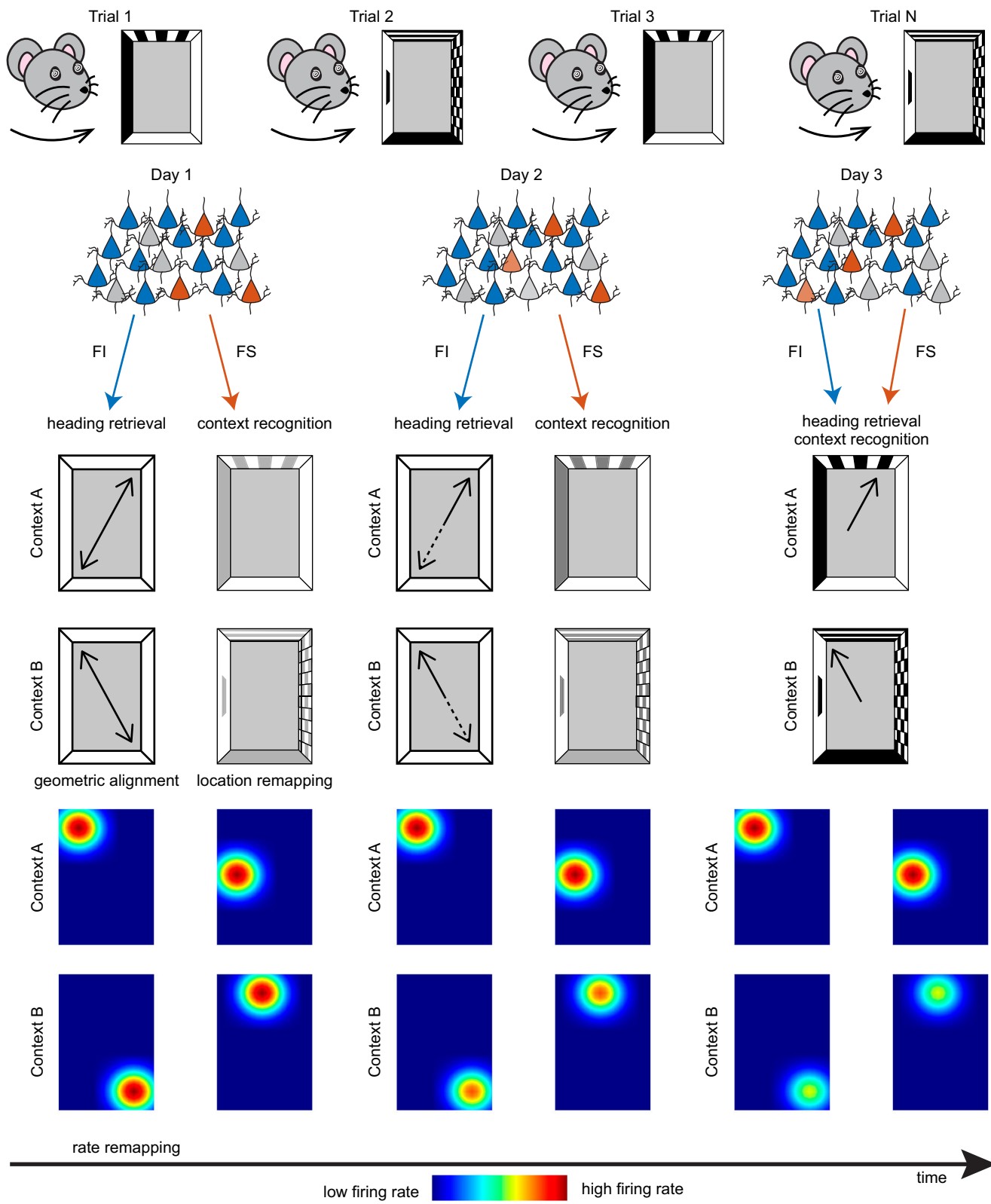

instance, Convict fish reared in circular chambers exhibited a greater reliance on features compared to their counterparts reared in rectangular environments[55]. Likewise, mice reared in a circular environment rich in features showed differences in reorientation compared to mice reared in geometrically enriched surroundings. Mice reared in circular environments displayed fast acquisition of featural information but were unable to encode geometry in a subsequent test, whereas mice reared in rectangular chambers showed the opposite pattern[56]. Thus,

although organisms may show a predisposition to rely on geometry to recover heading due to the inherent stability of geometric cues (e.g., the color of foliage may change with seasons, but the shape of the surrounding woods remains constant[18];), these representations are modifiable through experience. Consistent with these findings, our data provide support for geometry-based reorientation, which is associated with the formation of a cognitive map of the layout through incidental learning. However, this representation is not modular since

**Fig. 8 | Illustration of the proposed model for CA1 coding of heading retrieval and context recognition.** Disoriented mice trained in two different contexts differentiated by distinct, non-geometric visual cues and reward locations must regain their facing direction (heading retrieval) and recognize their environment (context recognition) to reorient. Two subpopulations of CA1 neurons with unique remapping and network stability properties play different roles in these processes. Initially, mice rely on the geometry of the surroundings to reorient, which correlates with FI cells (feature insensitive, depicted in blue) consistently aligned to environmental geometry over multiple days. This alignment serves to predict whether mice dig in the correct or a geometrically equivalent corner, indicating FI cells' critical role in heading retrieval. In contrast, FS cells (feature sensitive, depicted in red) display context-sensitive remapping, responding to non-geometric visual features, which likely facilitates context recognition. Over time, both cell types display increased rate remapping across context, which parallels increased responding in the correct cup location. This suggests that rate changes may serve to integrate featural information in both FI and FS cells, as animals learn the directional value of these cues. In the schematic, the ensemble of FI cells maintains consistency over days (top panel, blue cells), mirroring the stable geometric properties of the layout. Conversely, the ensemble of FS cells shows dynamic properties (top panel, red cells), paralleling the shift in reorientation strategies that incorporate features. The black arrows in the chambers indicate the reorientation strategies used over time (middle panel), and the intensity of place fields reflects the degree of rate remapping across days (bottom panel). These findings support geometry-based reorientation and the stable presence of neural representations of geometry, but they also demonstrate that these representations are not impermeant to other cues, as animals can incorporate new information through learning.

it can integrate featural information though changes in firing rate over time.

Another example of incidental learning involves the initial discrimination of contexts, which relies on recognizing the unique features defining them. This process is likely facilitated by the remapping of FS cells across contexts, which provides a neural mechanism for context recognition. However, unlike the consistent activity pattern of FI cells, the population similarity across contexts of FS cells becomes increasingly dissimilar with experience. Moreover, FS ensembles exhibit dynamic activity patterns, incorporating new cells over time, a characteristic observed in other associative learning tasks during different memory consolidation stages[50,57]. These observations indicate that FS cells are modulated by two forms of learning: incidental learning (i.e., initial context discrimination) and reinforcement learning (i.e., formation of feature-reward associations, which aid in reducing geometric errors). A study utilizing the cFos immediate early gene, a marker for learning, found that only a small fraction of CA1 cells were cFos positive. Similarly to our findings, these the cFos positive cells showed instability within exposures to familiar contexts, whereas cFos-negative neurons remained stable in familiar contexts and remapped in novel ones. These results suggested that distinct subpopulations of CA1 neurons with unique remapping properties encode either trial-to-trial experiential information or static aspects of the context, such as the shape of an enclosure. Furthermore, changes in rate served to integrate information about context into cFos positive cells[58].

The parallels between the Tanaka et al. study[58] and ours suggest that distinct subpopulations of CA1 neurons with inherent dynamic activity patterns are a critical feature of hippocampal processing. Interestingly, these similarities emerge even though in our study we separated cell types from a continuous distribution of similarity scores rather than using a selective molecular marker. Here, we used modeling to objectively select a separation threshold, a method previously used on continuous datasets[41,43]. Furthermore, some of the model parameters (e.g., inflexion point) that we used to categorize FI and FS cells have been previously used to discriminate cell types associated with task-related behavior[44]. However, it is worth noting that a potential caveat of this methodology is that categorization confidence decreases near the threshold. Despite this limitation, we show that the distinct properties of FI and FS cells do not reflect changes in registration quality or spatial information content.

We find that FI and FS also differ in the coherence of the active ensembles. FI cells, which exhibit context-independent place field rotations, maintain constant levels of average coherence (~40%) that are invariant to experience. In contrast, FS cells, displaying context-sensitive rotations, demonstrate a progressive increase in average coherence as time unfolds (~40% to 60%), suggesting learning modulation. Our findings also unveil a secondary coherence in both FI and FS cells within and across context. Notably, the primary and secondary coherencies align along the same geometric axis (0° and 180°) in FI cells, corroborating their role in representing geometry. Conversely, there is no discernible relationship between the primary and secondary coherencies in FS cells, with the primary coherence emerging as the only dominant rotation over time. A previous investigation in oriented mice using polygonal environments showed a similar average proportion of neurons exhibiting spatially coherent rotations between trials, as those observed in FI cells[59]. However, this study assessed coherency during free foraging, leaving uncertainties about whether coherence would increase in goal-oriented tasks as observed in FS cells from disoriented mice. It is possible that the coherency patterns observed in FI and FS cells emerge from distinct learning mechanisms, supporting incidental and reinforcement learning, respectively. These processes, in turn, may differentially influence heading retrieval and context recognition.

Incidental learning of geometry, which is defined by the position of boundaries, and reinforcement learning of features, such as landmarks, may arise from parallel learning systems[60]. The retrosplenial cortex and parahippocampal place area are vital for representing various aspects of scenes and landmarks[61–65]. Boundary information is encoded in the retrosplenial cortex, medial entorhinal cortex, postrhinal cortex, and parasubiculum[64,66–69], making these areas well-suited for conveying information about geometry. On the other hand, information about features likely originates in other regions. The postsubiculum plays a role in landmark control of head direction cells, suggesting its involvement in providing the directional value of features[70], and human fMRI studies showed that the right dorsal striatum is responsible for coding landmark locations[60]. Together, these data suggest that geometric and featural information may reach the hippocampus through distinct pathways to contribute to heading retrieval and context recognition, respectively. However, our study suggests that this information is gradually integrated in CA1 through changes in firing rate for more efficient reorientation.

Collectively, our findings provide new insight into a longstanding debate about how spatial reorientation is implemented in the brain. According to the geometric modular theory, reorientation is guided by a modular representation of the shape of the layout that is impermeant to featural cues. However, the use of features by disoriented navigators in certain conditions has been well-documented[4,71–75]. These observations led to an ongoing debate about how features are incorporated into reorientation strategies[1,74,76]. Our neural data provide strong support for geometry-based reorientation through the stable geometric alignment of FI cells, along with a second process that gradually incorporates feature/reward associations. This second process is manifested in the learning-modulated location remapping of FS cells, which is characterized by increased coherent alignment to opposite reward axes with training. These findings provide support for the two-process theory of reorientation set forth by Wang and Spelke[76]. However, we also show that rate changes in the entire active ensemble serve to integrate information about geometry and features over time, demonstrating that representations of geometry are not impermeant.

In summary, our results demonstrate that distinct CA1 subpopulations with unique remapping characteristics and temporal dynamics

play distinct functions during reorientation, differentially contributing to heading retrieval and context recognition. Understanding how neural representations adapt through learning during reorientation may serve to uncover the cognitive architecture that is critical for other tasks requiring complex cognitive processes.

## Methods

### Subjects
Male mice (C57BL/6 J, Jackson Laboratory, Bar Harbor, ME) aged 12–20 weeks old were placed on a 12 h light/dark cycle at 22 °C and 50% humidity. All experiments were carried out during the light portion of the light/dark cycle. Animals were food deprived until reaching 85% of their *ad libitum* body weight and food restriction continued until the last day of testing. Animal living conditions were consistent with the standard required by the Association of Assessment and Accreditation of Laboratory Animal care. All experiments were approved by the Institution of Animal Care and Use Committee of the University of Texas at San Antonio and the University of Iowa.

### Two-context reorientation task
Training began with a shaping phase during which food restricted mice were exposed to the reward buried in odor-masked bedding in a medicine cup placed inside their home cages at least 4 days prior to the commencement of the experiments. Mice received 12 trials per day that alternated between the two contexts. On the first day, the reward was placed on top of the bedding in the rewarded corner for the first 4 trials (2 trials in each context) and was then buried at sequentially deeper positions for the next 4 trials to train the animals to discriminate the contexts and associate each context with a distinct reward location (2 in each context). Animals were tested on the last 4 trials of day 1 and all trials the following days (12 trials per day). During testing trials, the reward was buried 1.5 cm from the surface of the cup, and animals received equal number of trials in each context. Before each trial, the animal was placed in a cylinder with a removable base that was placed on a turntable platform. To disorient the animal, the table was rotated 4 clockwise and 4 counterclockwise revolutions. The animal was then transported to the reorientation context, and the base was removed from underneath the animal. This procedure enabled animals to enter the chamber at random facing directions and marked the commencement of the trial. In all trials, animals remained in the chamber until they found the reward or after 3 min of sampling. The location of the first dig was used as the dependent measure for reorientation. First digs occurred in one of the four corners of each chamber: The correct corner (C) containing the reward, near corner (N) adjacent to the same short wall as the rewarded corner, geometric corner (G) diagonally opposite corner sharing the same geometric properties as the rewarded corner, and far corner (F) adjacent to the same long wall as the rewarded corner (Fig. 1a). Digs were counted when an animal displaced bedding using both front paws. Digs were scored by two experimenters blind to the experimental condition, and in the event of a dispute, a third experimenter confirmed the dig location. To avoid the use of distal room cues, the experimental chambers were rotated 90° after each trial and white noise was delivered from a centrally overhead speaker. To prevent the use of odor trails, the chambers were thoroughly cleaned with ethanol and all the cups were refilled with clean scented bedding during the inter-trial interval (3–5 min).

### Surgeries
Mice were anesthetized with isoflurane (3%; 1 L/min O$_2$ flow) and maintained at 1.5% isoflurane for the duration of surgery. Mice were placed in a stereotaxic frame (David Kopf Instruments) in a flat skull orientation. Body temperature was regulated using a heating pad and eyes were kept moist using a lubricant (Puralube Vet Ointment). Mice

were administered Rimadyl (5 mg/kg) pre-operatively and for two days following surgery.

**Electrophysiology surgery.** Mice were implanted with 6-tetrode microdrives above the right dorsal hippocampus (from bregma, AP: -1.6 – 2.1, ML: 1.6, DV: -1.0, from pia). A ground wire was connected to a screw placed in the contralateral occipital plate. The microdrives were affixed to the skulls with cyanoacrylate and dental cement.

Calcium imaging surgery: Mice were transfected with 400 nl of the calcium indicator GCamp6f (AAV1-hSyn-GCamp6f-WPRSE.SV40, Addgene; titer: ≥1 × 10$^{13}$ vg/mL) in dorsal hippocampal subregion CA1 (from bregma, AP: -2.1, ML: 1.8, DV: -1.65) at a rate of 50 nl / min using a Hamilton Syringe (CAT: 86257). A 2 mm diameter craniotomy was performed using the following coordinates: AP: -1.9, ML: 1.7. Superficial cortical tissue above CA1 was gently aspirated via sterile cold saline irrigation until the medial-lateral striations of the corpus callosum were removed and the diagonal striations of the fimbria axons were observed. Then, a 1.8 mm diameter gradient refractive index (GRIN) lens (Edmund Optics, 64–519) was implanted over CA1. The lens was lowered to a depth of -1.20 mm from the surface of the skull and depressed an additional 50 µm to compensate for any brain swelling during surgery. A stainless-steel screw was placed over the contralateral occipital plate and cyanoacrylate and dental cement were used to stabilize the GRIN lens. A surgical silicone adhesive (Kwik-Sil, World Precision Instruments) was applied to the exposed GRIN lens for protection as animals recovered from surgery. At least 4 weeks post-injection, animals underwent a base-plating procedure in which an aluminum baseplate was affixed to the skull. During base-plating, a miniaturized fluorescent endoscope (v3; miniscope.org) was attached to image calcium events in the anesthetized animal to find the optimal field of view containing visible cell structures. Baseplates were attached to the skull to maintain the field of view using dental cement. After the baseplate was secured, a black Delrin cap was magnetically attached to the baseplate to protect the GRIN lens while the mice were not being recorded. Optimal focal points were determined the following day once the dental cement was fully cured. In total, 14 animals were injected with GCamp6f, 5 animals were incorporated in the study and 9 mice were excluded for various reasons, including poor signal (N = 4), presence of traveling waves characterized by highly synchronous patterns of activity (N = 1), or disrupted timeline of recordings due to the Covid shutdowns (N = 4).

**Electrophysiological recordings.** Two weeks after surgery, animals were placed in an environment distinct from task-related arenas, and an experimenter searched for hippocampal CA1 pyramidal cells. The microdrive was connected to a tethered unity gain amplifier with green and red LEDs for tracking the position of the animal (tracking was also extracted using DeepLabCut, see below). The units were amplified between 2500 and 10,000 and filtered between 400 and 9000 Hz. The amplifier output was digitized at 32 kHz. The position of the animal and its electrophysiological data were recorded by Cheetah Data Acquisition software 6.4.2 (Neuralynx, Bozeman, MT). The electrode bundle was lowered by 5–15 µm per day until pyramidal cells were identified by their characteristic firing patterns[77]. Clusters were isolated using MClust software 4.4.07 (developed by A. David Redish, University of Minnesota). This process was conducted by concatenating all trials within a training day to minimize per-trial bias and track cells across trials with accuracy. Cells were included if they formed isolated gaussian ellipses with minimal overlap with surrounding cells and noise and waveforms were stable throughout the session. All cells were also inspected to rule out the presence of spiking events during the 2 ms refractory period. No attempt was made to track the same cells across days. After all data were collected, electrode placement was verified by deeply anesthetizing the animal and passing a current (0.1 mA for 5 seconds) through the tetrodes (52500 Lesion Making Device, Ugo

Basile) and perfusing the animal with 0.1 M PBS followed by 4% paraformaldehyde (PFA) made in 0.1 M PBS. The brains are placed in 4% PFA containing 3% ferrocyanide for 24 hr. and then incubated overnight in 30% sucrose-azide solution. The tissue was cryosectioned (50 μm, coronal) and collected on slides.

**Calcium-Imaging recordings and analysis.** Behavioral training began once optimal focal points were determined. The miniendoscope was attached prior to the commencement of each session and unattached at the end of recording before animals were returned to the home cage. During the experiments, the tether connecting the miniendoscope was attached to a commutator that allowed animals to move freely during reorientation. After behavioral experiments were concluded, animals were perfused in the same manner as the animals used for electrophysiology, and histology was confirmed by the presence of green fluorescent protein in CA1. Behavioral data were extracted using DeepLabCut[78,79]. Calcium-images were concatenated across trials for each session and motion corrected[80]. Individual cells were identified, denoised, and deconvoluted using a constrained nonnegative matrix factorization algorithm OASIS algorithm developed by ref. [81], as previously reported[82-84]. Inferred likelihood of spiking events (ILSE) were obtained from these deconvolutions and used as the measure of cellular activity.

## Calcium imaging cell registration

Cells identified on each day of recording were registered across days to allow longitudinal analysis of activity patterns across days. Cells were registered using a probabilistic approach (CellReg; MATLAB)[83]. Additionally, shape metrics of spatial footprints of registered cells (circularity, size, number of contiguous pixels) were computed and used to train a naïve Bayesian classifier against a subset of visually confirmed spatial footprints that were "cell-like". Only cells that were classified as cell-like were included in subsequent analysis (out of 3771 cells initially identified, 1469 were excluded based on shape metrics, resulting in 2302 total identified cells). Additionally, footprint quality, which is obtained from the registration output based on correlations of the spatial footprints for each registered cell, was evaluated as described by Sheintuch et al. (2017)[83].

## Place field analysis

First, animal position data was aligned so that the short wall adjacent to the rewarded corner is in the same location in each context across trials. An experimenter marked the boundaries of the chamber from position data and known trial chamber orientation, then applied a coordinate transformation (homography), converting the position data from video coordinates to physical coordinates in cm. Place fields were calculated from cell activity measured by spike times (electrophysiology) or by ILSE (calcium-imaging). The chamber was binned into 1 cm x 1 cm pixels and the number of spikes or summed ILSE that occurred in each pixel were counted to measure the activity map. The time spent in each pixel was calculated to measure the time map. The activity map is divided by the time map and then smoothed with an isometric Gaussian kernel ($\sigma = 3$ cm), resulting in the final place map. Pixels sampled for at least 0.05 s after smoothing were considered sampled. Only data from periods of movement that exceeded 2 cm/s were included for analysis.

## Best match rotation analysis

In all experiments, a best-match rotation analysis was used to quantify the orientation and coherence of the hippocampal map. First, rectangular place maps were compressed to squares by using anisotropic binning when calculating place maps (x position was binned into 1 cm bins and y position into 1.5 cm bins, resulting in 20 x 20 bins). Next, for each cell and each pair of trials, the rotation of the trial A place map (0°, 90°, 180°, or 270°) that maximized the pixel-to-pixel correlation to

the trial B place map was computed. The percent of pairwise trial comparisons for which each rotation yielded the best match was then calculated for each cell. These percentages were then averaged for each animal.

## Center-out method

Our best match rotation analysis was validated using a different quantification method that did not include compression of the environments into squares. First, place fields in each trial were identified as contiguous pixels that exceeded the 95th percentile of the map and the angle from the center of the map to the weighted centroid of the place field was calculated (center-out angle). For neurons with multiple place fields, the field with the largest area was used. Once the center-out angle for all trials was calculated, the difference in center-out angles was computed for each pairwise comparison for each cell (66 comparisons for 12 trials). The distribution of center-out angle differences was plotted as polar histograms for each day of recording. Distribution of center-out angles obtained from electrophysiological and calcium-imaging datasets were plotted separately. For center-out statistical analyses, an angle doubling procedure was applied to account for the bi-modality of most distributions[85], wherein each angle was doubled modulo 360. This step created unimodal distributions that could be tested to determine if they violated circularity. Circularity was determined by the function circ_r from the MATLAB toolbox CircStat[86].

## Digging behavior prediction using place field alignment

To predict digging behavior from place field alignment, the cells' center-out angle in each trial was computed (see Center Out Method above). To predict whether the animal made C or G digging response (N and F responses for error predictions), we trained a SVM classifier using the sine and cosine of each cells' center-out angle during C and G trials as predictors, validated using a leave-one-out approach (N and F for error predictions). To predict digging behavior across days, we trained a SVM using the center out angle from C or G trials on a training day and predicted C or G responses using the center-out angle of the withheld trial on the testing day. For these predictions, only cells active on the training session and the testing session were used, which was determined by cell registration.

## Context similarity analysis

Place maps from disoriented animals align to geometry, displaying 180° rotations between trials. To assess the similarity of place fields across contexts, maps were first aligned to the same orientation in each context to account for 180° rotations due to geometry by selecting the best sequence of rotations that maximized similarity for each pairwise comparison. This was done by comparing correlations without rotation of the second map (0°) or rotating it 180°. Then, an average aligned map was calculated for each context. The average context maps were, in turn, aligned relative to each other by selecting the maximal correlation between 0° or 180°. Correlations between the average aligned maps defined context similarity and were used to classify FS and FI cells.

## Correlation threshold selection

To select the best correlation threshold to characterize similarity scores, we first separated the cells into deciles using the distribution of context similarity (Fig. 3c). An overall Pearson's correlation coefficient between the average within and average across correlations for the associated cells was computed, which yielded a set of 10 values (one per decile, hereafter called Corr-Decile). For the statistical analysis of Corr-Decile function, we chose a non-linear asymptotic regression model[87]. Linear models (e.g., polynomic) were eliminated as they provided a low degree of fit or failed to improve the fit (Supplementary Fig. 6).

The model selection was based on a goodness-of-fit analysis among the non-linear models and was performed in two stages. First, we identified significant points (Roots, Extremas and Inflections) in the Corr-Decile function using the Taylor Regression algorithm[88]. Second, we selected the best model from an exhaustive pool of alternative models, according to the lowest values for Akaike's information criterion (AIC), residual standard error (RSE), and the highest value for log-likelihood (L-L)−a set of metrics gauging goodness of fit[89]. In total, 85 model types were tested, which were grouped according to three widely accepted classification of[90]: Convex/Concave or Exponential family Models (E-shaped; e.g. Asymptotic regression, Exponential, Power), Sigmoidal-Family models (S-shaped; e.g. Logistic), and Curves with Maxima/Minima (M-shaped; e.g. hormesis models, humped type models, etc.). E-Shaped type models were in general more suitable to fit the data. Among these models, the Asymptotic Regression model exhibited the best results in terms of goodness of fit (Supplementary Table 4)[89]. Once the model was selected, it was determined that the selection of the remapping threshold based on the fact that Decile 1 coincided with the function's Half-life, root, and maximal relative growth rate of the asymptotic regression model. Detailed description of the Asymptotic Regression model and the selection of the remapping threshold is in Supplemental Figs. 5–6 and Supplementary Table 4.

## Population similarity across contexts

To calculate similarity of population representations of Context A and Context B, a population vector was computed by stacking each cell's average aligned map in each context. The normalized dot product between each corresponding pixel across average aligned maps provided a measure of similarity in each population (FI vs. FS cells). High dot product indicated high similarity between average aligned maps in the population. Cells must have at least one place field in each context to be included for analysis (13 cells were excluded using this criterion). Place field maps were calculated during periods of movement (speed > 2 cm/s) (see Place Field Analysis above).

## Context prediction using firing location of FI and FS cells

Context was predicted by comparing the maps of a withheld trial with the average aligned maps of Context A and Context B. The average aligned maps were calculated by averaging the firing location of each cell in each context, excluding the withheld trial, for each cell group (FI vs. FS). Next, the average normalized dot product was computed between all the cells' maps in the withheld trial and the average aligned maps. The average aligned maps with the highest average dot product with the withheld trial maps predicted context for that trial. Correctly predicted trials were averaged for each session.

## Coherency analysis using Best Match Rotation

Coherency of the cell population was determined by measuring the orientation coherence in simultaneously recorded cells. This was done by determining the proportion of cells that yielded the same best match rotation [see Best Match Rotation (BMR) analysis section above] across trial pairs for all pairwise trial pairs. Cells that were inactive in at least one trial of a trial pair were excluded from coherence analysis. Coherency was ranked according to the proportion of cells that yielded the same BMR ranked from highest (1st BMR) to lowest (4th BMR). Chance was defined as equal distribution of each rotation (25%).

## Context prediction using firing rate

Trials in which animals made C or G responses were separated into two classes: Context A and Context B. Then, the mean firing rates (MFR) were used to train a support vector machine (SVM) for binary classification. A linear kernel was used with an empirical prior to train the SVM, which was cross-validated using a leave-one-out procedure (e.g., having one trial withheld at a time). Correctly predicted trials were

averaged for each session. Cells must have place fields in at least 4 trials, two in each context, to be included for analysis.

## Heading prediction using firing rate

Heading prediction was accomplished using the same procedure as above, except that the trials were separated into two classes based on digging location. C dig trials were grouped in one class and G dig trials were grouped in the other. Using a leave-one-out cross validation procedure, the support vector machine predicted whether the animals' digging side corresponded to the rewarded or geometrically equivalent locations. Correctly predicted trials were averaged for each session.

## Statistical analysis

**Behavioral analyses.** All data were checked for normality using the Shapiro-Wilk test and included Greenhouse-Geisser (G-G) corrections for sphericity violations. We used two-way repeated measures ANOVA for behavioral analysis of dig locations. The G-G approximation allowed us to correct the original degrees of freedom (df) by epsilon, the degree to which the sphericity was violated. For example, in the ANOVA of dig locations, we would multiply 3 and 39 (the original df) by the estimated value of epsilon at 0.7271, generating the adjusted df 2.18 and 28.36, respectively for the numerator and denominator of the corrected F value. Behavioral performance was further evaluated using Bayes Factors (BF)[91] to determine whether animals were digging at random (model $M_{null}$), or preferentially based on learning (model $M_{alt}$). Both models are based on Bernoulli distributions, where the likelihood function for dig outcomes for each day and mouse is defined as:

$$p(z, N | \theta) = \theta^z (1 - \theta)^{N-z} \quad (1)$$

$N$ represents the total number of trials, $z$ is the number of trials with correct outcomes (digs in C), and $\theta$ is the probability of a successful outcome. Digging patterns were evaluated considering 2 digging patterns: (A) Proportion of digs in the geometrically correct axis in each context digs (C/G Model); (B) Proportion of digs in the correct (C) corner (C Model). We define the models as follows:

## C/G Model

$M_{null}$: This model represents that the animal has not learned the task and the outcomes are the result of 50-50 chance (expressed as $\theta = 0.5$):

$$p(M_{null}) = 0.5^z (1 - 0.5)^{N-z} \quad (2)$$

$M_{alt}$: This model represents that the animal has learned the task, indicated as a probability between 0.5 and 0.9. In order to achieve a uniform distribution across the range of values, then an integration over probability $\theta$ is calculated as follows:

$$p(M_{alt}) = \frac{1}{0.9 - 0.5} \int_{0.5}^{0.9} \theta^z (1 - \theta)^{N-z} d\theta \quad (3)$$

The Bayes Factor for each animal (represented by '$i$') is calculated as follows:

$$BF_i = \frac{p(M_{alt})}{p(M_{null})} = \frac{\frac{1}{0.9 - 0.5} \int_{0.5}^{0.9} \theta^z (1 - \theta)^{N-z} d\theta}{0.5^z (1 - 0.5)^{N-z}} \quad (4)$$

## C Model

$M_{null}$: This model represents that the animal has not learned the task and the outcomes are the result of chance (25% of the time a mouse

would go to each of the 4 corners, expressed as $\theta = 0.25$):

$$p(M_{null}) = 0.25^z(1 - 0.25)^{N-z} \qquad (5)$$

$M_{alt}$: This model represents that the animal has learned the task, indicated as a probability between 0.25 and 0.9 (above chance):

$$p(M_{alt}) = \frac{1}{0.9 - 0.25} \int_{0.25}^{0.9} \theta^z(1 - \theta)^{N-z}d\theta \qquad (6)$$

We assume, a priori, that both models are equally probable, $p(M_{alt}) = p(M_{null})$, and so the BF for a given animal (represented by '$i$') is calculated as:

$$BF_i = \frac{p(M_{alt})}{p(M_{null})} = \frac{\frac{1}{0.9-0.25} \int_{0.25}^{0.9} \theta^z(1 - \theta)^{N-z}d\theta}{0.25^z(1 - 0.25)^{N-z}} \qquad (7)$$

For a group of independent subjects, in both model C and model C/G, we calculate the group's BF as:

$$BF_{group} = BF_1 \cdot BF_2 \cdots BF_N = \prod_i^N BF_i \qquad (8)$$

A conventional hypothesis decision threshold states that when $\log(BF) > \log(3) \approx 1.1$, there is substantial evidence for $M_{alt}$ (learning), and when $\log(BF) < \log_e(1/3) \approx -1.1$ there is substantial evidence in favor of $M_{null}$ (non-learning), where log is the natural logarithm[91]. We plot the $\log(BF)$s as cumulative distributions and denote the critical values as vertical dotted lines[40].

## Cell activity analyses

Dot product population vectors were analyzed with a two-way Ranked ANOVA variant described in previous work[92,93]. Ranked ANOVAs have power advantages in comparison to other tests[93], especially when experimental conditions have distributions that are sensitive to differences in the average ranks. All analyses involving FI and FS cells included Type III sums of squares corrections for unbalanced factorial ANOVAs, given the uneven sample size of the subpopulations. Rom's post-hoc tests were used for all regular, and robust ANOVAS, and multiple comparisons with Bonferroni corrections for ranked tests[94]. Robust $t$-tests were used for support vector machine classifications. Rate remapping was analyzed using a two-way repeated measures ANOVA. When the type of data exhibited irregular variability or deviation from normality, we conducted robust ANOVAs (Omnibus Fw statistic and Tw $t$-test trimmed with Rom's pos hoc corrections, see ref. [45], for details). This was the case for firing rate changes and spatial information content. The analysis of the effect sizes for firing rate changes was based on robust descriptive measures xi ($\xi$), and the statistical comparison of these effect measures was undertaken by means of the projection distance between the estimate of location and the estimate of the grand mean (see[92], especially the R function rmES.pro). In the case of one-sample tests with respect to the value of chance (e.g., Figs. 2e, 4d, 6c, 7e, 7f), we opted for a robust two-tailed variant, in which probability estimation is based on bootstrap extractions from the xi ($\xi$) effect size statistic, in order to maximize statistical power for small sample sizes (see[92], R function D.akp.effect.ci). Additionally, the analysis of the joint probabilities shown in Fig. 5c was approached with a Wilcoxon, one-tailed, signed-rank test. Therefore, two-tailed tests were used throughout the study, unless a one-tailed test is specified in the text. Complete descriptions of statistical parameters, including additional measures of central tendency, can be found in the statistical source code (https://doi.org/10.5281/zenodo.11454686). All statistical analyses were performed in the R programming environment version 4.3.0[95]. Statistical decisions were made using a significance probability set at .05.

## Reporting summary

Further information on research design is available in the Nature Portfolio Reporting Summary linked to this article.

## Data availability

Source data are provided with this paper. The data generated in this study have been deposited in the Mendeley (https://data.mendeley.com/datasets/gz2vbmsxnn/1) and GitHub (https://github.com/ManuMi68/MuLaNa2/tree/main/NeuroSpatialData) databases. Original recordings are available upon request due to the large size of the data files. Source data are provided with this paper.

## Code availability

Custom code used for analysis is available in the following repositories: Statistical analysis (R2023). https://htmlpreview.github.io/?https://github.com/ManuMi68/MuLaNa2/blob/main/NeuroSpatial.html. RamosÁlvarez, M.-M. (2024). ManuMi68/MuLaNa2: NeuroSpatial (v1.0.1). Zenodo. https://doi.org/10.5281/zenodo.11454686. Behavioral and neural code analysis (MATLAB 2019, 2022). Marc Normandin. (2024). marcnormandin/chengs_task_2c_ca1_MuLaNA: Final Version (Version v1). https://doi.org/10.5281/zenodo.11390953. celiagagliardi. (2024). celiagagliardi/two_context: two_context_final (Version v1). https://doi.org/10.5281/zenodo.11391365. Marc Normandin. (2024). marcnormandin/chengs_task_2c_ca1: Final Version (Version v1). https://doi.org/10.5281/zenodo.11390947. Additional programs and toolboxes used for acquisition or analysis. Neuralynx, Cheetah 6.4.2 acquisition system−https://neuralynx.fh-co.com/research-software/cheetah/. CNMF_E−https://github.com/zhoupc/CNMF_E. MClust 4.4.07 −https://redishlab.umn.edu/mclust. DeepLabCut−https://github.com/deeplabcut. Neuralynx MATLAB Netcom Utilities−https://neuralynx.com/software/category/matlab-netcom-utilities. boxplotGroupv2−https://www.mathworks.com/matlabcentral/fileexchange/74437-boxplotgroup. Distinguishable Colors−https://www.mathworks.com/matlabcentral/fileexchange/29702-generate-maximally-perceptually-distinct-colors. CircStat−https://www.mathworks.com/matlabcentral/fileexchange/10676-circular-statistics-toolbox-directional-statistics. CellReg−https://github.com/zivlab/CellReg. MiniscopeAnalysis_CNMF_E−https://github.com/marcnormandin/MiniscopeAnalysis_CNMF_E. NoRMCorre−https://github.com/flatironinstitute/NoRMCorre. homography_solve & homography_transform functions−https://www.mathworks.com/matlabcentral/answers/26141-homography-matrix.

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

## Acknowledgements

This works has been funded by NSF (NSF/IOS 1924732 to I.A.M.), NIH (R01 MH123260-01 to I.A.M.; F31 EY031582 to C.M.G.; RISE GMO60655 to M.R.L.). We extend our gratitude to Xiaojing Chen and the Knierim lab for their invaluable guidance in training us to implant lenses over CA1 for our calcium-imaging experiments.

## Author contributions

C.M.G. collected data, conducted behavioral and in vivo recording experiments, wrote code for analysis, and wrote the manuscript. M.E.N. wrote code for analysis and contributed to writing the manuscript, A.T.K. conducted experiments and conducted analysis. M.M.R.A. conducted statistical analyses, M.R.L. and J.B.J. contributed to data collection, R.A.E. contributed to experimental design and writing the manuscript, I.A.M. supervised all experiments, analysis, and writing of the manuscript.

## Competing interests

The authors declare no competing interests.
