## [Peer Review File · Nature Communications]

Distinct neural mechanisms for heading retrieval and context recognition in the hippocampus during spatial reorientationREVIEWER COMMENTS

Reviewer #1 (Remarks to the Author):

This study assesses the behavioral and place cell responses of disoriented mice in a task where the mouse has to find a buried reward in one corner of a rectangular environment. The authors show that the mice search for the buried reward in the correct corner and its rotational equivalent. This result implies that the disoriented mice use the geometry of the environment to reorient. The mice also successfully distinguish between two rectangular environments with different contextual cues. This result implies that the mice can use context to discriminate between environments. Interestingly, the mice seem to shift from a geometric solution to the use of the polarising landmarks within each rectangle with training.

For the place cell recordings, place fields were observed to either stay in the same place or rotated by 180 degrees within a given rectangle. Across the two rectangles, place fields tended to either be found in the same location (feature insensitive cells) or to remap (feature sensitive cells). The latter showed greater differences across contexts with training, while the former tended to remain in the same location with additional experience. Calcium imaging of these cells showed, for consistently active cells, more stability on day three for the feature insensitive cells (compared with day 1) and more remapping across contexts for the feature sensitive cells. Place cells were not influenced by reward location but did show firing rate differences across contexts.

The behavioral results are interesting and are similar to those published previously by the same group (Julian et al., 2015). The advance in the current paper may be the finding of a shift in strategy with continued training.

Likewise, compared to a previous study assessing place cells in this task (Keineth et al., 2017), the current findings provide additional detail on place cell responses and show that some cells tend to be invariant across contexts while others remap.

The current findings are consistent with previous studies and provide additional information on the behaviour of place cells across contexts and acquisition. Given the previously published studies on this task, the significance of the current work is difficult to anticipate. However, the methodology is sound, the analyses appropriate and thorough, and the conclusions are well supported.

Reviewer #2 (Remarks to the Author):

Summary

In this study Gagliardi et al. investigate the neural mechanisms within the context that support important features of spatial memory: context recognition and heading retrieval. A previous study from this lab demonstrated that spontaneous rotations of place fields would sometime spontaneously rotate following disorientation during a spatial memory task (Keinath et al, 2017). These rotations were not random, but were anchored to the geometry of the arena, and importantly, they occurred primarily when animals made errors in digging for a food reward (e.g., a 180 degree field rotation would predict that the animal dug for food reward in the corner diagonally opposite the correct food location). A behavioral study from the same lab (Julian et al., 2015) demonstrated that these behavioral errors and rotations occurred even when animals had to utilize non-geometric features to guide their digging location in two different contexts. This study concluded that contextual identification and heading retrieval are governed by distinct neural mechanism. In the current study, Gagliardi characterize these mechanisms using calcium imaging and electrophysiology to record real-time neural activity as mice perform a spatial memory reorientation task similar to that in Julian et al. (2015) which requires mice to learn that reward is hidden in one corner (upper-right) of one rectangular arena (Context A), while it is hidden in a different corner (upper-left) of an identically shaped arena (Context B) that has distinct non-geometric features from Context A.

The authors find that a large subset of cells in the hippocampus maintain a similar set of place fields between contexts, which they dub feature insensitive (FI) cells since their firing configuration is not influence by the non-geometric features of each environment. A small subset of cells does move, or remap, between contexts, and the authors call these feature sensitive (FS) cells. In line with their previous work, FI cells exhibit rotations that match the geometry of the arena while FS cells do not. Decoding analyses demonstrate that FI and FS cell activity predicts heading and context, respectively, supporting the authors hypothesis that FI and FS cells provide separate neural mechanisms to support these distinct cognitive demands. In one of the most interesting findings, FS cells also gain the ability to decode heading on day 3 of the task, which is imp. This suggests that integrating heading and context retrieval into the same group of neurons could underlie the increase in behavioral performance observed in Figure 1. This hints that the “conjunctive encoding” or “mixed selectivity” observed in hippocampal neurons is not innate but might be learning dependent, and this learning dependence has been overlooked previously since most such studies employ overtrained animals and do not record learning data.

Overall, this study provides an important advancement in our understanding of the neural mechanisms that underlie spatial memory and orientation. Gagliardi et al. demonstrate that these cognitive mechanisms are supported by separate sets of neurons in the hippocampus that

independently code for heading/context retrieval (and intriguingly might begin to mix on day 3). However, there are several key issues listed in the “major concerns” section below that need to be addressed in order to heighten confidence in the robustness of the results and guide larger interpretations: the lack of a direct, cell-to-cell assessment of feature insensitivity between arenas, the lack of individual animal/session data provided, and a lack of clarity in how aligned maps were calculated in the majority of the figures.

Major concerns

The key issue is that the analyses performed do not directly assess if cells are truly feature insensitive (FI) or feature sensitive (FS). This is for three reasons which are also identified in the sections below: 1) the definition of “feature” is not clear, 2) many cells identified as FI could actually be FS cells due to performing rotations for each cell individually rather than rotating all cells’ place fields together the same amount, and 3) FS cells have much lower stability even within a context, suggesting that they might simply be less reliable, have lower firing rates, or undergo representational drift. Additionally, the authors do not provide any rationale for the cutoff used to separate FS vs. FI cells, which need to be addressed.

Is the term “feature” used exclusively to mean “non-geometric features” (p. 10) such as lines on the wall or different odors between contexts? I suggest making this explicitly clear early on to avoid confusion. What is included as a “feature”? Is time between exposures considered a feature? Without a clear definition it is difficult to properly interpret most of the results.

Almost all the plots are presented using data pooled across all animals. To get a better sense of the inter-animal variability for all results, the authors need to provide data for individual animals wherever possible, in particular for the plots and analyses noted below.

It is not clear how map alignment was performed from page 10 onwards. This is crucial to understanding the definition of FI and FS cells throughout the paper and interpreting the results. Per Keinath et al. (2017), place fields tend to rotate together across the neuronal population and that the orientation of the neuronal population is important for predicting behavior. However, Figure 3A and the text imply that rotations for each trial were calculated for each cell independently, e.g., cell #1’s rotation sequence might be 10010 while cell #2 might have the rotation 01010 for the same session. Likewise, it is unclear if a similar method was used for between context aligned map correlations – were only the best correlations used? The semi-transparent rate map in Figure 3A seems to indicate that between-arena correlations were calculated at 0 and 180 degree rotations, and only the highest was used, but this information is not found anywhere in the text or methods. As such, the authors could be overestimating the proportion of FI cells and missing FS cells. Take the example below (in attached file with sketch) – the blue, red, and green place fields maintain the

same configuration but rotate 180 degrees between trials in A and B. After alignment, these place fields would have high correlations and would be FI cells. However, the beige-dashed and purple dashed fields would also have a high correlation after alignment and could be designated as FI cells, even though they behave very differently as they do not rotate with the rest of the cells during Trial #1 in Context B.

With this in mind, many FI cells could actually be sensitive to the features of different contexts. One way to address this is to assess pairwise rotations between cells. Do pairs of FI cells always (or almost always) rotate together the same amount between trials across contexts, whereas FS cells do not? Results from Figure 7 indirectly addresses this within the same context, however a direct comparison of rotations between cells on a trial-by-trial basis across contexts would increase confidence that FI cells are truly insensitive to non-geometric contextual features.

General comments

How/why was 0.3 selected as the context similarity cutoff? How robust are the rest of the findings, in particular the decoding analyses in Figure 7-8, to this threshold?

The authors are inconsistent in the exclusion of days and rotation amounts from their analyses. For example, Figures 2 and 6 consider rotations of 0, 90, 180, and 270 degrees while Figure 3/7 only considers 0 and 180. Why? Likewise, why is only Day 1 and Day considered for FS vs FI cell correlations analyses in Figure 3/4 while Day 2 is also included for rotational analyses in Figure 5?

The authors focus on 0 and 180 degree rotations in many analyses, yet many 90 and 270 degree rotations are observed at both the behavioral level (F and N trials) and the neural level (90 and 270 degree place field rotations in Figure 2). Are these phenomena related? Do 90 and 270 degree rotations of place fields occur mainly when mice perform F or N trials? If FI cells are truly important for heading retrieval, one would predict that 90 and 270 degree place field rotations of these cells would occur primarily during F and N trials.

The authors leverage one of the strengths of miniscope calcium imaging: the ability to record from large cell ensembles. However, they do not utilize the other strength of imaging – the ability to confidently track cells across days — and therefore miss the opportunity to address a key question about the permanence of FS vs FI cells. Are these cells drawn from separate pools that stay separate on different days? Or do FS cells and FI cells flip-flop on different days? Answering this question, for example by training the decoder from Figure 7 on FI cells from Day 1 and then

decoding heading from those same cells on Days 2 and Days 3, would help ascertain if FI cells are hard-wired to be FI cells can change their identity (phenotype?) with experience. This would provide valuable insights into the larger circuit mechanisms involved in shaping the responses of these cells. For example, if FI cells are truly hard-wired to be FI cells, do they receive stronger input from head-direction cells in thalamus or entorhinal cortex than other cells?

Specific comments

Figure 1A-B: Please provide individual data points for behavioral performance to aid in reproducibility by other researchers and provide an accurate picture of behavioral variability and how it changes from days 1-3.

Figure 1E: Are these behavioral plots for both arenas combined? Are behavioral strategies invariant between context A and context B? In other words, is $\log(\text{BF})_A$ shown in Figure S1 correlated with $\log(\text{BF})_B$?

Figure S1 and page 7: Are BF analyses significant on all days here or just 2 and 3 like full model above? State result in main text. See note below on Figure S1.

Figure 2C: Please provide individual data points. How is the post hoc Rom's test being applied? Were as each day-rotation compared to every other day-rotation? Or were days grouped together for each rotation before comparing across rotations? It is not possible to tell from the figure.

Pg 10: The two possibilities outlined at the top of the page are hard to follow. A graphical schematic outlining each would make this much clearer and leave less up to the interpretation of the reader. For example, does "geometric alignment" mean cells have the same configuration relative to each other in both contexts?

Figure 3A and pg. 10 text: Why are only 0 and 180 used for map alignment? I understand that these rotations are predominantly supposed to align to the arena geometry per Keinath et al., 2017, but Figure 2 shows that 90 and 270 rotations also occur fairly frequently.

Figure 3C: How consistent are these proportions across mice? Please provide data points for individual mice.

Figure 3E: It appears that FS cells are generally unstable and less reliable, as indicated by their lower within context correlations. This makes it questionable if they are truly “feature sensitive” or just bad place cells? Likewise, lower correlations could result from lower event rates for FS cells, especially for calcium imaging. To accurately characterize these cells and assure that their feature sensitivity is not an artifact of lower event rates/firing rates or poor spatial specificity, the authors should quantify event/firing rate and spatial information (or another metric of place field quality) between FI and FS cells.

Figure 4: How were cells that had few or no transients in one context handled?

Why is only Day 1 and Day considered for FS vs FI cell correlations analyses in Figure 3/4 while Day 2 is also included for rotational analyses in Figure 5? Exclusionary criteria should be clearly noted in the text/methods.

Figure 4B: are correlations for all cells active in both sessions? Or does this include inactive cells too?

Figure 5 and related text: How were cells with multiple place fields handled? Did you use the COM of all fields or just the field with the highest in-field event rate? Considering only the strongest field or taking the average angle of all fields for cells with multiple fields could further sharpen the distributions.

Pg 16, middle: The “possibility” referred to in this section is not clear and need to be clarified to interpret the rest of the section related to Figure 6: “This result could happen if both FS and FI cell align to geometry within context despite having different phenotypes across context. To investigate this possibility...” What is a “phenotype”? A place field location? Being an FI vs an FS cell? Or is it how that cells place field aligns/rotates on G vs C trials?

Pg. 18, bottom: “Conversely, FS cells, which respond to features (at least in identically shaped contexts defined by such features), predict context across days, and only predict heading once animals form feature/reward associations and these cells display stability within each chamber.” Are FS cells invariant in their contextual discrimination across days? e.g., are FS cells identified on day 1 also good at performing contextual discrimination (and not heading discrimination) on day 3? If they are, this would strengthen their role as truly feature selective.

Figure 7B, 7D: As noted above, the authors should provide individual data points overlaid on these bar graphs.

Figure 8A: This schematic is confusing. Why are only three points highlighted in Context A? Why only one point for the across context example? It might help to connect the same cell in context A and context B with lines. Also, rate remapping generally uses the PEAK in-field firing rate, not the mean rate (Leutgeb 2008?). What does this look like if you use peak firing rate? Last, what does this look like with calcium imaging? This is one of a few studies that directly compares results between imaging and electrophysiology, so providing Ca²⁺ results too would provide valuable information for other researchers even if the ultimate conclusion is negative (e.g., that the Ca²⁺ does not have the resolution to perform traditional rate remapping analyses).

Figure 8F: If I'm interpreting this correctly, combined with Keinath et al. (2017) this suggests that cells are both rotating AND rate-remapping on G trials. What is the authors interpretation of this result? Are FI cells truly insensitive to non-geometric shouldn't they retain the same firing rate regardless of if they were next to the striped wall vs the white wall in Context A, for example, since both walls are short?

Figure 8E-F: Provide individual data points. Decoding accuracy increases for context and heading retrieval after one day. Is this effect driven by FS/FI cells, respectively? In other words, do these plots look the same when broken down by FI/FS cells?

Figure S3B: Which figure is this related to?

Figure S4: How does this support Figure 6? How are "stable" and "remapping" cells related to FI/FS cells? The "stable" (presumably FI?) cells appear to have a 0/180 rotation preference, but the remapping (presumably FS?) cells do not, which does not agree with Figure 6.

Also, how does Figure S4 support Figure 6? There are no stats provided. Many of the "remapping" (FS?) cells appear to have no preference for 0/180.

Methods: Which CNMF implementation was used for calcium imaging analysis? Caiman? Minian? Something else?

In our hands not all injected mice obtain suitable imaging following viral infusion. How many mice were used in total for this study, including total infused with the GCaMP virus and the final number imaged (N = 5 in Figure 2...).

The authors indicate a minimum of 4 weeks between viral infusion and baseplate attachment. What was the mean and maximum time between viral infusion and base plate attachment? In our hands, using this high a titer of virus ($> 1 \times 10^{13}$ vg/mL) can frequently cause overexpression as early as 4-6 weeks post injection, resulting in aberrant calcium activity (traveling waves), especially over time. What steps, if any, were taken to ensure healthy expression of GCaMP?

Minor issues and typos

Pg. 4, typo "...however, lost navigator must ..."

Pg. 8 second paragraph: there are no STAR issues in Nature journals.

Pg 8. The N value for single units (7) listed in the text is inconsistent with that listed in the Figure 2B legend (6).

Bottom of page 16. Typo – extra period between "context" and "(Figure S4)"

Methods – "Subjects": missing period between 'cycle' and "Animals".

Methods – "Two context reorientation task": Animals were tested on the last 4 trials of day 1 and the 2 following days (12 trials per day)". This is unclear. Are they tested on all 12 trials on day 2 and 3? Or only on the last 4 trials as on day 1?

Reviewer #3 (Remarks to the Author):

The data gathered for this study are sufficient to address questions about how animals learn to make distinctions between physically symmetric locations in environments, presumably by learning to incorporate features that are unique to each within-environment location. Relative to prior work, the study makes notable advances in towards answering these questions by incorporating a second environment, studying the behavior and neural data over multiple days, and utilizing multiple recording methods to reduce bias introduced with a single method and leverage the advantage of calcium imaging in recording large numbers of neurons and registering them across days. The overall impression of these results is that they represent an important advance for the field, would be applicable to interpreting other studies, and are likely replicable (Figures 3 and 4 are especially strong). However, the precise impact of the findings presented is difficult to fully appreciate given the wide mix of analysis approaches, the large number of terms which are not strictly defined (though in some cases can be inferred), and many instances where writing is

unclear. My opinion is that no further experiments need to be performed, and new analyses should be adjustments of prior effort, but some amount of rewriting and reorganization is required to convey the novelty of these findings to their full potential.

Major concerns:

Undefined terms – a number of these are specific enough to this study that a field expert not familiar with the authors' prior work would have to infer meaning from context

- Modular/modularity – does this indicate a representation that is wholly replaceable, or one that is independent of others?

- Disoriented (page 5, first paragraph of results) – critical feature of the task, define it up front

- Environmental geometry/layout – assume something like “the relative position of features that constrain movement, such as walls, doorways, and edges;” alongside here would be an explanation of similarity in this environment

- Non-geometric features – sensory modes are good examples but not a strict definition

- Long wall right/left

- Absolute corner location

- Geometric equivalent

- Rotational equivalent

- Reorientation behavior

- Heading [behavior]

- Behavioral strategies (page 7 paragraph 3)

The order of some results seems motivated by the “class” of analysis rather than theoretical chain of thought. For example, Figure 1 establishes that animals are a bit confused by the symmetry of the environments but seem to understand that the environments are different, figure 2 shows that place cells exhibit a similar type of confusion, it would be natural here to ask how tight that relationship is on a per-trial basis, which isn't addressed until Figure 7. This ordering makes it challenging to follow the authors' intuition for the progression of results.

The stated motivations for figure 3 are challenging to understand with respect to the specific analyses that follow. Figure 3 is described as being motivated to find how neurons encode “featural information,” but, agnostic to this specific experiment, one would more broadly ask about the relationship of place fields across contexts, a long-studied phenomenon. There is a strong a priori constraint provided on this question by the finding in figure 2 that ~60% of neurons remap between trials according to the 2-fold symmetry of the environment; this seems only tacitly acknowledged by the language at the top of page 10. Additionally, it has not been established by this point in the text that there even is remapping between contexts under these experimental conditions, and it is

suggested that there may not be due to the equivalent shapes of the boxes. The hypotheses offered in the middle of the first paragraph are contingent on these and other assumptions and analyses that have not been fully developed at this point in the text. This links to the point above that the connection between hippocampal activity and behavior that it is presumed to support has not yet been established.

Requested analysis: While the 0.3 correlation score was previously established by another paper, it is not clear from the distribution shown in Figure 3B that this indicates a fundamental functional divergence, and the analyses that follow could (cynically) be interpreted as post-hoc justification for that criterion. Additionally, the findings presented in 3E are redundant with this definition (also, how can the curves for FS and FI across overlap when they are defined by the criterion on the x-axis?). I recommend that the authors further subdivide neurons by their context similarity scores (e.g. into quartiles or even deciles, since only 11% of cells are below 0.3), or make scatterplots showing the relationship of (e.g.) each cell's correlation within environments as a function of its correlation across environments. If the quartile+ method were employed for Figure 4B, this would help us understand whether there is a continuum of population similarities or bimodal (or more) clustering.

Rate remapping, context coverage in general, specifically on page 20: Authors need to detail both how much of the environment the animals explored on each trial (it seems from the methods that animals were pulled out after a correct dig) and how trial average firing rates were computed. Here's the issue: if a majority of place cells have similar rotation-aligned place fields across trials and contexts, and animal behavior is divergent across contexts due to the different locations of C and G wells and given high performance, it follows that the animals would sample different portions of a given cell's place field across contexts. (Example: if a cell has a place field in the upper left corner of both boxes, and in box A the animal runs from the middle to the upper right he will barely skirt that place field while in box B the mouse runs from the middle to the upper left that cell will go nuts). This would result in differences in firing rates not strictly attributable to differences from cells' incorporating feature information but instead from uneven sampling. The authors need to expand on how firing rates were compared across trials: whether these maps were restricted only to comparable bins visited in both contexts on a pair of trials, or if it was the average firing rate of a cell (while it was firing?) for that trial independent of animal's position, or some other method. This is relevant to most analyses, but especially to Figure 8.

Requested analysis: Figure 5C, legend: it is claimed that the increasing circularity of these distributions indicates that "each FS cell remaps independently from each other;" however, these data are pooled across trials, and so it is possible that FS cells could remap coherently within a trial but circularly across trials. To demonstrate independent remapping, circularity needs to be shown on a single trial basis. Please indicate how uniform rotation remapping is on a single trial basis.

Minor concerns:

Strong language assuming necessity of findings for behaviors and the cognitive processes underlying each behavior could be tempered – these findings are correlations, some with notable

variability, and the absence of behavioral manipulations or comparisons across brain regions means these findings are less definitive than presented

- This sentence from the abstract is not supported by the results: “Efficient heading retrieval and context recognition require integration of featural and geometric information in the active network through rate changes.” The data show that rate changes are correlated with improved heading retrieval, but requirement is not demonstrated.

- Page 25, paragraph 1: “CA1 is AN ideal substrate...”

Figures need additional labeling for clarity

- Figure 1 can almost be understood entirely without the text. Additional labelling is needed for panel E to explain that this is not merely another method of quantifying D but accounts for individual variability.

- Figure S2 polar plots need units for radial distance

- Findings related to polar measures (angle differences) need to more specifically say where the peaks are with appropriate decimal error when relevant (e.g. 5B not exactly at 0/180)

- 2A legend needs to say that the dotted line and arrow indicates compression for analysis

- Figure 2B,C: Heading over the left bar plots should say “electrophysiology” to reduce confusion with these results’ being about the proportion of single cells best aligned at each rotation

- Figure 3E, 4B,D,E need some kind of label indicating which comparisons are significantly different

- Figure 8A needs to say somewhere (legend, perhaps) that each circle is the average firing rate for a single trial

- Figure 8C: it seems that these are the averages of rate differences, not just the average rates as stated in the legend (add a label to the color scale as well)

Close copyedit for typos, grammar, vague references. Examples:

- Page 4, paragraph 2: “This pattern OF results”

- Page 4, paragraph 2: “moreover, IT/THIS PATTERN suggests” – number agreement of “it” with “this pattern”

- Page 25, paragraph 1: “ideal substrate to integrate THESE cognitive processes”

Methods should detail whether any distal cues are present in the room and any efforts made to reduce other stimuli such as noise from computers, vents, etc.

For many comparisons, it is quickly lost and often not mentioned in the figure whether it is within or across contexts. “Context similarity” is at different times used to refer to either

Circular definitions and results, especially like this example:

- Page 16, paragraph 2: “FS cells displayed feature sensitive remapping,” this is guaranteed since this is how FS cells were defined.

Figure 6C/D: legend states that “0 degree best match rotations also occurred more often than 180 degree in FS cells in both contexts,” implying this was not the case for FI cells, but the significance bars on both plots are identical.

The text supporting Figure 7 overcomplicates the question of how often does the alignment of the maps associate with the animal’s behavioral decisions, which is a separate question from how reliably the population activity is self-similar within context and distinct across contexts on a single trial basis. Two possible solutions: first, reorganize the text to make these points more evident. Second, choose a different approach to these analyses which is more intuitive to a reader, reducing the burden of understanding each separate analysis approach; this should use the same method for both analyses, such as the SVM from 7a or max population correlation with average map, as in 7c, or a widely used clustering method indicating how often a trial ends up in the wrong cluster.

The authors should offer some account for what the cells that rotate 90 or 270 are doing. In those cases, are the best aligned maps perhaps lower correlations than best aligned maps from 0 or 180? Are those proportions similar to other non-task related remapping seen in other studies?

Other:

Kinsky et al 2018 also showed “random” remapping across visits to the same context but coherent rotations of place fields aligned to environmental geometry for each remapping event: are the proportions of remapped cells similar as to here? Can differences be attributed to reward motivated behavior in this experiment?

From descriptions in the intro, it’s hard to tell the difference is between the adaptive cue combination theory vs. associative learning model: is the idea that adaptive cue combination works with a distribution of probability over the whole set of cues, and the associative learning model works with cues independently of each other? Also, unsure who in the sentence is doing the “adjusting” of their salience.

Requested analysis: How often is the second dig in the correct location? Greater chance of this following G digs than other? On the flip side, how often when the animals dig in a wrong N or F well do they then go to the C well vs. G?

Page 14, paragraph 1: in this situation “discrimination” is a less ambiguous term than recognition to describe decreasing similarity of activity across contexts. In this same paragraph, be cautious about the use of “remapping,” since it is sometimes used to refer to geometric rotation of place cells over trials in the same context and at other to refer to location remapped place fields across contexts

Page 20 Paragraph 1: mixing the current study’s findings in with a description of findings from another study makes it hard to keep track of what’s being explained.

Reviewer #4 (Remarks to the Author):

This report probes the cognitive and neural processes that underlie navigation and hippocampal function in mice, in a behavioral situation that requires the animals to distinguish between two task contexts—signaled by distinctive markings on the walls of the chamber in which the task is given—and to reorient themselves after slow turning in that environment. It combines three general methods—behavior, electrophysiological recordings of individual place cells in the hippocampus, and calcium imaging to allow re-identification of cells across time. Analyses of the three methods together make a powerful case for two separate neural systems in the hippocampus, one focused on the enduring shape of the environment (the locations, distances and directions of the bounding walls of the chamber) and the other focused on the transient changes in task demands that are signaled by visual cues in the chambers (patterns on the walls that differ from one another but that do not change the shape of the chamber itself).

Prior to the experiment proper, hungry rats (below normal body weight) learned that buried, consumable food would be provided in the room at different locations in each context (for example, either directly left of a visual cue in one room or directly right of a different visual cue in another room). On 3 days, they were then tested in a rectangular chamber with one of the sets of context cues in a state of disorientation. To find the food, they needed (a) to determine which task context they were in, and (b) reorient themselves within that context and dig for the food. The location of an animal's first dig provided the behavioral measure of their reorientation, but if I'm reading their methods right, the animals were allowed to continue searching in case of failure until they found and consumed the food: thus, if their first dig was wrong, they had an opportunity to explore the room further. Consistent with much past research (most beautifully, with the 2015 Julian et al paper), the authors found that reorientation on day 1 was based on the shape of the room, leading to high and approximately equal search at the correct location and at a location that was 180-degrees rotated away from it, providing evidence both for context recognition and for geometry-based reorientation. Over the next 2 days, however, first searches at the correct corner exceeded those at any other corner, suggesting that reorientation itself began to be influenced by the task context. The heart of the paper uses electrophysiology and calcium imaging to probe how the mice accomplished this feat.

Caveat: I am not an expert in the latter methods and hope other reviewers will give them a closer evaluation than I can provide. But the authors describe them clearly enough for reader like me to follow them. They distinguish between two kinds of cells in the hippocampus that are active and predict the animal's behavior in these experiments: a larger population of cells, dubbed FI, respond to the chamber's geometry, independently of the task context: across trials, they fire most at either the same location or at a location that is 180 degrees rotated from it, consistent with a process of reorienting by the chamber's rectangular shape and without regard to the pattern that signals the food's location in the current task context. A smaller proportion of cells, dubbed FS, change their firing depending on the task context. Interestingly, FI cell firing predicts reorientation behavior on all 3 days and never predicts behavior in relation to the context, whereas FS cell firing predicts context behavior on all 3 days and predicts reorientation on day 3, suggesting a learning effect. Further analyses show that correctly digging at the location of the food—which requires binding of

information for the task context and the animal's heading—is predicted by the firing rates of the hippocampal neurons that are active on that trial, on day 3 but not earlier.

These findings are beginning to solve what has always struck me as a great mystery. In many studies of navigation (including in humans, for example in the cited papers led by Doeller), the hippocampus appears automatically to track a navigator's position and heading in relation to the borders of the navigable environment—and not in relation to movable objects or transient visual cues. Yet in just as many or more studies of memory, the hippocampus is found to be critically involved in episodic memory: a capacity that requires that it be responsive to all sorts of transient events. I think this paper begins to suggest how to reconcile these two sets of findings. As we all were reminded during the pandemic-imposed lockdown, episodic memory gets really bad if the transient events that we want to remember all happen in the same location. Such memories may require that transient events be recorded in the context of a stable and enduring representation of places in the navigable layout. I think this paper is beginning to suggest how the hippocampus accomplishes this task, and I find it highly worthy of publication in Nature Communications.

Elizabeth Spelke

Response to reviewers.

We thank the reviewers for their thoughtful and constructive criticisms.

Below please find a detailed response to the concerns of each reviewer point by point. The reviewers' comments are in black and our answers in blue.

REVIEWER COMMENTS

Reviewer #1 (Remarks to the Author):

This study assesses the behavioral and place cell responses of disoriented mice in a task where the mouse has to find a buried reward in one corner of a rectangular environment. The authors show that the mice search for the buried reward in the correct corner and its rotational equivalent. This result implies that the disoriented mice use the geometry of the environment to reorient. The mice also successfully distinguish between two rectangular environments with different contextual cues. This result implies that the mice can use context to discriminate between environments. Interestingly, the mice seem to shift from a geometric solution to the use of the polarizing landmarks within each rectangle with training.

For the place cell recordings, place fields were observed to either stay in the same place or rotated by 180 degrees within a given rectangle. Across the two rectangles, place fields tended to either be found in the same location (feature insensitive cells) or to remap (feature sensitive cells). The latter showed greater differences across contexts with training, while the former tended to remain in the same location with additional experience. Calcium imaging of these cells showed, for consistently active cells, more stability on day three for the feature insensitive cells (compared with day 1) and more remapping across contexts for the feature sensitive cells. Place cells were not influenced by reward location but did show firing rate differences across contexts.

The behavioral results are interesting and are similar to those published previously by the same group (Julian et al., 2015). The advance in the current paper may be the finding of a shift in strategy with continued training.

Likewise, compared to a previous study assessing place cells in this task (Keinath et al., 2017), the current findings provide additional detail on place cell responses and show that some cells tend to be invariant across contexts while others remap.

The current findings are consistent with previous studies and provide additional information on the behaviour of place cells across contexts and acquisition. Given the previously published studies on this task, the significance of the current work is difficult to anticipate. However, the methodology is sound, the analyses appropriate and thorough, and the conclusions are well supported.

We thank the reviewer for his/her thorough comments and appreciation of our analysis methods. Regarding the significance of the current findings, we would like to point out a few conclusions that highlight the novelty of our current study:

1) In the study by Julian et al. (2015), we only conducted behavior. The current study expanded the significance of our previous findings by assessing the neural correlates of context recognition and heading retrieval.

2) We evaluated the effects of overtraining to demonstrate that geometric strategies are non-encapsulated. These results have strong theoretical implications because theories of reorientation still argue about whether geometry-based strategies are modular or not. Our data

supports a strong tendency for initial geometry-based reorientation along with an associative learning process that allows the incorporation of non-geometric, featural cues. Once the directional value of the featural cues is learned, their information is integrated into the representation of geometry through firing rate changes. These findings give neural support to the two-process theory set forth by Wang and Spelke (2002).

3) Most studies of reorientation, including those involving brain recordings, look at the characteristics of different cues without recording during learning. Even in our previous study in the hippocampus, predictions were computed on day 3, after animals had learned the task (Keinath et al., 2017). Here, we recorded during learning to evaluate the switch in reorientation strategies and the changes in neural representations as animals learn to associate feature cues with reward.

4) Finally, following the reviewers' suggestions, we have now incorporated several changes that strengthen the conclusions and novelty of our paper. Briefly, we now include a thorough validation of the remapping threshold, an in-depth analysis of the rotation patterns in FI and FS cells, an evaluation of cell identity over time, and a demonstration that early representations of geometry have predictive value across training days.

Reviewer #2 (Remarks to the Author):

Summary

In this study Gagliardi et al. investigate the neural mechanisms within the context that support important features of spatial memory: context recognition and heading retrieval. A previous study from this lab demonstrated that spontaneous rotations of place fields would sometime spontaneously rotate following disorientation during a spatial memory task (Keinath et al, 2017). These rotations were not random, but were anchored to the geometry of the arena, and importantly, they occurred primarily when animals made errors in digging for a food reward (e.g., a 180-degree field rotation would predict that the animal dug for food reward in the corner diagonally opposite the correct food location). A behavioral study from the same lab (Julian et al., 2015) demonstrated that these behavioral errors and rotations occurred even when animals had to utilize non-geometric features to guide their digging location in two different contexts. This study concluded that contextual identification and heading retrieval are governed by distinct neural mechanism. In the current study, Gagliardi characterize these mechanisms using calcium imaging and electrophysiology to record real-time neural activity as mice perform a spatial memory reorientation task similar to that in Julian et al. (2015) which requires mice to learn that reward is hidden in one corner (upper-right) of one rectangular arena (Context A), while it is hidden in a different corner (upper-left) of an identically shaped arena (Context B) that has distinct non-geometric features from Context A.

The authors find that a large subset of cells in the hippocampus maintain a similar set of place fields between contexts, which they dub feature insensitive (FI) cells since their firing configuration is not influence by the non-geometric features of each environment. A small subset of cells does move, or remap, between contexts, and the authors call these feature sensitive (FS) cells. In line with their previous work, FI cells exhibit rotations that match the geometry of the arena while FS cells do not. Decoding analyses demonstrate that FI and FS cell activity predicts heading and context, respectively, supporting the authors hypothesis that FI and FS cells provide separate neural mechanisms to support these distinct cognitive demands. In one of the most interesting findings, FS cells also gain the ability to decode heading on day 3 of the task, which is imp. This suggests that integrating heading and context retrieval into the same group of neurons could underlie the increase in behavioral performance observed in Figure 1. This hints that the “conjunctive encoding” or “mixed selectivity” observed in

hippocampal neurons is not innate but might be learning dependent, and this learning dependence has been overlooked previously since most such studies employ overtrained animals and do not record learning data.

Overall, this study provides an important advancement in our understanding of the neural mechanisms that underlie spatial memory and orientation. Gagliardi et al. demonstrate that these cognitive mechanisms are supported by separate sets of neurons in the hippocampus that independently code for heading/context retrieval (and intriguingly might begin to mix on day 3). However, there are several key issues listed in the “major concerns” section below that need to be addressed in order to heighten confidence in the robustness of the results and guide larger interpretations: the lack of a direct, cell-to-cell assessment of feature insensitivity between arenas, the lack of individual animal/session data provided, and a lack of clarity in how aligned maps were calculated in the majority of the figures.

Major concerns

The key issue is that the analyses performed do not directly assess if cells are truly feature insensitive (FI) or feature sensitive (FS). This is for three reasons which are also identified in the sections below: 1) the definition of “feature” is not clear, 2) many cells identified as FI could actually be FS cells due to performing rotations for each cell individually rather than rotating all cells’ place fields together the same amount, and 3) FS cells have much lower stability even within a context, suggesting that they might simply be less reliable, have lower firing rates, or undergo representational drift. Additionally, the authors do not provide any rationale for the cutoff used to separate FS vs. FI cells, which need to be addressed.

We thank the reviewer for his/her thorough review and criticisms. Below we address each point in detail. After reading the reviews we noticed that there was some overlap between comments from reviewer 2 and 3. We thank these reviewers for the useful suggestions, which have helped to strengthen our conclusions. In those cases, we combined the suggestions from both reviewers and generated a combined response. To facilitate the task for each reviewer, these responses are copied below the point raised by each of them.

The points listed in this section as major concerns are addressed in depth in the sections below as reviewer 2 brings them up. Here, we only provide a brief answer describing how we addressed the main points listed above.

- 1) *We clarified that that feature refers to the visual cues present on the walls of the chambers.*
“Throughout this paper, we will use the term feature exclusively to refer to the visual cues placed on the walls of each chamber.” Pg.5
- 2) *In the current version of the manuscript, we conducted additional analysis to rule out the possibility that FI cells were misclassified as FS (details below). To this end, we examined individual cell rotation patterns to examine the possibility that a cell could have been mis-classified as FI when, in reality, it was FS. This could occur if cells display stability in context A and B but rotates 180 between contexts, which following alignment would be misclassified as FI. We exhaustively analyzed the rotation sequences to rule out this possibility. Among all cells recorded we found only 1 cell in 1 session following this pattern (Figure S6). Additionally, we conducted a coherency analysis and establish different patterns of population rotations across context in FI and FS. This analysis led us to uncover the presence of a primary and second coherency characterized by distinct temporal dynamics in FS and FI cells. These differences in combination with the*

thorough validation of the remapping threshold that we provide in the current version of the paper (see point 5 below and answer 5 in the detailed response section) provide confidence that we did not misclassify cells in our analysis.

- 3) *We now report spatial information (SI) content in FS and FI cells. Although there was a reduction in spatial information in FI cells across days, there were no significant changes in FS cells, suggesting that their remapping properties were not due to poor spatial tuning. Moreover, there were no differences between FS and FI on any day.*
- 4) *A requirement for validation of the remapping threshold was raised by reviewers 2 and 3. We followed the recommendation of reviewer 3 who suggested to divide the distribution of similarity scores in deciles and analyze the data to justify the selection of the threshold. To this end, we plotted correlations within and across context dividing the scatterplots in deciles from the similarity correlations shown in Figure 3C. Then, we modeled the function representing the correlation corresponding to each decile using a regression model that only required 2 parameters (detailed explanation of our modeling approach is below). We demonstrate that decile 1, which corresponds to correlations between 0 and 0.3 is the most informative to classify the data.*

Below we address every point raised by Reviewer 2 in depth.

1. Is the term “feature” used exclusively to mean “non-geometric features” (p. 10) such as lines on the wall or different odors between contexts? I suggest making this explicitly clear early on to avoid confusion. What is included as a “feature”? Is time between exposures considered a feature? Without a clear definition it is difficult to properly interpret most of the results.

We used the term “feature” to exclusively refer to non-geometric features. In the paper, we clarified the meaning of features to avoid confusion: “Throughout this paper, we will use the term feature exclusively to refer to the visual cues placed on the walls of each chamber .” Pg. 5

We also added in the introduction the following clarification: “Non-geometric featural cues encompass elements that do not define the shape of the navigable space, such as signs, objects and textures with distinct visual, olfactory, and auditory characteristics.” Pg. 3

2. Almost all the plots are presented using data pooled across all animals. To get a better sense of the inter-animal variability for all results, the authors need to provide data for individual animals wherever possible, in particular for the plots and analyses noted below.

We remade all graphs in the manuscripts adding individual data points. In some cases, in which the analysis was done per cell, but the graphs included many conditions, we presented box plots and/or violin plots along with the histograms to show extend of variability (Figures S8 and S9). We now also added a table with the individual digging proportions (Table S1). Finally, please also note that we included all the individual Bayes factors for the behavioral analysis in Figure 1, which also illustrate individual performance.

3. It is not clear how map alignment was performed from page 10 onwards. This is crucial to understanding the definition of FI and FS cells throughout the paper and interpreting the results. Per Keinath et al. (2017), place fields tend to rotate together across the neuronal population and that the orientation of the neuronal population is important for predicting behavior. However, Figure 3A and the text imply that rotations for each trial were calculated for each cell

independently, e.g., cell #1's rotation sequence might be 10010 while cell #2 might have the rotation 01010 for the same session.

We thank the reviewer for pointing out that the results and method description were unclear. Since 0° and 180° coincide with the rotational symmetry of the rectangle and dominate the pattern of rotations within and across context in most cells (Figures 2C, S2), we had to first align the maps relative to this axis to determine if there was remapping across context. In the current version of the paper, we eliminated the binary sequences because we changed the prediction method following a suggestion of reviewer 3 (binary sequences were only used for prediction). Below we explain the alignment method, which we hope is more straightforward now. We also conducted coherency analysis to prove that there is coherency above chance in the population. However, it is important to note that even though we find coherency above chance levels, not every cell rotates coherently, which coincides with previous observations in oriented mice (Kinsky et al., 2018). Therefore, the alignment must be done per cell to isolate the different cell response types.

We clarified the method to assess context similarity in the text as follows: "Place maps for each cell were aligned to the same orientation within each context by selecting the highest correlation between 0° or 180° rotations that maximized spatial map similarity for all pairwise comparisons within each context (Figure 3B). Then, an average aligned map was computed for each context and these two average maps were aligned relative to each other (e.g., best match between 0° or 180° rotations). Context similarity was defined as the highest pixel-to-pixel cross-correlation between the rotated or non-rotated average aligned maps in each context." Pg. 9

We also introduced a schematic of the possibilities, as suggested by the reviewer to facilitate understanding of the potential rotation patterns (Figure 3A)

Finally, we corrected the examples and Figure 3B to make it clear for readers.

4. Likewise, it is unclear if a similar method was used for between context aligned map correlations – were only the best correlations used? The semi-transparent rate map in Figure 3A seems to indicate that between-arena correlations were calculated at 0- and 180-degree rotations, and only the highest was used, but this information is not found anywhere in the text or methods. As such, the authors could be overestimating the proportion of FI cells and missing FS cells. Take the example below (in attached file with sketch) – the blue, red, and green place fields maintain the same configuration but rotate 180 degrees between trials in A and B. After alignment, these place fields would have high correlations and would be FI cells. However, the beige-dashed and purple dashed fields would also have a high correlation after alignment and could be designated as FI cells, even though they behave very differently as they do not rotate with the rest of the cells during Trial #1 in Context B.

With this in mind, many FI cells could actually be sensitive to the features of different contexts. One way to address this is to assess pairwise rotations between cells. Do pairs of FI cells always (or almost always) rotate together the same amount between trials across contexts, whereas FS cells do not? Results from Figure 7 indirectly addresses this within the same context, however a direct comparison of rotations between cells on a trial-by-trial basis across contexts would increase confidence that FI cells are truly insensitive to non-geometric contextual features.

We thank the reviewer for pointing out that our schematic was not clear and a possible caveat in the estimation of FS and FI cells. We are also grateful for his/her schematic, which we have now incorporated in our paper to illustrate how the cells could be rotating across context

(Figure 5A). In the text, we made it clear that we indeed take the highest correlation between the average aligned maps. In the next paragraphs we address the concern of the reviewer in two different ways. 1) We analyzed if our alignment method could have led to misclassification of cells. 2) We evaluated the coherency in FS and FI to determine if they were indeed different (this point was also raised by reviewer 3 and we used the reviewers' suggestions to address it).

1) Reviewer 2 is correct about a potential caveat of our alignment method. A misclassification could occur if a cell is stable in context A and B but rotates 180° between contexts. We were not originally concerned about this because looking at our data we did not observe this pattern. However, to address the possibility in depth, we isolated any cell that yielded stable opposite patterns across context using the alignment rotational sequences generated per context (we programmatically record the pattern of rotations used for alignment, which facilitated this analysis). Among all recorded neurons, we found only 1 cell showing stability in each context and remapping across contexts (shown below). This cell was recorded with tetrodes on day 1 (animal JJ9, TT5, cell 04). Although we did not attempt to record the same cells on following days using electrophysiological techniques, it is likely that some of the tetrode cells were the same. However, the high stability pattern in each context was not observed in JJ9, tetrode 5, in any cell on days 2 and 3 (or any other cells). Therefore, considering the low occurrence of this cell type, and the fact that this pattern was not present when the animals distinguished the contexts more efficiently, led us to conclude that this rotation happened by chance. We addressed this in the paper and show this unique pattern in the supplement (Figure S6).

We added: "It is important to highlight that if a cell remaps between contexts, but the field locations are 180° from each other, then our alignment method may erroneously categorize such cells as FI when in fact it is a FS cells. To address this potential misclassification, we examined cells exhibiting high stability in each context (those not requiring alignment). Among all recorded cells, only one cell displayed the described pattern on day 1, but this pattern did not recur on subsequent days. Consequently, we attributed this occurrence to chance and concluded that these cells were not representative of our dataset (Figure S6)." Pg.10

Figure S6. Complement of Figure 3. The place cell maps shown above correspond to a cell identified among 2669 cells. This cell exhibited high stability in each context but underwent a 180° rotation between contexts (animal JJ9, tetrode 5, cell 4). While this cell serves as an illustration of a perfect FS cell that could have been misclassified by our alignment procedure, no other cells displayed this consistent pattern (we used the sequence of rotations generated by our alignment method to identify any cell that showed stability in one or both contexts, finding only the cell shown above). Notably, this specific cell was recorded with tetrodes on Day 1, and similar cells were not observed on days 2 or 3 in the same animal. Consequently, we concluded that this pattern of results was not representative of our dataset. Trials 1 to 4, which included a visible reward, were excluded from the analysis.

2) The point about whether FI and FS cells rotate together the same amount between trials within and across context was raised by reviewer 2 and 3. The schematic provided by reviewer 2 indicated that we needed to disentangle whether the remapping across context was coherent or global, and reviewer 3 suggested that our center out analysis could have occluded coherency because we were averaging across trials and animals. We conducted exhaustive analyses to test how the cells were rotating across context and concluded that both reviewers were correct: There was coherency in both FS and FI subpopulations. Below we describe the new analysis and interpretation of the results.

To determine if there was coherency in the population, we assessed the proportion of best match rotation (BMR) angles (0° , 90° , 180° , and 270°) across trial pairs in FS and FI cells within and across context and averaged the data per animal (Figure 5B shows schematic of method). This analysis yielded 4 proportions of cells rotating together between trials (based on the 4 angles used for the analysis), which are shown in Figure 5C-D. Coherent remapping in the population would be indicated by a predominant proportion of cells sharing the same BMR angle between trial pairs). To determine significance, the coherent proportions of cells were compared with chance assuming equal distribution of angles (25%).

Using this analysis, we found that both FS and FI cells displayed coherency. Both cell types show a dominant coherency (1st BMR) as well as a second one (2nd BMR) that were above chance level (although the significance of the secondary coherency varied with cell type and day of training). Interestingly, the 1st and 2nd BMR displayed consistent levels that were above chance across days in FI cells; however, the 1st coherency increased, while the 2nd coherency decreased across days in FS cells (Figure 5E). This indicated that both cell types displayed coherent patterns of rotation within and across context, but the coherency in FI cell is constant, whereas the coherency in FS is modulated by learning.

To examine the angular relationship between the 1st and 2nd BMR, we plotted the joint distribution of their rotations (Figure 5F). We found that in FI cells the 1st and 2nd BMR maintained a geometric relationship (e.g., if cells displayed a 180° angular rotation in the 1st BMR, the 2nd BMR would remain stable at 0°). This result further supports the involvement of FI cells in representations of geometry. However, in FS cells there was no clear relationship between the 1st and 2nd BMR. Interestingly, the 1st BMR marginal probability indicated that the 1st coherency in FS cells aligns to opposite axes across context, a pattern that could serve to facilitate context recognition.

General comments

5. How/why was 0.3 selected as the context similarity cutoff? How robust are the rest of the findings, in particular the decoding analyses in Figure 7-8, to this threshold?

This point was raised by reviewers 2 and 3 so we generated a common response following the suggestions of both reviewers.

We thank reviewers 2 and 3 for making suggestions to strengthen the selection of the remapping threshold. To validate this threshold, we first used reviewer's 3 suggestion of plotting the within and across similarity scores by dividing the distribution of average similarity correlations in deciles (Figure S4A). Then, we calculated the correlation of the within and across similarity scores for each decile and used a modeling approach to analyze the function (Figure S4B). Decile 1, comprising correlations between 0 and 0.3, emerged as the most informative because it coincided with the function's half-life and root. Second, following the suggestion of reviewer 2, we also plotted context prediction as a function of remapping threshold. Below is our detailed response to this critical point.

1. Decile analysis and modeling.

Figure S4. Complement of Figure 3. Validation of remapping threshold. A. Scatterplots showing average correlations within and across context for individual cells separated in deciles obtained from the similarity distribution shown in Figure 3C. B. Asymptotic regression model of the overall correlation decile function. The red dot indicates the half-life of the function, which coincides with decile 1 (correlations across context between 0 and 0.3) and the root of the function (value that makes the function 0 on the y axis). Finally, in the asymptotic regression model, the relative growth rate is not constant. It attains its peak when $Y = 0$ and diminishes as Y increases. This suggests that Decile 1, corresponding to $Y = 0$, represents the point at which the rate of change is maximized. This indicates that the first decile is the most informative to discriminate across context.

Description of modeling approach

To select the best correlation threshold to characterize similarity scores, we first separated the cells into deciles using context similarity distribution shown in old Figure 3B (current Figure 3C). For each cell, we plotted the average within and across context correlation (average correlations included all pairwise comparisons corresponding to each cell) and generated scatterplots for each decile (Figure S4A). We then calculated a Pearson's correlation coefficient between the average within and average across correlations for the associated cells per decile, which yielded a set of 10 values (one per decile). We noted that the largest difference occurred between the first and second decile. To understand this change, we used a modeling approach to fit the data (Figure S4B).

To identify the optimal non-linear model, a two-stage goodness-of-fit analysis was conducted. First, significant points (Roots, Extremas, and Inflections) were pinpointed in the correlation decile function shown in Figure S5B using the Taylor Regression algorithm (Christopoulos, 2014). Root values denote the X-axis values leading to null Y-axis values, while inflection points, maxima, and minima were determined through derivatives of the function. Second, the best model was selected from an exhaustive pool of alternatives based on the lowest values for Akaike's information criterion (AIC), Residual Standard Error (RSE), and the highest value for log likelihood (L-L) a set of metrics gauging goodness of fit (Ritz et al., 2015). Notably, Sigmoidal (based on inflection points) and type Maxima/Minima models were excluded due to their inability to yield a satisfactory fit for the data.

This led us to focus on concave-convex or exponential-type models, from which the Asymptotic Regression model (Pineiro & Bates, 2000) exhibited the best results in terms of goodness of fit [AIC = -27.75, RSE = .0484, L-L= 17.88, see below for details].

Asymptotic Regression model: $Correlation = q1 + (q2 - q1) \cdot \exp(-\exp(q3) \cdot Decile)$.
In this model, q1: Asymptote, q2: Origin, and q3: Rate (on a logarithmic scale).

The term "Origin" refers to the initial correlation value, while the "Asymptote" signifies the point at which correlations stabilize. The "Rate" parameter captures the speed of the correlation function change from the origin to the asymptote, computed as the natural logarithm of the rate constant. The estimated values for the three parameters were all statistically significant (Asymptote = 0.83, Origin = -0.78, LogRate = -0.35; $p < .05$). We then estimated the half-life [$X_{0.5} = \log 2 / Rate = 0.99 \approx 1$, see Figure S4B], which identified the correlation value that reached half of the function's rate. The concept of half-life, a parameter commonly used to characterize exponential changes, is detailed in Pineiro and Bates (2000). In exponential models, the half-life plays a role analogous to the inflection point in sigmoidal (logarithmic) models. In the Correlation-Decile function, Decile 1, comprising correlation values between 0 and 0.3, corresponds to the function's half-life. Additionally, Decile 1 also corresponds to the root of the polynomial function, which is the value that turns the function to zero. Finally, the relative growth rate in the asymptotic regression model (first derivative definition is shown on Figure S4B) is not constant, reaching its maximum value when $Y = 0$ and decreasing as Y increases. This suggests that Decile 1, corresponding to $Y = 0$, represents the point at which the rate of change

is maximized. These results suggest that the optimal correlation threshold value to separate cells based on remapping properties is 0.3.

Table S4

Parameter	Estimate	sd	t	Pr(> t)
Asymptote	0.8300	0.02	34.84	0.0000***
Origin	-0.7800	0.21	-3.65	0.0082**
Rate (Log)	-0.3500	0.15	-2.37	0.0497*

Note: *** $p \leq .001$, ** $p \leq .01$, * $p \leq .05$

RSE = 0.0484, log likelihood = 17.88, Akaike's information criterion = -27.75

References

References

Christopoulos, D.T. (2014). Roots, extrema, and inflection points by using a proper Taylor regression procedure. SSRN. <https://dx.doi.org/10.2139/ssrn.2521403>

Pinheiro, J.C. and Bates, D.M. (2000) Mixed-effects models in S and Splus. Springer.

Ritz, C., Baty, F., Streibig, J. C., and Gerhard, D. (2015) Dose-Response Analysis Using R. PLOS ONE, 10(12), e0146021

2) Context prediction as a function of remapping threshold

As requested by reviewer 2, we also plotted context prediction as a function of the remapping threshold. Note that low remapping thresholds are best at predicting context; however, the prediction plateaus between 0.2-0.3 thresholds. At values higher than 0.3, there is a strong decay in the context prediction accuracy. We believe that the modeling of the decile correlation function strengthens our selection of the threshold value 0.3. Therefore, we only included the graphs shown below in this letter.

Context prediction as a function of remapping threshold for, tetrode, calcium, and combined data. As expected, context prediction decreases as thresholds values decrease. However, there is a plateau between 0.2-0.3 correlation values, and a substantial decline at higher correlation values.

6. The authors are inconsistent in the exclusion of days and rotation amounts from their analyses. For example, Figures 2 and 6 consider rotations of 0, 90, 180, and 270 degrees while Figure 3/7 only considers 0 and 180. Why? Likewise, why is only Day 1 and Day considered for FS vs FI cell correlations analyses in Figure 3/4 while Day 2 is also included for rotational analyses in Figure 5?

The Figure numbers have changed in the current version of the paper. However, it is worth noting that the only instance in which we only consider 0 and 180 rotations is for the alignment used to generate average similarity across context, as we explained in point 3, we cannot separate the cells that rotate on opposite axis across context if we use all angles. However, in all the analysis of best match rotations, including our coherency analysis, the 4 angles were considered. We now excluded the cumulative curves with consistently active and temporary active cells (previous Figure 4D and 4E) because we conducted the analysis suggested by the reviewer (point 8 below) checking the identity of the FI and FS cells. Finally, we added day 2 to the population vector curves (Figure 4B).

7. The authors focus on 0- and 180-degree rotations in many analyses, yet many 90 and 270 degree rotations are observed at both the behavioral level (F and N trials) and the neural level (90 and 270 degree place field rotations in Figure 2). Are these phenomena related? Do 90 and 270 degree rotations of place fields occur mainly when mice perform F or N trials? If FI cells are truly important for heading retrieval, one would predict that 90 and 270 degree place field rotations of these cells would occur primarily during F and N trials.

To determine what happened during 90° and 270° rotations, we first tried to predict errors using the center out measure. This measure evaluated the angle from the center of the arena to the center of mass of the place field. We hypothesized that if 90 and 270 rotations were associated with errors, we would be able to predict behavior on these trials. To this end, we separated all the error trials where animals dug in Near or Far locations and trained a support vector machine classifier, withholding the trial to be classified. Error prediction was not significant on any day (Mean predictions for error trials: Day 1: 65±16%, $p=0.16$, Day 2: 50±14%, $p=0.49$; Day 3: 58±16%, $p > .05$).

A failure to predict error trials could be due to the fact that these trials are less common. Therefore, to further explore what happens during behavioral sequences associated with 90 and 270 rotations we analyzed the sequence of digs from Trial A to Trial B as a function of best match rotation angle. In other words, for each angular rotation between trials (0, 90, 180, and 270), we plotted the behavioral sequences observed. The results showed that C/G and C/C were the most likely dig sequences. Notably, 90° and 270° rotations happened sporadically, even when animals made C/C or C/G response sequences (we corroborated this result with a variety of methods, and all converged to the same conclusion). Therefore, we can only conclude that the alignment is not perfect and there is some noise in the neural data.

In our previous study we observed similar rotations during reorientation in the traditional rectangular one context task (~15% 90° and 270° rotations; Keinath et al., 2017). Interestingly, we also observed some random rotations in an isosceles triangle, an environment without rotational symmetry [72% stability (0° rotation), with ~15% rotations coinciding with either 120° or 240°, suggesting that random rotations happen even in the absence of rotational symmetry, Keinath et al., 2017). These sporadic rotations may reflect the inherent noise in neural systems.

These data are now added in Supplemental Figure S3. We also added a sentence to the results: “we plotted the counts that corresponded to each pair of behavioral responses for each best match rotation angle (Figure S4). We found that correct behavioral sequences (C/G or G/C) were prevalent at all best match rotations, including 90° and 270°, suggesting that non-geometric rotations happen sporadically and may reflect noise at the neural level.” Pg. 8.

Figure S3. Proportion of sequence of digs per trial as a function of place field angular rotation (0°, 90°, 180°, 270°). Note that C/C and C/G are the most prevalent dig sequences, which is illustrated in vivid yellow/green colors. 90° and 270° rotations happen sporadically, even during C/C G/C or C/G sequences, which may reflect inherent noise in neural data. Colors indicate response counts, with yellow reflecting higher counts and dark blue lower ones.

8. The authors leverage one of the strengths of miniscope calcium imaging: the ability to record from large cell ensembles. However, they do not utilize the other strength of imaging – the ability to confidently track cells across days – and therefore miss the opportunity to address a key question about the permanence of FS vs FI cells. Are these cells drawn from separate pools that stay separate on different days? Or do FS cells and FI cells flip-flop on different days? Answering this question, for example by training the decoder from Figure 7 on FI cells from Day 1 and then decoding heading from those same cells on Days 2 and Days 3, would help ascertain if FI cells are hard-wired to be FI cells can change their identity (phenotype?) with

experience. This would provide valuable insights into the larger circuit mechanisms involved in shaping the responses of these cells. For example, if FI cells are truly hard-wired to be FI cells, do they receive stronger input from head-direction cells in thalamus or entorhinal cortex than other cells?

We conducted the analysis suggested by the reviewer and the results are shown in Figure 6. We registered the cells on day 1 and examined their remapping characteristics on days 2 and 3. We found that most FI cells remained stable on days 2 and 3. Therefore, we trained the decoder using FI cells registered on day 1 or day 2 and predicted heading on day 3 ($p < .01$ for both predictions). Additionally, we trained the decoder with data from day 1 to predict day 2 ($p = .054$). The results indicated that FI cells can predict heading across days, which coincides with the fact that their identity remains stable. Interestingly, when we train the decoder with data from day 1 to predict heading the same day, we do not reach significance ($p = 0.16$), but the data from day 1 significantly predicts heading on day 3. This suggests that although the map of geometry forms fast, the effective use of the map for heading retrieval improves with experience.

We suggest that the identity stability of FI cells is due to the fact that in our task geometry is a stable environmental cue. We do not have enough evidence to conclude that the representation of geometry is hardwired, as postulated by the modular theory. Our belief is that reorientation representations can be modified by experience. There is evidence showing that animals raised in circular environments do not use geometry to reorient on subsequent tasks involving geometry. For instance, Convict Fish reared in circular chambers exhibited a greater reliance on features compared to their counterparts reared in rectangular environments (Brown et al., 2007). Likewise, mice reared in a circular chamber rich in features demonstrated differences in reorientation compared to mice reared in geometrically enriched surroundings. Specifically, mice reared in circular environments displayed quicker acquisition of featural information, but were unable to encode the chamber's geometry, whereas mice reared in rectangular chambers showed the opposite pattern (Twyman et al., 2013). These outcomes imply that experiential factors play a pivotal role in influencing the utilization of geometry and likely exert a substantial impact on its cognitive representations.

Conversely, FS cells do not display consistent identity across days, which illustrates that FS ensembles are dynamic. Since the total percentage of FS cells remains relatively constant across testing days, these results suggest that the ensembles of FS cells incorporate temporarily active cells (e.g., cells that become part of the active ensemble only on specific testing days). The differences in identity stability between FS and FI may reflect the fact that FI cells respond to the stable shape of the environment (geometry), whereas FS cells respond to feature/reward associations, which are dynamic components modulated by experience. Studies on engram cells participating in learning have underscored the dynamic properties of these cells (Kitamura et al., 2017; DeNardo et al., 2019). To address this important point, we added a section (pg.12), Figure 6, and a paragraph in the discussion (pg. 14-15).

Specific comments

9. Figure 1A-B: Please provide individual data points for behavioral performance to aid in reproducibility by other researchers and provide an accurate picture of behavioral variability and how it changes from days 1-3.

We added individual data points to the graphs shown in Figure 1. Table S1 displays individual data points per context and day (complement of Figures 1C) and Figures(1F and 1E

show individual data points. Additionally, individual data can be seen in the BF shown in Tables S2 and S3.

10. Figure 1E: Are these behavioral plots for both arenas combined? Are behavioral strategies invariant between context A and context B? In other words, is $\log(\text{BF})_A$ shown in Figure S1 correlated with $\log(\text{BF})_B$?

Figure 1E **now Figure 1G** tests the model that animals dig in the correct cup over other locations for both contexts combined, whereas **Figure 1F** tests the model that animals dig in the correct geometric axis (Correct and Geo cups) more than in the incorrect axis (Near and Far cup locations) in each context.

Note that the cumulative functions for subjects are very similar across contexts [**Figure 1F (old Figure 1S)**]. Although animals seem to learn context B faster than A on day 2 (Percentage of animals showing negative BF values on Day 2: Context A: 28.57%, Context B: 7.14%), this effect diminishes on Day 3 (Percentage of animals showing negative BF values on Day 3: Context A: 35.71%; Context B: 21.43%). Percentages are obtained from the cumulative proportion where each curve cuts the line drawn at zero value. This information was added to the legend of Figure 1F.

Relevant part of legend added to Figure 1: “ F) Cumulative proportion of Bayes Factor (BF) on days 1 to 3 per context evaluating the alternative model (M_{Alt}) that animals preferentially dug on a distinct rewarded axis in each context vs. the null model (M_{null}) that animals dug by chance. The global BF showed credibility in support of M_{Alt} in each context across days [Context A: group BFs (days 1, 2, and 3): 4.16, 7.23 and 6.94; Context B: group BFs (days 1, 2, and 3): 4.18, 8.60, and 8.56; for details on individual data see Table S2]. Note that in both contexts the group BFs are >1.1 indicating credibility for learning. Moreover, the point at which the cumulative proportion intercepts the dotted line at 0 shows that even though there are some subtle differences in the rate of learning on day 2, these differences decrease on day 3 (Percentages on negative values: Day 2: context A: 28.57%, Context B: 7.14%; Day 3: Context A: 35.71%, Context B: 21.43%). G) Cumulative proportion of individual Bayes Factors (BF) across days 1 to 3 combining contexts evaluating the alternative model (M_{Alt}) that animals increased digging in C locations with experience vs. the null model (M_{null}) that animals dug by chance. Global BFs indicate that animals do not preferentially dig in the C cup on day 1 (Group BF=-2.60), favoring M_{null} . However, M_{Alt} is favored on Days 2 and 3 (Group BF= 15.06 and 17.68, respectively), favoring the model that supports increased digging in the C cup location. Conventional values showing the border marking credibility for M_{alt} ($\log(\text{BF}) > \log(1/3)= 1.1$) and M_{null} ($\log(\text{BF}) < \log(1/3)= -1.1$) are indicated by vertical dashed lines. The value of half (0.5) of the sample is marked by a horizontal dashed line. ”

We also want to note that correlations of Bayes factors can be deceiving and cannot be used to evaluate learning. For example, in the hypothetical cases shown below a perfect correlation does not illustrate dissimilar learning between 2 contexts or a zero-correlation equivalent learning. Therefore, we used the intersection points with the zero vertical line to evaluate learning in each context on days 2 and 3.

[1] A ≠ B Perfect Correlation			[4] A = B Null Correlation		
Sj	A	B	Sj	A	B
1	-0.1	-100	1	-0.1	0.1
2	0.1	100	2	0.1	0.1
3	0.2	200	3	0.2	0.1
4	0.3	300	4	0.3	0.1
5	0.4	400	5	0.4	0.1
6	0.5	500	6	0.5	0.1
Correlation 1.00			Correlation 0.00		

11. Figure S1 and page 7: Are BF analyses significant on all days here or just 2 and 3 like full model above? State result in main text.

In old Figure S1 (now Figure 1F) the group BF on days 1, 2, and 3 supported the alternative model (Malt). In all cases the global BF was > Log BF(3)=1.1, indicating credibility for the model that supports learning. This was clarified in the main text and the caption of Figure 1. We address this point in detail above (Reviewer's point 10).

12. Figure 2C: Please provide individual data points. How is the post hoc Rom's test being applied? Were as each day-rotation compared to every other day-rotation? Or were days grouped together for each rotation before comparing across rotations? It is not possible to tell from the figure.

Individual data points were added to Figure 2C. Since the interaction day by rotation was not significant, the post hoc Rom's test was calculated by grouping across days. We clarified this on Page 8, paragraph 3, in the following sentence: "Post hoc analysis on the main effect of the rotation variable (days grouped together for each rotation before comparing across rotations) showed that..."

13. Pg 10: The two possibilities outlined at the top of the page are hard to follow. A graphical schematic outlining each would make this much clearer and leave less up to the interpretation of the reader. For example, does "geometric alignment" mean cells have the same configuration relative to each other in both contexts?

We thank the reviewer for the suggestion. We added a schematic illustrating the possibilities, which is depicted below. Possibility 1 suggests that cells align to geometry in each context, but there is location remapping across context. Possibility 2 suggests that one group of cells shows the same geometric alignment within and across context (same configuration), showing no evidence of feature-sensitive remapping across context, but another group shows geometric alignment within context and location remapping across context, illustrating partial remapping in the population.

14. Figure 3A and pg. 10 text: Why are only 0 and 180 used for map alignment? I understand that these rotations are predominantly supposed to align to the arena geometry per Keinath et al., 2017, but Figure 2 shows that 90 and 270 rotations also occur fairly frequently.

Our previous findings and current results strongly indicate that 0° and 180° are the most dominant rotations in disoriented animals. We hypothesized that only cells showing distinct rotations within and across context could be involved in context recognition. Therefore, we used 0° and 180° rotations to separate cells that primarily respond to geometry vs. cells that may respond to other cues. If we align the maps using all possible angles, it would be impossible to distinguish the cells that remain stable across context vs. those that do not. In point 7 above we answered how we evaluated the 90° and 270° rotations.

15. Figure 3C: How consistent are these proportions across mice? Please provide data points for individual mice.
Individual data points are added to Figure S5.

16. Figure 3E: It appears that FS cells are generally unstable and less reliable, as indicated by their lower within context correlations. This makes it questionable if they are truly “feature sensitive” or just bad place cells? Likewise, lower correlations could result from lower event rates for FS cells, especially for calcium imaging. To accurately characterize these cells and assure that their feature sensitivity is not an artifact of lower event rates/firing rates or poor spatial specificity, the authors should quantify event/firing rate and spatial information (or another metric of place field quality) between FI and FS cells.

We calculated spatial information content and show the data in Figure S7. We found a small decrease in spatial information in FI cells; however, the spatial information content of FS did not change across days and there were no differences between FS and FI cells on any day of training. These results indicate that the remapping of FS is not due to poor spatial quality.

17. Figure 4: How were cells that had few or no transients in one context handled? Why is only Day 1 and Day 3 considered for FS vs FI cell correlations analyses in Figure 3/4 while Day 2 is also included for rotational analyses in Figure 5? Exclusionary criteria should be clearly noted in the text/methods. Figure 4B: are correlations for all cells active in both sessions? Or does this include inactive cells too?

The population vectors only included cells active in both contexts in each session because the dot product could not be calculated otherwise. This was added to the method pg. 21. The new Figure 4B now shows the dot product values including day 2.

We replaced previous Figure 4D and 4E following the reviewer's suggestion in point 8. Now we show pie charts indicating the stability of cell identity across days. Note that while most FS cells retain their identity, FI cells do not. The differences in identity stability between FS and FI may relate to the fact that FS respond to stable characteristics of the environment (geometry), whereas FI cells respond to feature/ reward associations, which are dynamic components modulated by experience. Previous studies on populations of cells involved in associative forms of learning have highlighted the dynamic nature of these ensembles (DeNardo et al., 2019; Kitamura et al., 2017). We added this new analysis in Figure 6.

*DeNardo, L. A., Liu, C. D., Allen, W. E., Adams, E. L., Friedmann, D., Fu, L., . . . Luo, L. (2019). Temporal evolution of cortical ensembles promoting remote memory retrieval. *Nat Neurosci*, 22(3), 460-469. doi:10.1038/s41593-018-0318-7*
*Kitamura, T., Ogawa, S. K., Roy, D. S., Okuyama, T., Morrissey, M. D., Smith, L. M., . . . Tonegawa, S. (2017). Engrams and circuits crucial for systems consolidation of a memory. *Science*, 356(6333), 73-78. doi:10.1126/science.aam6808*

18. Figure 5 and related text: How were cells with multiple place fields handled? Did you use the COM of all fields or just the field with the highest in-field event rate? Considering only the strongest field or taking the average angle of all fields for cells with multiple fields could further sharpen the distributions.

Figure 5 was replaced following the coherency analysis suggested by the reviewer. However, when we used the center out method for prediction, if a cell had multiple fields, we calculated the angle from the center to the largest field. This was added to the method. pg. 20

19. Pg 16, middle: The “possibility” referred to in this section is not clear and need to be clarified to interpret the rest of the section related to Figure 6: “This result could happen if both FS and FI cell align to geometry within context despite having different phenotypes across context. To investigate this possibility...” What is a “phenotype”? A place field location? Being an FI vs an FS cell? Or is it how that cells place field aligns/rotates on G vs C trials?

We apologize for the lack of clarity in this section. We rewrote the sentence and guided the reader to the new schematic (Figure 3A). Specifically, we wanted to discuss the possibility that FI cells may display the same rotational configuration within and across context (cells rotate along the same geometric axis), whereas FS exhibit a different rotational configuration within and across context. We modified the text as follows:

“However, it is unclear how context recognition is represented in the hippocampus to support this behavior. One possibility is that cells display geometric alignment within each context (i.e., rotating 180° along the same axis on a trial-by-trial basis), but location remapping across contexts (e.g., rotating on different axes in each context; Figure 3A, left panel). A second possibility is that one subpopulation of cells exhibits context-insensitive geometric rotations (e.g., same configuration in both contexts), while another subpopulation displays context-sensitive location remapping (Figure 3A, right panel).” pg. 9.

20. Pg. 18, bottom: “Conversely, FS cells, which respond to features (at least in identically shaped contexts defined by such features), predict context across days, and only predict heading once animals form feature/reward associations and these cells display stability within each chamber.” Are FS cells invariant in their contextual discrimination across days? e.g., are FS cells identified on day 1 also good at performing contextual discrimination (and not heading discrimination) on day 3? If they are, this would strengthen their role as truly feature selective.

While FS cells can accurately predict context on any given day, only a limited fraction of FS cells maintain their remapping characteristics over consecutive days. Consequently, it's not feasible to predict context on day 3 using the cells registered on day 1. It is possible that FS cells display dynamic ensemble properties because these cells are modulated by the formation of feature-reward associations, which changes with experience. This dynamic characteristic is also evident in memory systems, where "engram" ensembles underlying learning associations undergo changes over time, with only a small subset of neurons being consistently active during retrieval (Kitamura et al., 2017).

In contrast, FI cells appear to possess an inherent stability, which may reflect the fact that geometric properties are constant in our task.

We address these differences in pg. 12 (Results), pg. 15 (Discussion)

“The perception of geometry and context involves incidental learning independent of reinforcement (e.g., when animals are placed in an environment, they immediately perceive its shape and unique features). In our study, one example of incidental learning is exemplified by the swift alignment of FI cells to geometry, which persists even after animals increase correct responses. Moreover, the identity of FI cells is stable over time (i.e., a cell showing stability across context on day 1 continues displaying the same characteristic on day 3). Remarkably, the map orientation of FI cells established on day 1 are maintained across days and can even predict heading behavior on day 3, further indicating that the representations of geometry are consistent. *Another instance of incidental learning is the initial remapping across context of FS cells. However, unlike FI cells that are not modulated by learning, population similarity of FS cells becomes more dissimilar with experience. Additionally, FS cells display dynamic properties, incorporating new cells into the active ensemble over time, a pattern that has been observed in other associative learning tasks (DeNardo et al., 2019; Kitamura et al., 2017). These observations suggest the presence of two learning mechanisms in FS cells, one involving incidental learning and a second one involving reinforcement learning.* Notably, in addition to facilitating context discrimination, feature/reward associations also improve heading retrieval because they serve to disambiguate geometrically equivalent locations. Since firing rate in the active ensemble gradually serves to predict digging behavior and context on a trial-by-trial basis over time, our data suggest that the rate code is the neural mechanism that integrates information about features and geometry for efficient reorientation.”

21. Figure 7B, 7D: As noted above, the authors should provide individual data points overlaid on these bar graphs.

We added data points to all plots in Figure 7.

22. Figure 8A: This schematic is confusing. Why are only three points highlighted in Context A? Why only one point for the across context example? It might help to connect the same cell in context A and context B with lines. Also, rate remapping generally uses the PEAK in-field firing rate, not the mean rate (Leutgeb 2008?). What does this look like if you use peak firing rate? Last, what does this look like with calcium imaging? This is one of a few studies that directly compares results between imaging and electrophysiology, so providing Ca²⁺ results too would provide valuable information for other researchers even if the ultimate conclusion is negative (e.g., that the Ca²⁺ does not have the resolution to perform traditional rate remapping analyses).

We modified the schematic to make it clear.

We selected mean firing rate because previous studies found that when animals explored open fields using stereotyped patterns of exploration, rate changes using mean firing rate are associated with stereotyped behavior and place field directionality (Navratilova et al., 2012; Schwindel et al., 2016). Our previous research on reorientation established that disoriented animals explore the environments by sampling edges in a stereotyped manner (Normandin et al., 2022). Therefore, we wanted to use a measure that could be sensitive to this behavioral pattern. Nevertheless, we now added a comparison using peak firing rate showing that results are similar (Figure S12B). We also tested different occupancy thresholds (50%, 60%, and 70%), as requested by reviewer 3. In the paper we added the following sentences to justify the selection of mean firing rate: “Since we previously found that disoriented mice sample the environment in a stereotyped manner (i.e., sampling edges around the environment) (Normandin et al., 2022), we used the mean firing rate, which has been shown to be affected by directional changes in stereotyped exploration (Navratilova et al., 2012; Schwindel et al., 2016)” pg. 13.

Lastly, we could not get rate changes using calcium data. Calcium transients are only an indirect measure of activity. We show these results in Supplemental Figure S11.

23. Figure 8F: If I'm interpreting this correctly, combined with Keinath et al. (2017) this suggests that cells are both rotating AND rate-remapping on G trials. What is the authors interpretation of this result? Are FI cells truly insensitive to non-geometric shouldn't they retain the same firing rate regardless of if they were next to the striped wall vs the white wall in Context A, for example, since both walls are short?

The reviewer is correct. However, we never claimed that FI were completely insensitive to contextual information because the rate changes happen in all cells; rather their location specific firing is insensitive to features. It is worth noting that the geometric alignment of the cells starts immediately, whereas the rate remapping emerges over time, as animals learn to form feature/reward associations. To avoid confusion, we now clarified that the cells that we call FI have place fields displaying context-insensitive rotations, whereas FS cells have place fields that remap across context. This is one of the most interesting findings we provide because, according to the modular theory, geometric representations should be impermeable, but we find that they are not. We suggest that rate changes provide a mechanism for integrating both geometric and non-geometric information.

We clarified this point in the in the results:

“Using the context similarity cutoff value of 0.3, we divided the cells shown in the context similarity distribution (Figure 3C) and categorized them as “feature-sensitive” (FS) if similarity across context was equal or below 0.3 (i.e., cells displayed location remapping across context), or as “feature-insensitive” (FI) if similarity across context was greater than 0.3 (i.e., cells displaying stability of place fields across context). It is worth noting that this definition only refers to whether the place fields display context-sensitive rotations or not.” pg. 9.

And the discussion:

“These rate changes coincide with the ability of firing rate to predict both context and dig location, implying that rate changes integrate geometric and featural information within the entire ensemble, thereby facilitating efficient reorientation. pg.” 15

24. Figure 8E-F: Provide individual data points. Decoding accuracy increases for context and

heading retrieval after one day. Is this effect driven by FS/FI cells, respectively? In other words, do these plots look the same when broken down by FI/FS cells?

We plotted together FS and FI because the yield of tetrode data is very low. Considering that we only get 10-14% of FS per animal, we cannot do predictions per cell type. However, the same rate remapping trends were found in both cell types (Figure S13).

25. Figure S3B: Which figure is this related to?

Figure S3B was related to old Figure 3C (current Figure 3D). We updated it to show individual data points in electrophysiology and Calcium data to indicate that the recording technique did not affect the distribution of cells (Figure S5).

26. Figure S4: How does this support Figure 6? How are "stable" and "remapping" cells related to FI/FS cells? The "stable" (presumably FI?) cells appear to have a 0/180 rotation preference, but the remapping (presumably FS?) cells do not, which does not agree with Figure 6. Also, how does Figure S4 support Figure 6? There are no stats provided. Many of the "remapping" (FS?) cells appear to have no preference for 0/180.

We eliminated the old Supplemental Figure S4 (polar plots showing alignment per context) because it caused confusion. In the current version of the paper, we show the alignment within context (previously shown in the old Figure 6) in the supplement in Figures S8 and Figure S9 (boxplots embedded in violin plots to show variability of the data shown in Figure S8).

27. Methods: Which CNMF implementation was used for calcium imaging analysis? Caiman? Minian? Something else?

We used Oasis (https://github.com/zhoup/OASIS_matlab)

Reference:

Friedrich, J., Zhou, P., Paninski, I. Fast online deconvolution of calcium imaging data. Plos Computational Biology, e1005423. <https://doi.org/10.1371/journal.pcbi.1005423>

This was clarified in the method.pg. 19

28. In our hands not all injected mice obtain suitable imaging following viral infusion. How many mice were used in total for this study, including total infused with the GCaMP virus and the final number imaged (N = 5 in Figure 2...).

We injected 14 animals with GCamp6f. Among these mice, 5 were included in the study. The remaining 9 were excluded for the following reasons: 1 exhibited traveling waves (patterns of synchronization that happened at regular intervals and traveled over the field of view), 4 did not have good signal, and 4 had disrupted experimental timelines due to the Covid shutdowns. We started this project at the beginning of Covid and the shutdowns prevented us from running these animals 4 weeks after the injections. We added these exclusions in the method section. pg. 18

29. The authors indicate a minimum of 4 weeks between viral infusion and baseplate attachment. What was the mean and maximum time between viral infusion and base plate attachment? In our hands, using this high a titer of virus (> 1x10¹³ vg/mL) can frequently cause overexpression as early as 4-6 weeks post injection, resulting in aberrant calcium activity (traveling waves), especially over time. What steps, if any, were taken to ensure healthy expression of GCaMP?

The reviewer is correct about potential toxic effect of GCAMP. We have observed the toxic effects in an animal that exhibited traveling waves ~6 weeks after GCAMP injection (mouse was excluded from analysis as described in the method). The mean time for base plate attachment in the animals included in the study was 4.5 weeks (range 4-5 weeks). We visually inspected all the video recordings and excluded the animal that showed traveling waves. This animal showed atypical hubs of activity that travel through the visual field in a time scale ranging over several seconds. The mice included in the study displayed normal place fields, as illustrated by the spatial information content. We now included exclusion reasons in the method (pg.19).

Minor issues and typos

30. Pg. 4, typo "...however, lost navigator must ..."

Corrected

31. Pg. 8 second paragraph: there are no STAR issues in Nature journals.

Corrected

32. Pg 8. The N value for single units (7) listed in the text is inconsistent with that listed in the Figure 2B legend (6).

The inconsistency occurred due to the fact that one tetrode animal only yielded 1 day of data because on days 2 and 3 the recorded files were corrupted, and we could not recover the data. Therefore, we included 7 animals (day 1) but only 6 had complete data sets (days 2 and 3). We clarified this in the results.

33. Bottom of page 16. Typo – extra period between “context” and “(Figure S4)”

Corrected

34. Methods – “Subjects”: missing period between ‘cycle’ and “Animals”.

Corrected.

35. Methods – “Two context reorientation task”: Animals were tested on the last 4 trials of day 1 and the 2 following days (12 trials per day)”. This is unclear. Are they tested on all 12 trials on day 2 and 3? Or only on the last 4 trials as on day 1?

Day 1 was the first day of training. Therefore, the first 4 trials displayed a visible reward. For behavioral analysis, we only used trials in which the reward was buried between 1.5-2 cm below the surface across days, which limited the number of trials included on day 1. In a pilot experiment, we tried to give animals 20 trials on day 1 (to have more testing trials) but the mice got tired after 13-14 trials and stopped digging. We clarified the training procedure in the method. Pg. 17

Reviewer #3 (Remarks to the Author):

The data gathered for this study are sufficient to address questions about how animals learn to make distinctions between physically symmetric locations in environments, presumably by learning to incorporate features that are unique to each within-environment location. Relative to prior work, the study makes notable advances in towards answering these questions by incorporating a second environment, studying the behavior and neural data over multiple days, and utilizing multiple recording methods to reduce bias introduced with a single method and leverage the advantage of calcium imaging in recording large numbers of neurons and

registering them across days. The overall impression of these results is that they represent an important advance for the field, would be applicable to interpreting other studies, and are likely replicable (Figures 3 and 4 are especially strong). However, the precise impact of the findings presented is difficult to fully appreciate given the wide mix of analysis approaches, the large number of terms which are not strictly defined (though in some cases can be inferred), and many instances where writing is unclear. My opinion is that no further experiments need to be performed, and new analyses should be adjustments of prior effort, but some amount of rewriting and reorganization is required to convey the novelty of these findings to their full potential.

We thank the reviewer for his/her thorough and constructive criticism. We followed his/her advice adding clarifications, re-structured the paper, validated the 0.3 threshold value, analyzed coherency, and answered questions about rate. Below we addressed the concerns of each specific point in detail.

Major concerns:

1. Undefined terms – a number of these are specific enough to this study that a field expert not familiar with the authors' prior work would have to infer meaning from context

We thank the reviewer for this suggestion. In the current version of the paper, we clarified the terminology (see below).

- Modular/modularity – does this indicate a representation that is wholly replaceable, or one that is independent of others?

The concept of modularity refers to a representation that is independent and encapsulated (Fodor, 1983). It also implies that the cognitive module is innate and has evolutionary value (Gallistel, 1990).

We clarified this concept in the introduction.

“The geometric module view, on the other hand, proposes that navigators reorient based solely on a cognitive representation of environmental geometry that is impermeable to non-geometric information (Gallistel, 1990; Hermer & Spelke, 1996; Lee & Spelke, 2010a, 2010b). Supporters of this perspective propose that the brain possesses inherent, self-contained representations of spatial layouts. These encapsulated modules hold significance for survival as they depend on the consistent and stable features of the environment (Gallistel, 1990).” pg.3

- Disoriented (page 5, first paragraph of results) – critical feature of the task, define it up front

We added how we disoriented and its purpose.

“Before each trial, mice underwent a disorientation procedure to disrupt their internal sense of direction. Mice were placed in a plastic tube with a removable base, which was then placed on a rotating platform that rotated clockwise and counterclockwise in an alternating fashion (Figure 1B). Following disorientation, the tube and base were promptly moved to one of the two rectangular chambers and the mice were released by removing the base, allowing mice to enter the chamber at random facing directions” Pg.5

- Environmental geometry/layout – assume something like “the relative position of features that constrain movement, such as walls, doorways, and edges

We added the following information:

“Environmental geometry pertains to the shape of the layout defined as the traversable space delimited by the relative position of walls/boundaries” pg.3.

- Non-geometric features – sensory modes are good examples but not a strict definition

We added:

“Non-geometric featural cues encompass elements that do not define the shape of the navigable space, such as signs, objects and textures with distinct visual, olfactory, and auditory characteristics” pg. 3

- Long wall right/left

We added:

Long wall right refers to the axis where the long wall is to the right of the reward.

Long wall left refers to the axis where the long wall is to the left of the reward.

- Absolute corner location

Each axis was defined by absolute corner location: In context A, C and G cups were placed where the long wall was to the right, whereas N and F cups were placed where the long wall was to the left; therefore, the geometrically correct axis would be “long wall right”. Note the opposite is true for Context B where the geometrically correct axis would be “long wall left” Pg. 6

- Geometric equivalent

We clarified:

geometric equivalent location sharing the same geometric properties as C, pg.5

- Rotational equivalent

This term was replaced with geometric equivalent.

“Reorientation behavior

We defined reorientation in the abstract:

Reorientation, the process of regaining one’s bearings after becoming lost . pg. 2

- Heading [behavior]

In the abstract we replaced heading with facing direction to clarify the meaning of heading:

“Recovery of facing direction within that context (heading retrieval)”

- Behavioral strategies (page 7 paragraph 3)

We changed the sentence:

To confirm the shift in reorientation strategies (i.e., the incorporation of features to recover heading),

2. The order of some results seems motivated by the “class” of analysis rather than theoretical chain of thought. For example, Figure 1 establishes that animals are a bit confused by the symmetry of the environments but seem to understand that the environments are different, figure 2 shows that place cells exhibit a similar type of confusion, it would be natural here to ask how tight that relationship is on a per-trial basis, which isn’t addressed until Figure 7. This ordering makes it challenging to follow the authors’ intuition for the progression of results. The stated motivations for figure 3 are challenging to understand with respect to the specific analyses that follow. Figure 3 is described as being motivated to find how neurons encode “featural information,” but, agnostic to this specific experiment, one would more broadly ask about the relationship of place fields across contexts, a long-studied phenomenon. There is a strong a priori constraint provided on this question by the finding in figure 2 that ~60% of neurons remap between trials according to the 2-fold symmetry of the environment; this seems only tacitly acknowledged by the language at the top of page 10. Additionally, it has not been established by this point in the text that there even is remapping between contexts under these experimental conditions, and it is suggested that there may not be due to the equivalent shapes

of the boxes. The hypotheses offered in the middle of the first paragraph are contingent on these and other assumptions and analyses that have not been fully developed at this point in the text. This links to the point above that the connection between hippocampal activity and behavior that it is presumed to support has not yet been established.

We thank the reviewer for suggesting restructuring the order of our analyses. We left Figures 1 (behavior) and 2 (geometric alignment) as before because we first had to show that cells align to geometry to explain our method to detect remapping across context. However, following the description of geometric alignment, we restructured the order of the paper following the reviewer's suggestions. We believe this order is much more intuitive (Thank you!!)

General organization:

- 1. Behavior*
- 2. Neural substrate of heading retrieval: Geometric alignment*
- 3. Heading predictions*
- 4. Neural substrate of Context recognition: Do cells remap across context?*
- 5. Identification of FI and FS cells*
- 6. Demonstration that context-sensitive FS predict context across days*
- 7. Evaluation of coherence in FS and FI*
- 8. Evaluation of cell identity across days*
- 9. Demonstration that rate changes serve to integrate featural and contextual information in the active ensemble.*

3. Requested analysis: While the 0.3 correlation score was previously established by another paper, it is not clear from the distribution shown in Figure 3B that this indicates a fundamental functional divergence, and the analyses that follow could (cynically) be interpreted as post-hoc justification for that criterion. Additionally, the findings presented in 3E are redundant with this definition (also, how can the curves for FS and FI across overlap when they are defined by the criterion on the x-axis?). I recommend that the authors further subdivide neurons by their context similarity scores (e.g. into quartiles or even deciles, since only 11% of cells are below 0.3), or make scatterplots showing the relationship of (e.g.) each cell's correlation within environments as a function of its correlation across environments. If the quartile+ method were employed for Figure 4B, this would help us understand whether there is a continuum of population similarities or bimodal (or more) clustering.

In the current version of the manuscript, we validated the remapping threshold following reviewers' 2 and 3 suggestions (see below). We also agree with the reviewer 3 that the cumulative curves shown in Figure 3E were somewhat redundant. Therefore, we eliminated them.

This point was raised by reviewers 2 and 3 so we generated a common response following the suggestions of both reviewers.

We thank reviewers 2 and 3 for making suggestions to strengthen the selection of the remapping threshold. To validate this threshold, we first used reviewer's 3 suggestion of plotting the within and across similarity scores by dividing the distribution of average similarity correlations in deciles (Figure S4A). Then, we calculated the correlation of the within and across similarity scores for each decile and used a modeling approach to analyze the function (Figure S4B). Decile 1, comprising correlations between 0 and 0.3, emerged as the most informative because it coincided with the function's half-life and root. Second, following the suggestion of reviewer 2, we also plotted context prediction as a function of remapping threshold. Below is our detailed response to this critical point.

Decile analysis and modeling.

Figure S4. Complement of Figure 3. Validation of remapping threshold. A. Scatterplots showing average correlations within and across context for individual cells separated in deciles obtained from the similarity distribution shown in Figure 3C. B. Asymptotic regression model of the overall correlation decile function. The red dot indicates the half-life of the function, which coincides with decile 1 (correlations across context between 0 and 0.3) and the root of the function (value that makes the function 0 on the y axis). Finally, in the asymptotic regression model, the relative growth rate is not constant. It attains its peak when $Y = 0$ and diminishes as Y increases. This suggests that Decile 1, corresponding to $Y = 0$, represents the point at which the rate of change is maximized. This indicates that the first decile is the most informative to discriminate across context.

Description of modeling approach

To select the best correlation threshold to characterize similarity scores, we first separated the cells into deciles using context similarity distribution shown in Figure 3B. For each cell, we plotted the average within and across context correlation (average correlations included all pairwise comparisons corresponding to each cell) and generated scatterplots for each decile (Figure S4A). We then calculated a Pearson's correlation coefficient between the average within and average across correlations for the associated cells per decile, which yielded a set of 10 values (one per decile). We noted that the largest difference occurred between the first and second decile. To understand this change, we used a modeling approach to fit the data (Figure S4B).

To identify the optimal non-linear model, a two-stage goodness-of-fit analysis was conducted. First, significant points (Roots, Extremes, and Inflections) were pinpointed in the correlation decile function shown in Figure S5B using the Taylor Regression algorithm (Christopoulos, 2014). Root values denote the X-axis values leading to null Y-axis values, while inflection points, maxima, and minima were determined through derivatives of the function. Second, the best model was selected from an exhaustive pool of alternatives based on the lowest values for Akaike's information criterion (AIC), Residual Standard Error (RSE), and the highest value for log likelihood (L-L) a set of metrics gauging goodness of fit (Ritz, et al., 2015). Notably, Sigmoidal (based on inflection points) and type Maxima/Minima models were excluded due to their inability to yield a satisfactory fit for the data.

This led us to focus on concave-convex or exponential-type models, from which the Asymptotic Regression model (Pinheiro and Bates, 2000) exhibited the best results in terms of goodness of fit [AIC = -27.75, RSE = .0484, L-L= 17.88, see below for details].

Asymptotic Regression model: $Correlation = q_1 + (q_2 - q_1) \cdot \exp(-\exp(q_3) \cdot Decile)$.
In this model, q_1 : Asymptote, q_2 : Origin, and q_3 : Rate (on a logarithmic scale).

The term "Origin" refers to the initial correlation value, while the "Asymptote" signifies the point at which correlations stabilize. The "Rate" parameter captures the speed of the correlation function change from the origin to the asymptote, computed as the natural logarithm of the rate constant. The estimated values for the three parameters were all statistically significant (Asymptote = 0.83, Origin = -0.78, LogRate = -0.35; $p < .05$). We then estimated the half-life [$X_{0.5} = \log_2 / Rate = 0.99 \approx 1$, see Figure S4B], which identified the correlation value that reached half of the function's rate. The concept of half-life, a parameter commonly used to characterize exponential changes, is detailed in Pinheiro and Bates (2000). In exponential models, the half-life plays a role analogous to the inflection point in sigmoidal (logarithmic) models. In the Correlation-Decile function, Decile 1, comprising correlation values between 0 and 0.3, corresponds to the function's half-life. Additionally, Decile 1 also corresponds to the root of the polynomial function, which is the value that turns the function to zero. Finally, the relative growth rate in the asymptotic regression model (first derivative definition is shown on Figure S4B) is not constant, reaching its maximum value when $Y = 0$ and decreasing as Y increases. This suggests that Decile 1, corresponding to $Y = 0$, represents the point at which the rate of change is maximized. These results suggest that the optimal correlation threshold value to separate cells based on remapping properties is 0.3

Table S4

Parameter	Estimate	sd	t	Pr(> t)
Asymptote	0.8300	0.02	34.84	0.0000***
Origin	-0.7800	0.21	-3.65	0.0082**
Rate (Log)	-0.3500	0.15	-2.37	0.0497*

Note: *** $p \leq .001$, ** $p \leq .01$, * $p \leq .05$

RSE = 0.0484, log likelihood = 17.88, Akaike's information criterion = -27.75

References

Christopoulos, D.T. (2014). *Roots, extrema, and inflection points by using a proper Taylor regression procedure*. SSRN. <https://dx.doi.org/10.2139/ssrn.2521403>
 Pinheiro, J.C. and Bates, D.M. (2000) *Mixed-effects models in S and Splus*. Springer.
 Ritz, C., Baty, F., Streibig, J. C., and Gerhard, D. (2015) *Dose-Response Analysis Using R*. PLOS ONE, 10(12), e0146021

2) Context prediction as a function of remapping threshold

As requested by reviewer 2, we also plotted context prediction as a function of the remapping threshold. Note that low remapping thresholds are best at predicting context; however, the prediction plateaus between 0.2-0.3 thresholds. At values higher than 0.3, there is a strong decay in the context prediction accuracy. We believe that the modeling of the decile correlation function strengthens our selection of the threshold value 0.3. Therefore, we only included the graphs shown below in this letter.

Context prediction as a function of remapping threshold for, tetrode, calcium, and combined data. As expected, context prediction decreases as thresholds values decrease. However, there is a plateau between 0.2-0.3 correlation values, and a substantial decline at higher correlation values.

4. Rate remapping, context coverage in general, specifically on page 20: Authors need to detail both how much of the environment the animals explored on each trial (it seems from the methods that animals were pulled out after a correct dig) and how trial average firing rates were computed. Here's the issue: if a majority of place cells have similar rotation-aligned place fields across trials and contexts, and animal behavior is divergent across contexts due to the different locations of C and G wells and given high performance, it follows that the animals would sample

different portions of a given cell's place field across contexts. (Example: if a cell has a place field in the upper left corner of both boxes, and in box A the animal runs from the middle to the upper right he will barely skirt that place field while in box B the mouse runs from the middle to the upper left that cell will go nuts). This would result in differences in firing rates not strictly attributable to differences from cells' incorporating feature information but instead from uneven sampling. The authors need to expand on how firing rates were compared across trials: whether these maps were restricted only to comparable bins visited in both contexts on a pair of trials, or if it was the average firing rate of a cell (while it was firing?) for that trial independent of animal's position, or some other method. This is relevant to most analyses, but especially to Figure 8.

Thank you for raising this important point. The reorientation task requires short trials because prolonged exposure to features under oriented conditions influences reorientation strategies (Gray et al., 2005; Dudchenko et al., 1997). However, animals sampled a significant portion of the environment on each trial. We increased the motivation to explore by food depriving the mice before and during the experiment.

To thoroughly evaluate the possibility raised by the reviewer, we also calculated area of the environment and cups visited (Figure S12A and B). Except for a few trials, animals covered at least 60% of the arena across trials, and more than 70% of the cup areas. To further address the reviewer's concern, in the current version of the paper we plotted rate remapping as a function of occupancy using either mean or peak firing rate finding similar results (Figure S12C and D)

The mean firing rates were calculated by creating a spike map and a dwell map and dividing the spike activity by the dwell activity.

Finally, we also want to point out that we exhaustively examined the reorientation behavior of animals before they make a choice (Keinath et al., 2017; Normandin et al., 2022). We found that animals always walk around the enclosure, sampling edges and angles before making a choice. Furthermore, we previously demonstrated that this sampling of walls is essential for the use of geometry (Normandin et al., 2022). Therefore, although the possibility raised by the reviewer is very logical, "if a cell has a place field in the upper left corner of both boxes, and in box A the animal runs from the middle to the upper right he will barely skirt that place field while in box B the mouse runs from the middle to the upper left", our data show that mice normally do not go directly to a cup after being placed in the middle of the environment, but rather sample around all the borders before making a choice.

5. Requested analysis: Figure 5C, legend: it is claimed that the increasing circularity of these distributions indicates that "each FS cell remaps independently from each other;" however, these data are pooled across trials, and so it is possible that FS cells could remap coherently within a trial but circularly across trials. To demonstrate independent remapping, circularity needs to be shown on a single trial basis. Please indicate how uniform rotation remapping is on a single trial basis.

We are very grateful to reviewers 2 and 3 for their suggestions regarding this point. Reviewer 2 indicated that we needed to disentangle whether the remapping across context was coherent or global and provided a schematic that we now incorporated in the current version of the manuscript, and reviewer 3 suggested that our center out analysis could have occluded coherency because we were averaging across trials and animals. We conducted exhaustive analyses to test how the cells were rotating across context and concluded that both reviewers were correct: There was coherency in both FS and FI subpopulations. Below we describe the new analyses and how we changed the interpretation of the results.

To determine if there was coherency in the population, we assessed the proportion of best match rotation (BMR) angles (0°, 90°, 180°, and 270°) across trial pairs in FS and FI cells

within and across context and averaged the data per animal (Figure 5B shows schematic of method). This analysis yielded 4 proportions of cells rotating together between trials (based on the 4 angles used for the analysis), which are shown in Figure 5C-D. Coherent remapping in the population would be indicated by a predominant proportion of cells sharing the same BMR angle between trial pairs). To determine significance, the coherent proportions of cells were compared with chance assuming equal distribution of angles (25%).

Using this analysis, we found that both FS and FI cells displayed coherency. Both cell types show a dominant coherency (1st BMR) as well as a second one (2nd BMR) that were above chance level (although the significance of the secondary coherency varied with cell type and day of training). Interestingly, the 1st and 2nd BMR displayed consistent levels that were above chance across days in FI cells; however, the 1st coherency increased, while the 2nd coherency decreased across days in FS cells (Figure 5E). This indicated that both cell types displayed coherent patterns of rotation within and across context, but the coherency in FI cell is constant, whereas the coherency in FS is modulated by learning.

To examine the angular relationship between the 1st and 2nd BMR, we plotted the joint distribution of their rotations (Figure 5F). We found that in FI cells the 1st and 2nd BMR maintained a geometric relationship (e.g., if cells displayed a 180° angular rotation in the 1st BMR, the 2nd BMR would remain stable at 0°). This result further supports the involvement of FI cells in representations of geometry. However, in FS cells there was no clear relationship between the 1st and 2nd BMR. Interestingly, the 1st BMR marginal probability indicated that the 1st coherency in FS cells aligns to opposite axes across context, a pattern that could serve to facilitate context recognition.

Minor concerns:

6. Strong language assuming necessity of findings for behaviors and the cognitive processes underlying each behavior could be tempered – these findings are correlations, some with notable variability, and the absence of behavioral manipulations or comparisons across brain regions means these findings are less definitive than presented
We corrected the language to reflect that our findings show correlations.

7. This sentence from the abstract is not supported by the results: “Efficient heading retrieval and context recognition require integration of featural and geometric information in the active network through rate changes.” The data show that rate changes are correlated with improved heading retrieval, but requirement is not demonstrated.
We changed the wording of the abstract: “Efficient heading retrieval and context recognition correlate with rate changes reflecting integration of featural and geometric information in the active ensemble.”

8. Page 25, paragraph 1: “CA1 is AN ideal substrate...”
This sentence was eliminated

Figures need additional labeling for clarity

8. Figure 1 can almost be understood entirely without the text. Additional labelling is needed for panel E to explain that this is not merely another method of quantifying D but accounts for individual variability.

We added the following information to the old Panel D (new panel G to clarify that BF provide information about individual performance. “Cumulative proportion of individual Bayes Factors

(BF) across days 1 to 3 combining contexts evaluating the alternative model (MAlt) that animals increased digging in C locations with experience vs. the null model (Mnull) that animals dug by chance.”

9. Figure S2 polar plots need units for radial distance

We added units to Figure S2

10. Findings related to polar measures (angle differences) need to more specifically say where the peaks are with appropriate decimal error when relevant (e.g. 5B not exactly at 0/180)
Mean peaks +/- errors

This information was added to figure S2.

11. 2A legend needs to say that the dotted line and arrow indicates compression for analysis

Clarification added.

12. Figure 2B,C: Heading over the left bar plots should say “electrophysiology” to reduce confusion with these results’ being about the proportion of single cells best aligned at each rotation

Correction added.

13. Figure 3E, 4B,D,E need some kind of label indicating which comparisons are significantly different

We added significance bars to the population vector curves; other cumulative curves were omitted to comply with reviewers’ suggestions.

14. Figure 8A needs to say somewhere (legend, perhaps) that each circle is the average firing rate for a single trial

Old Panel 8A (new Panel 7A) was redesigned for clarity.

15. Figure 8C: it seems that these are the averages of rate differences, not just the average rates as stated in the legend (add a label to the color scale as well)

This typo was corrected.

16. Close copyedit for typos, grammar, vague references. Examples:

- Page 4, paragraph 2: “This pattern OF results”

- Page 4, paragraph 2: “moreover, IT/THIS PATTERN suggests” – number agreement of “it” with “this pattern”

- Page 25, paragraph 1: “ideal substrate to integrate THESE cognitive processes”

All these typos were corrected, and paper was edited to minimize any additional errors.

17. Methods should detail whether any distal cues are present in the room and any efforts made to reduce other stimuli such as noise from computers, vents, etc.

We had this information in the method, but it was buried in the behavioral protocol. We now moved these sentences to bottom of the behavioral protocol description. Briefly, we rotated the chamber every trial 90 degrees, used a black curtain and added white noise to prevent the use of external cues/noises to guide reorientation. Lastly, the chambers were thoroughly cleaned with ethanol after each trial and the scented bedding was replaced in each cup.

“To avoid the use of distal room cues, the experimental chambers were rotated 90° after each trial and white noise was delivered from a centrally overhead speaker. To avoid the use of odor trails,

the chambers were thoroughly cleaned with ethanol and all the cups were refilled with clean scented bedding during the inter-trial interval (3-5 min). “ pg. 17

18. For many comparisons, it is quickly lost and often not mentioned in the figure whether it is within or across contexts. “Context similarity” is at different times used to refer to either
We made sure it was clear in all captions or the results whether comparisons were within or across context.

19. Circular definitions and results, especially like this example:
- Page 16, paragraph 2: “FS cells displayed feature sensitive remapping,” this is guaranteed since this is how FS cells were defined.
We edited the paper to avoid redundancy and circular definitions.

20. Figure 6C/D: legend states that “0 degree best match rotations also occurred more often than 180 degree in FS cells in both contexts,” implying this was not the case for FI cells, but the significance bars on both plots are identical.
We apologize for this typo, the significance on the Figures was correct. The typo was in the legend and was corrected. Please note that now the old Figure 6 is in the Supplement (Figure S8).

21. The text supporting Figure 7 overcomplicates the question of how often does the alignment of the maps associate with the animal’s behavioral decisions, which is a separate question from how reliably the population activity is self-similar within context and distinct across contexts on a single trial basis. Two possible solutions: first, reorganize the text to make these points more evident. Second, choose a different approach to these analyses which is more intuitive to a reader, reducing the burden of understanding each separate analysis approach; this should use the same method for both analyses, such as the SVM from 7a or max population correlation with average map, as in 7c, or a widely used clustering method indicating how often a trial ends up in the wrong cluster.
We have reorganized the information of the paper, and we use a different prediction method to determine heading. Specifically, we use the center-out angle, which follows the description of the best match rotations. Since we explain the center-out angle to corroborate the pixel to pixel cross-correlation method in the previous paragraph, the prediction flows better and will be more intuitive to the reader. Pg. 8

22. The authors should offer some account for what the cells that rotate 90 or 270 are doing. In those cases, are the best aligned maps perhaps lower correlations than best aligned maps from 0 or 180? Are those proportions similar to other non-task related remapping seen in other studies?

This point was also raised by reviewer 2. Therefore, we provide a common answer.

To determine what happened during 90° and 270° rotations, we first tried to predict errors using the center out measure. This measure evaluated the angle from the center of the arena to the center of mass of the place field. We hypothesized that if 90 and 270 rotations were associated with errors, we would be able to predict behavior on these trials. To this end, we separated all the error trials where animals dug in Near or Far locations and trained a support vector machine classifier, withholding the trial to be classified. Error prediction was not significant on any day (Mean predictions for error trials: Day 1: 0.65 ± 0.16 , $p=0.16$, Day 2: 0.50 ± 0.14 , $p=0.49$; Day 3: 0.58 ± 0.16 , $p > .05$).

To determine what happened with 90 and 270 rotations, we then analyzed the sequence of digs from Trial A to Trial B as a function of rotation angle. In other words, for each angular rotation between trials (0, 90, 180, and 270), we plotted the behavioral sequences observed. The results showed that C/G and C/C were the most likely dig sequences. Notably, 90° and 270° rotations happened sporadically, even when animals made C/C or C/G response sequences (we corroborated this result with a variety of methods, and all converged to the same conclusion). Therefore, we can only conclude that the alignment is not perfect and there is some noise in the neural data.

In our previous study we observed similar rotations during reorientation in the traditional rectangular one context task (~15% 90° and 270° rotations; Keinath et al., 2017). Interestingly, we also observed some random rotations in an isosceles triangle, an environment without rotational symmetry [72% stability (0° rotation), with ~15% rotations coinciding with either 120° or 240°, suggesting that random rotations happen even in the absence of rotational symmetry]. These sporadic rotations may reflect the inherent noise in neural systems. These data are now added in Supplemental Figure S4. We also added a sentence to the results: “we plotted the counts that corresponded to each pair of behavioral responses for each rotation angle (Figure S4). We found that correct behavioral sequences (C/G or G/C) were prevalent at all angular rotations, including 90° and 270°, suggesting that non-geometric rotations happen sporadically and may reflect noise at the neural level.” pg. 8.

Figure S3. Proportion of sequence of digs per trial as a function of place field angular rotation (0°, 90°, 180°, 270°). Note that C/C and C/G are the most prevalent dig sequences, which is illustrated in vivid yellow/green colors. 90° and 270° rotations happen sporadically, even during C/C G/C or C/G sequences, which may reflect inherent noise in neural data. Colors indicate response counts, with yellow reflecting higher counts and dark blue lower ones.

Other:

23. Kinsky et al 2018 also showed “random” remapping across visits to the same context but coherent rotations of place fields aligned to environmental geometry for each remapping event: are the proportions of remapped cells similar as to here? Can differences be attributed to reward motivated behavior in this experiment?

Kinsky et al. showed that the average percentage of coherent cells within a session was ~45% (Kinsky et al., 2018, Figure 1I). Our data show a mean coherency ~40% for FI cells and ~60% for FS cells. The high coherency of FS cells is likely attributed to the fact that we use a goal-oriented task and rewards. The secondary coherency, which is significantly different from the control distribution in FI cells across days, is likely since disorientation drives more attention to geometry than free foraging.

24. From descriptions in the intro, it’s hard to tell the difference is between the adaptive cue combination theory vs. associative learning model: is the idea that adaptive cue combination works with a distribution of probability over the whole set of cues, and the associative learning model works with cues independently of each other? Also, unsure who in the sentence is doing the “adjusting” of their salience.

Thank you for pointing out that we were not clear differentiating these theories. The reviewer is correct on the assumption that the adaptive cue combination theory relies on probabilistic inference based on the relative salience/reliability of all available cues, whereas the associational theory evaluates how distinct cues compete by gaining or losing strength based on a modified version of the Rescorla and Wagner model. Consequently, these theories also differ in terms of the weight given to learning. Learning is necessary for the associational theory but not for the adaptive cue combination one.

We modified the sentences in the introduction as follows:

“These ideas were developed in two more recent theories: 1) The adaptive cue combination theory, which proposes that navigators use probabilistic inference to evaluate all available cues based on their salience and reliability (Newcombe & Huttenlocher, 2006; Xu et al., 2017) and 2) The associative learning model, which suggests that available cues compete with each other by gaining or losing associative strength through learning (Miller, 2009; Miller & Shettleworth, 2007). According to these theories, subjects reorient relying on only one process that uses geometry, features, or a combination of both depending on which cue or combination is the most informative”. Pg. 4

25. Requested analysis: How often is the second dig in the correct location? Greater chance of this following G digs than other? On the flip side, how often when the animals dig in a wrong N or F well do they then go to the C well vs. G?

To address this question, we generated a matrix that shows the likelihood of digging in any cup after an error in Near, Geo, or Far. We eliminated correct as first dig because, in most cases when animals dug in the correct cup first, there was no second dig. These results are Figure S1

Figure S1. Complement of Figure 1. Digging counts following errors in Near (N), Geo (G), or Far (F). Following errors in G, N, or F the likelihood that the next dig was Correct (C) was higher than digging in error locations. This is evident in the higher counts observed in column C, depicted in darker blue colors. Following G digs, there was a moderate probability of errors in N or F. Other error combinations were less likely. These data indicate that when animals made errors, they self-corrected going to the C location in most cases. Correct first choices are not shown because following these digs animals stayed in the C cup until the reward was found.

26. Page 14, paragraph 1: in this situation “discrimination” is a less ambiguous term than recognition to describe decreasing similarity of activity across contexts. In this same paragraph, be cautious about the use of “remapping,” since it is sometimes used to refer to geometric rotation of place cells over trials in the same context and at other to refer to location remapped place fields across contexts.

We used the word Recognition to link this study to the terminology of our previous paper (Julian et al.); however, we now clarify in parenthesis that this refers to context discrimination.

“In a previous behavioral study, we used a novel two-context reorientation paradigm to show that disoriented mice use geometry for heading retrieval, while simultaneously using non-geometric features for context recognition (Julian et al., 2015).” Pg. 4

Additionally, we clarified the use of the word “remapping” as follows: “Note that we use the word remapping only to describe the shift in the preferred firing location across context, we refer to geometric alignment to describe the rotations that occur within context. Pg. 9”

27. Page 20 Paragraph 1: mixing the current study’s findings in with a description of findings from another study makes it hard to keep track of what’s being explained.

Thank you for the suggestion. We left the references but omitted the discussion of Shin et al. in the results.

Reviewer #4 (Remarks to the Author):

This report probes the cognitive and neural processes that underlie navigation and hippocampal function in mice, in a behavioral situation that requires the animals to distinguish between two task contexts—signaled by distinctive markings on the walls of the chamber in which the task is given—and to reorient themselves after slow turning in that environment. It combines three general methods—behavior, electrophysiological recordings of individual place cells in the hippocampus, and calcium imaging to allow re-identification of cells across time. Analyses of the three methods together make a powerful case for two separate neural systems in the hippocampus, one focused on the enduring shape of the environment (the locations, distances and directions of the bounding walls of the chamber) and the other focused on the transient changes in task demands that are signaled by visual cues in the chambers (patterns on the walls that differ from one another but that do not change the shape of the chamber itself). Prior to the experiment proper, hungry rats (below normal body weight) learned that buried, consumable food would be provided in the room at different locations in each context (for example, either directly left of a visual cue in one room or directly right of a different visual cue in another room). On 3 days, they were then tested in a rectangular chamber with one of the sets of context cues in a state of disorientation. To find the food, they needed (a) to determine which task context they were in, and (b) reorient themselves within that context and dig for the food. The location of an animal’s first dig provided the behavioral measure of their reorientation, but if I’m reading their methods right, the animals were allowed to continue searching in case of failure until they found and consumed the food: thus, if their first dig was wrong, they had an opportunity to explore the room further. Consistent with much past research (most beautifully, with the 2015 Julian et al paper), the authors found that reorientation on day 1 was based on the shape of the room, leading to high and approximately equal search at the correct location and at a location that was 180-degrees rotated away from it, providing evidence both for context recognition and for geometry-based reorientation. Over the next 2 days, however, first searches at the correct corner exceeded those at any other corner, suggesting that reorientation itself began to be influenced by the task context. The heart of the paper uses electrophysiology and

calcium imaging to probe how the mice accomplished this feat.

Caveat: I am not an expert in the latter methods and hope other reviewers will give them a closer evaluation than I can provide. But the authors describe them clearly enough for reader like me to follow them. They distinguish between two kinds of cells in the hippocampus that are active and predict the animal's behavior in these experiments: a larger population of cells, dubbed FI, respond to the chamber's geometry, independently of the task context: across trials, they fire most at either the same location or at a location that is 180 degrees rotated from it, consistent with a process of reorienting by the chamber's rectangular shape and without regard to the pattern that signals the food's location in the current task context. A smaller proportion of cells, dubbed FS, change their firing depending on the task context. Interestingly, FI cell firing predicts reorientation behavior on all 3 days and never predicts behavior in relation to the context, whereas FS cell firing predicts context behavior on all 3 days and predicts reorientation on day 3, suggesting a learning effect. Further analyses show that correctly digging at the location of the food—which requires binding of information for the task context and the animal's heading—is predicted by the firing rates of the hippocampal neurons that are active on that trial, on day 3 but not earlier.

These findings are beginning to solve what has always struck me as a great mystery. In many studies of navigation (including in humans, for example in the cited papers led by Doeller), the hippocampus appears automatically to track a navigator's position and heading in relation to the borders of the navigable environment—and not in relation to movable objects or transient visual cues. Yet in just as many or more studies of memory, the hippocampus is found to be critically involved in episodic memory: a capacity that requires that it be responsive to all sorts of transient events. I think this paper begins to suggest how to reconcile these two sets of findings. As we all were reminded during the pandemic-imposed lockdown, episodic memory gets really bad if the transient events that we want to remember all happened in the same location. Such memories may require that transient events be recorded in the context of a stable and enduring representation of places in the navigable layout. I think this paper is beginning to suggest how the hippocampus accomplishes this task, and I find it highly worthy of publication in *Nature Communications*.

Elizabeth Spelke

We thank the reviewer for her positive appraisal of our paper. It means a lot to us considering her pioneering work in the field of reorientation.

REFERENCES

- Brown, A. A., Spetch, M. L., & Hurd, P. L. (2007). Growing in circles: rearing environment alters spatial navigation in fish. *Psychol Sci*, 18(7), 569-573. doi:10.1111/j.1467-9280.2007.01941.x
- Christopoulos, D. T. (2014). Roots, Extrema and Inflection Points by using Taylor Regression Procedure. SSRN. Retrieved from <https://ssrn.com/abstract=2521403>
- DeNardo, L. A., Liu, C. D., Allen, W. E., Adams, E. L., Friedmann, D., Fu, L., . . . Luo, L. (2019). Temporal evolution of cortical ensembles promoting remote memory retrieval. *Nat Neurosci*, 22(3), 460-469. doi:10.1038/s41593-018-0318-7
- Dudchenko, P. A., Goodridge, J. P., & Taube, J. S. (1997). The effects of disorientation on visual landmark control of head direction cell orientation. *Exp Brain Res*, 115(2), 375-380. doi:10.1007/pl00005707
- Fodor, J. A. (1983). *The Modularity of Mind*: The MIT Press.

- Gallistel, C. R. (1990). Representations in animal cognition: an introduction. *Cognition*, 37(1-2), 1-22. doi:10.1016/0010-0277(90)90016-d
- Gray, E. R., Bloomfield, L. L., Ferrey, A., Spetch, M. L., & Sturdy, C. B. (2005). Spatial encoding in mountain chickadees: features overshadow geometry. *Biol Lett*, 1(3), 314-317. doi:10.1098/rsbl.2005.0347
- Hermer, L., & Spelke, E. (1996). Modularity and development: the case of spatial reorientation. *Cognition*, 61(3), 195-232. doi:10.1016/s0010-0277(96)00714-7
- Keinath, A. T., Julian, J. B., Epstein, R. A., & Muzzio, I. A. (2017). Environmental Geometry Aligns the Hippocampal Map during Spatial Reorientation. *Curr Biol*, 27(3), 309-317. doi:10.1016/j.cub.2016.11.046
- Kinsky, N. R., Sullivan, D. W., Mau, W., Hasselmo, M. E., & Eichenbaum, H. B. (2018). Hippocampal Place Fields Maintain a Coherent and Flexible Map across Long Timescales. *Curr Biol*, 28(22), 3578-3588 e3576. doi:10.1016/j.cub.2018.09.037
- Kitamura, T., Ogawa, S. K., Roy, D. S., Okuyama, T., Morrissey, M. D., Smith, L. M., . . . Tonegawa, S. (2017). Engrams and circuits crucial for systems consolidation of a memory. *Science*, 356(6333), 73-78. doi:10.1126/science.aam6808
- Lee, S. A., & Spelke, E. S. (2010a). A modular geometric mechanism for reorientation in children. *Cogn Psychol*, 61(2), 152-176. doi:10.1016/j.cogpsych.2010.04.002
- Lee, S. A., & Spelke, E. S. (2010b). Two systems of spatial representation underlying navigation. *Exp Brain Res*, 206(2), 179-188. doi:10.1007/s00221-010-2349-5
- Miller, N. (2009). Modeling the effects of enclosure size on geometry learning. *Behav Processes*, 80(3), 306-313. doi:10.1016/j.beproc.2008.12.011
- Miller, N. Y., & Shettleworth, S. J. (2007). Learning about environmental geometry: An associative model. *Journal of Experimental Psychology-Animal Behavior Processes*, 33(3), 191-212. doi:10.1037/0097-7403.33.3.191
- Navratilova, Z., Hoang, L. T., Schwindel, C. D., Tatsuno, M., & McNaughton, B. L. (2012). Experience-dependent firing rate remapping generates directional selectivity in hippocampal place cells. *Front Neural Circuits*, 6, 6. doi:10.3389/fncir.2012.00006
- Newcombe, N. S., & Huttenlocher, J. (2006). Development of Spatial Cognition. In *Handbook of child psychology: Cognition, perception, and language, Vol. 2, 6th ed.* (pp. 734-776). Hoboken, NJ, US: John Wiley & Sons Inc.
- Normandin, M. E., Garza, M. C., Ramos-Alvarez, M. M., Julian, J. B., Eresanara, T., Punjaala, N., . . . Muzzio, I. A. (2022). Navigable Space and Traversable Edges Differentially Influence Reorientation in Sighted and Blind Mice. *Psychol Sci*, 33(6), 925-947. doi:10.1177/09567976211055373
- Pinheiro, J. C., & Bates, D. M. (2000). *Mixed-effects models in S and S-PLUS*. New York: Springer.
- Ritz, C., Baty, F., Streibig, J. C., & Gerhard, D. (2015). Dose-Response Analysis Using R. *PLoS One*, 10(12), e0146021. doi:10.1371/journal.pone.0146021
- Schwindel, C. D., Navratilova, Z., Ali, K., Tatsuno, M., & McNaughton, B. L. (2016). Reactivation of Rate Remapping in CA3. *J Neurosci*, 36(36), 9342-9350. doi:10.1523/JNEUROSCI.1678-15.2016
- Twyman, A. D., Newcombe, N. S., & Gould, T. J. (2013). Malleability in the development of spatial reorientation. *Developmental Psychobiology*, 55(3), 243-255. doi:10.1002/dev.21017
- Xu, Y., Regier, T., & Newcombe, N. S. (2017). An adaptive cue combination model of human spatial reorientation. *Cognition*, 163, 56-66. doi:10.1016/j.cognition.2017.02.016

REVIEWER COMMENTS

Reviewer #1 (Remarks to the Author):

The authors have addressed my comments satisfactorily.

Reviewer #2 (Remarks to the Author):

In this study, Gagliardi investigated neural mechanisms that could support two important features of spatial orientation: heading retrieval and context identification. They combine a unique context-guided spatial orientation task with electrophysiological and calcium imaging recordings in freely moving mice to uncover two types of neurons in the hippocampus: Feature Insensitive (FI) cells, which encode the geometry of an environment but do not distinguish between two separate contexts with the same geometry, and Feature Sensitive (FS) cells, whose firing is modulated by features that differ between two identically shaped arenas. They exhaustively characterize the similarities and differences between FI and FS cells, showing that FI cell coding remains relatively stable across days while FS cell coding changes with learning. These findings are of interest not only to the field of spatial navigation but will also appeal to a broader audience by providing a general mechanism for how the hippocampus binds spatial and non-spatial information to support memory formation and recall.

The revised manuscript is clear and comprehensive. The addition of data points to plots demonstrates a clear effect while simultaneously representing across-animal variability, which, combined with the addition of many methodological details, will greatly aid other researchers in understanding, reproducing, and expanding these findings. The revised manuscript addresses all of my major concerns, and the data provided supports the authors' conclusions. My only remaining issue relates to how rate changes in FS vs. FI cells over time could code for heading direction or context retrieval. The specifics of this issue are listed immediately following this paragraph, followed by a few minor/specific comments.

Remaining issue/question

Figure 7 – are the rate-remapping SVM classifier results driven differentially by FI vs. FS cells? In other words, do you get the same results if you train and test the classifier with one cell type only? This is related to point 23 from the first review and will be interesting regardless of the outcome. If

FS cells drive this result and an FI-only SVM was not able to decode heading or context based on rate alone, that would further strengthen the author's point in the discussion that FS cells undergo experience-dependent modulation while FI cells do not. If, alternatively, FI cells could classify heading, this would suggest that FI cells could be modulated by learning and could incorporate non-geometric information by adjusting their firing rate.

Minor comments

Page 8 “Therefore, we plotted the counts that corresponded to each pair of behavioral responses for each best match rotation angle (Figure S3). We found that correct behavioral sequences (C/G or G/C) were prevalent at all best match rotations...”. What is the “pair of behavioral responses?” From the figure legend it seems like this is the first and second dig location in a given trial. Adding labels to the confusion matrices axes would help clarify this.

Page 9 “BMR analysis” is used but this acronym has not yet been defined. I assume it means best match rotation?

Page 10: I found the second part of the following statement confusing (underlined): cells were defined as ‘“feature-insensitive” (FI) if similarity across context was greater than 0.3 (i.e., cells displaying stability of place fields across context). It is worth noting that this definition only refers to whether the place fields display context-sensitive rotations or not.’ I know that this was added to address my question about rate remapping, but it is tough to understand as written and without the context of the review comment. I suggest addressing the rate issue more directly by stating that these definitions only account for changes in the location of firing, not changes in the rate of firing, which is addressed later on.

Page 10: “The population similarity in spatial representations across context was lower in FS than FI cells across days (Figure 4B).” This is expected, right?

Page 10: context should be plural in the conclusion to the penultimate paragraph on this page

Figure 5F: If I’m interpreting these plots correctly, $BMR1 \neq BMR2$, right? Making the diagonal a different color not on the colormap used (white?) would make this clear.

Page 12 (bottom)-13(top): FI and FS are missing “cells” after each.

Methods – Population Vector Similarity “Only cells present in context A and B were included in the analysis.” What does present mean? Does 1 spike or calcium event count or was there a different threshold?

Reviewer #2 (Remarks on code availability):

I have looked at the code but have not run it as I could not find any sample data to run the code on. The last update to the code was in February 2022. It appears to have many general-use functions for performing data analysis in MATLAB with a Readme file that outlines how to set up a project. However, I could not find any record of the code used for producing the figures in this paper, which makes sense as the last commit/change was from almost two years ago. In short, the code appears to be useful but it is tough to evaluate without example data to test it and I could not find any specifics about how the figures in this manuscript were generated.

Reviewer #3 (Remarks to the Author):

The detailed responses to reviewer comments are appreciated and very helpful in assessing the manuscript. I have some additional and remaining concerns with the current version of the manuscript; my sincere apologies for asking for new analyses in a second round of revisions, but there are remaining points where additional information is needed.

Figure 1C: What are the inset parentheses percentages? Not described either on pages 5/6 of the text or in the figure legend

Page 8: should $F > 0.05$ actually be $p > 0.05$?

Figure 2B: left column maps should be oriented vertically like those across the top for easy comparison (e.g. comparing trial 1 with trial 2 rotation 0 requires the reader to mentally rotate that left column trial 1 map to see that it is the same as the 0 degree rotation map from trial 2; this would also bring that left column trial 2 map into alignment with the same map for the trial 2 images in the right half of the figure)

Figure 2C could use a label on the x-axis saying that this is the best match rotation

Page 9/Methods: “after an exhaustive examination of various models,...” in some way describe the other models. Also, it’s not really clear from the description in this same paragraph what “most informative” is in reference to.

Figure S4: if the cells were grouped by their correlation across environments, how is it that all of these plots span a highly overlapping range of correlations on their x and y axes?

Please explain further how the relationship between within environment and across environment remapping necessarily indicates that this an appropriate threshold. When I suggested comparing these metrics, this was to ensure that across-environment remapping couldn’t just be attributed to a general instability which might also be seen by looking at within-environment stability. I do see that the correlation of those measures is much lower <0.3 than for other deciles, but I don’t follow the logic as to how that justifies this threshold (maybe that’s fine and this method is beyond my training, but I suspect other readers also won’t follow this line of thought, at least from the explanation provided).

When I suggested subdividing the dataset into quartiles, deciles, etc. I guess it wasn’t totally clear that I wasn’t asking for a justification of the threshold, so I’ll try to explain better; I suggest that the subset method be used to check whether dividing FS and FI cells into only 2 groups necessarily resulted in the differences observed in the following analyses. What I meant was: for many of the analyses which compare FI and FS cells, as determined by the 0.3 correlation threshold, instead perform these analyses using each of the deciles to show whether these other features of the data also follow this threshold or of they follow a progression. For example, in 4b, instead of cumulative distributions of population similarity across environments for only FI and FS, do this population similarity for each subset of correlation values 0-0.3, 0.3-0.42, 0.42-0.51, etc. In this case, if population similarity also stratifies by this threshold, then all but one of the CDF curves should cluster over to the right and only the FS curve should be towards the left; alternatively, the CDFs of population similarity may progress smoothly as a function of which chunk of correlations the cells have (and a similar smooth progression as a function of across-environment similarity may also emerge in other analyses).

This also highlights an additional measure that should be reported, which is the percentage of FS neurons that had place fields in both environments. This would help readers understand whether FS cells are doing something different from FI cells by their coherence with the population when firing or simply by restricting their firing to a single field/environment.

Page 12/13: I am still concerned that the exact threshold for FS/FI cells is influencing some of the analyses in this figure. One explanation for the finding in Figure 6B is that because the average change in correlation is the same between groups (as seen in Figure S10), FS cells are more likely to

become FI cells just because the mean of that group's across-environment correlation is closer to the threshold than that mean for the FI cells. This might be addressed by setting the FS/FI threshold at the 50th percentile of across-environment correlations, or in some other way account for the difference in group size/range of correlations.

Another issue that should be addressed is a comparison of the across-day registration quality and confidence of how well isolated the ROIs are of FS and FI cells; or, better yet, looking at quality and ROI isolation as a function of the across-environment correlation. The concern here is that if there are cases where the OASIS algorithm combines two neurons, or the method for registering neurons swaps two neurons, either event is more likely to turn a cell from FS to FI because there are many fewer FS cells and randomly adding firing events to a cell will spread out its event map, leading to higher likelihood of overlap across conditions compared.

Page 13: the inclusion of occupancy metrics in S11A/B is appreciated; better yet (at the authors' discretion) would be a few examples trial trajectories as shown in the response letter. Additionally, it should be confirmed in the text (main or figure legends) that bins were only compared when there was sufficient occupancy of that bin on the trial or trials in the given comparison before smoothing the rate/occupancy maps. This is critical, since it cannot be assumed that bins with low/no occupancy had no cell activity and no analysis can be made of those bins for those trials in the absence of sufficient data.

REVIEWER COMMENTS

Reviewer #1 (Remarks to the Author):

The authors have addressed my comments satisfactorily.
We thank reviewers 1 and 4 for their previous comments.

Reviewer #2 (Remarks to the Author):

In this study, Gagliardi investigated neural mechanisms that could support two important features of spatial orientation: heading retrieval and context identification. They combine a unique context-guided spatial orientation task with electrophysiological and calcium imaging recordings in freely moving mice to uncover two types of neurons in the hippocampus: Feature Insensitive (FI) cells, which encode the geometry of an environment but do not distinguish between two separate contexts with the same geometry, and Feature Sensitive (FS) cells, whose firing is modulated by features that differ between two identically shaped arenas. They exhaustively characterize the similarities and differences between FI and FS cells, showing that FI cell coding remains relatively stable across days while FS cell coding changes with learning. These findings are of interest not only to the field of spatial navigation but will also appeal to a broader audience by providing a general mechanism for how the hippocampus binds spatial and non-spatial information to support memory formation and recall.

The revised manuscript is clear and comprehensive. The addition of data points to plots demonstrates a clear effect while simultaneously representing across-animal variability, which, combined with the addition of many methodological details, will greatly aid other researchers in understanding, reproducing, and expanding these findings. The revised manuscript addresses all of my major concerns, and the data provided supports the authors' conclusions. My only remaining issue relates to how rate changes in FS vs. FI cells over time could code for heading direction or context retrieval. The specifics of this issue are listed immediately following this paragraph, followed by a few minor/specific comments.

Remaining issue/question

Figure 7 – are the rate-remapping SVM classifier results driven differentially by FI vs. FS cells? In other words, do you get the same results if you train and test the classifier with one cell type only? This is related to point 23 from the first review and will be interesting regardless of the outcome. If FS cells drive this result and an FI-only SVM was not able to decode heading or context based on rate alone, that would further strengthen the author's point in the discussion that FS cells undergo experience-dependent modulation while FI cells do not. If, alternatively, FI cells could classify heading, this would suggest that FI cells could be modulated by learning and could incorporate non-geometric information by adjusting their firing rate.

We thank the reviewer for this suggestion. We added this analysis to the Supplement (Figure S16). Please note that the low yield of electrophysiological methods limited our

ability to have high power for the predictions per cell type, especially for FS cells. Nevertheless, the results indicate similar trends in both FS and FI cells predictive power of heading and context. Notably, there were no differences in prediction accuracy between cell types, suggesting that the rate changes integrate information in the whole ensemble.

In the results, we had previously presented the idea that rate changes serve to integrate task-relevant information as animals learn the task. We now emphasized that this happens in all cell types.

“These results indicate that the rate code incorporates information about context and heading as animals gain experience with the task.” Pg. 15

Page 8 “Therefore, we plotted the counts that corresponded to each pair of behavioral responses for each best match rotation angle (Figure S3). We found that correct behavioral sequences (C/G or G/C) were prevalent at all best match rotations...”. What is the “pair of behavioral responses?” From the figure legend it seems like this is the first and second dig location in a given trial. Adding labels to the confusion matrices axes would help clarify this.

“Pair of behavioral responses” refers to the dig choices in the trial pairs that yielded a best match rotation for each rotation tested. A label was added indicating that the y axis corresponds to Trial A and x axis to Trial B (new Figure S4, old Figure S3). We used Trial A and B rather than First and Second dig to distinguish this analysis from the one shown in Figure S1, which examines first and second dig within a trial.

Page 9 “BMR analysis” is used but this acronym has not yet been defined. I assume it means best match rotation?

We added the definition on page 8: “To further validate geometric alignment, we conducted a best match rotation (BMR) analysis, as previously described (Keinath et al., 2017).”

Page 10: I found the second part of the following statement confusing (underlined): cells were defined as ‘“feature-insensitive” (FI) if similarity across context was greater than 0.3 (i.e., cells displaying stability of place fields across context). It is worth noting that this definition only refers to whether the place fields display context-sensitive rotations or not.’ I know that this was added to address my question about rate remapping, but it is tough to understand as written and without the context of the review comment. I suggest addressing the rate issue more directly by stating that these definitions only account for changes in the location of firing, not changes in the rate of firing, which is addressed later on.

Thank you for the suggestion. We eliminated the confusing sentence and replaced it with the reviewer’s suggestion:

“Using the cutoff value of 0.3, we divided the cells shown in the context similarity distribution (Figure 3C) and categorized neurons as “feature-sensitive” (FS) if similarity across contexts

after alignment was equal to or below 0.3 (i.e., cells displayed location remapping across contexts), or as “feature-insensitive” (FI) if similarity across contexts was greater than 0.3 (i.e., cells displayed stability of place fields across contexts). It is worth noting that these definitions only account for changes in the location of firing, not changes in the rate of firing, which we address later in the study.”, pg 10

Page 10: “The population similarity in spatial representations across context was lower in FS than FI cells across days (Figure 4B).” This is expected, right?

The reviewer is correct. This was an expected result based on the single cell data. The population vectors corroborated the effect at the population level. Moreover, this analysis also reveals that this effect is modulated by learning since the population similarity on days 2 and 3 is lower than on day 1.

Page 10: context should be plural in the conclusion to the penultimate paragraph on this page

We corrected this typo and made sure we were consistent throughout the manuscript.

Figure 5F: If I’m interpreting these plots correctly, $BMR1 \neq BMR2$, right? Making the diagonal a different color not on the colormap used (white?) would make this clear.

Thank you for the suggestion. The diagonal was changed to white

Page 12 (bottom)-13(top): FI and FS are missing “cells” after each.

The word cells was added.

Methods – Population Vector Similarity “Only cells present in context A and B were included in the analysis.” What does present mean? Does 1 spike or calcium event count or was there a different threshold?

We apologize for the vague statement. In the context of population vectors, we meant that the cells had to have a place field in both contexts. This meant that we excluded 13 FS cells because they had a place field in only 1 context (4.9% of the total). We added this clarification in the population vector section. We did not have a threshold for inferred spikes.

“Cells must have at least one place field in each context to be included for analysis (13 cells were excluded using this criterion). Place field maps were calculated during periods of movement (speed > 2 cm/s)” pg. 23

Reviewer #2 (Remarks on code availability):

I have looked at the code but have not run it as I could not find any sample data to run the code on. The last update to the code was in February 2022. It appears to have many general-use functions for performing data analysis in MATLAB with a Readme file that outlines how to set up a project. However, I could not find any record of the code used for producing the figures in this paper, which makes sense as the last commit/change was from almost two years ago. In short, the code appears to be useful but it is tough to evaluate without example data to test it and I could not find any specifics about how the figures in this manuscript were generated.

We apologize for the confusion, MuLaNa1 code uploaded in 2022 is our generic code that can be applied to any project. This code was used to generate place cell maps and the initial best match rotation analysis. We now added code specific to this project that can be available at the following links:

Data and code availability:

Processed data are stored in “csv” format on:

<https://github.com/ManuMi68/MuLaNa2/tree/main/NeuroSpatialData>.

Processed data are also available in Mendeley

<https://data.mendeley.com/datasets/gz2vbmsxnn/1>

Mendeley Data, V1, doi: 10.17632/gz2vbmsxnn.1

General analysis code is available on GitHub in the MuLaNa2 repository:

<https://github.com/ManuMi68/MuLaNa2>.

Matlab repository for the paper Figures:

https://github.com/ceIiagagliardi/two_context

https://github.com/marcnormandin/chengs_task_2c_ca1

R statistical repository:

<https://htmlpreview.github.io/?https://github.com/ManuMi68/MuLaNa2/blob/main/NeuroSpatial.html>.

Original recordings are available upon request.

In the current version of the paper, we validated all the code and made sure all the statistical analyses were consistent throughout for all figures.

Reviewer #3 (Remarks to the Author):

The detailed responses to reviewer comments are appreciated and very helpful in assessing the manuscript. I have some additional and remaining concerns with the current version of the manuscript; my sincere apologies for asking for new analyses in a second round of revisions, but there are remaining points where additional information is needed.

We thank the reviewer for his/her thoughtfulness and deep examination of our analyses. After reading the reviewer's detailed comments we had a better understanding of the issue that was not clear. In the current resubmission, we provide more methodological explanations and added the reviewer's requested additional analyses. We believe that the clarifications and combined analyses provide compelling evidence that the distinctions between FS and FI cells do not reflect differences in registration quality or spatial information content, rather they reflect inherent characteristics of different subpopulations in CA1. Below we provide a detailed answer to each of the points raised by the reviewer.

1. Figure 1C: What are the inset parentheses percentages? Not described either on pages 5/6 of the text or in the figure legend

The parentheses show the standard error of the mean. This information was added to Figure caption 1.

2. Page 8: should $F > 0.05$ actually be $p > 0.05$?

This typo was corrected.

3. Figure 2B: left column maps should be oriented vertically like those across the top for easy comparison (e.g. comparing trial 1 with trial 2 rotation 0 requires the reader to mentally rotate that left column trial 1 map to see that it is the same as the 0 degree rotation map from trial 2; this would also bring that left column trial 2 map into alignment with the same map for the trial 2 images in the right half of the figure)

The schematic was corrected.

4. Figure 2C could use a label on the x-axis saying that this is the best match rotation

A label was added to Figure 2C.

5. Page 9/Methods: "after an exhaustive examination of various models,..." in some way describe the other models. Also, it's not really clear from the description in this same paragraph what "most informative" is in reference to.

We evaluated models of various complexities, including exponential, humped, and sigmoidal-family models. The schematic shown below was added as Figure S6 to describe the approach in detail.

In the context of our model, most informative refers to the critical point. As described in the schematic, the critical value is the value that turns the function to zero (y axis), coinciding with the half-life parameter (the value at which the function reaches half of its rate), and the value that maximizes the relative growth of the function. In other words, decile 1 comprises values that markedly differ from the rest, indicating that correlations across context critically change outside the first decile range.

6. Figure S4: if the cells were grouped by their correlation across environments, how is it that all of these plots span a highly overlapping range of correlations on their x and y axes?

Please explain further how the relationship between within environment and across environment remapping necessarily indicates that this an appropriate threshold. When I suggested comparing these metrics, this was to ensure that across-environment remapping couldn't just be attributed to a general instability which might also be seen by looking at within-environment stability. I do see that the correlation of those measures is much lower <0.3 than for other deciles, but I don't follow the logic as to how that justifies this threshold (maybe that's fine and this method is beyond my training, but I suspect

other readers also won't follow this line of thought, at least from the explanation provided).

When I suggested subdividing the dataset into quartiles, deciles, etc. I guess it wasn't totally clear that I wasn't asking for a justification of the threshold, so I'll try to explain better; I suggest that the subset method be used to check whether dividing FS and FI cells into only 2 groups necessarily resulted in the differences observed in the following analyses. What I meant was: for many of the analyses which compare FI and FS cells, as determined by the 0.3 correlation threshold, instead perform these analyses using each of the deciles to show whether these other features of the data also follow this threshold or if they follow a progression. For example, in 4b, instead of cumulative distributions of population similarity across environments for only FI and FS, do this population similarity for each subset of correlation values 0-0.3, 0.3-0.42, 0.42-0.51, etc. In this case, if population similarity also stratifies by this threshold, then all but one of the CDF curves should cluster over to the right and only the FS curve should be towards the left; alternatively, the CDFs of population similarity may progress smoothly as a function of which chunk of correlations the cells have (and a similar smooth progression as a function of across-environment similarity may also emerge in other analyses).

We thank the reviewer for his/her additional suggestions and thorough focus on our analysis. This point raises several issues that we answer first in general (below) and in more detail in the following sections of this letter.

-The overlap in across-context correlation values (y axis) in Figure S5 (old Figure S4) arises from individual pairwise trial comparisons, whereas the context similarity values that define the deciles (Figure 3C) arise from an overall alignment that maximizes similarity across all trials.

-We acknowledge that cells in decile 1 have lower correlation values, even within context. However, this is not due to poor registration quality (Figure S13A) or differences in spatial information content (Figure S9A-B). Furthermore, we find that these cells have an important function discriminating the contexts at the population and single cell level, which is greater than that observed from cells in other deciles (Figure S12). Additionally, these cells show patterns of coherence (Figure 5) and population similarity across contexts that are modulated by learning (Figure 4B), suggesting that they play a critical role in task-contingencies. As we note in the discussion (pg. 16), unstable cells in CA1 have been previously shown to have important roles in learning (Tanaka et al., 2018). Therefore, taking into account the validation measures we provide (see also Figure S13B), we feel confident that FS cells in decile 1 have physiological relevance.

-We now understand that the reviewer did not ask for a threshold validation per se. However, in this resubmission we left the modeling approach to validate the threshold because it provides an objective measure to segregate the cells based on correlation

values. Regarding the new analyses required by the reviewer, below we show what we added in this resubmission.

A) Using other analyses to determine the relevance of FS cells (decile1)

A.1) We plotted the spatial information content by dividing the cells into deciles from the across context similarity distribution (Figure 3C). Our data show that there are no differences in spatial information content across deciles (Figure S9B).

A.2) Following the reviewer's suggestion, we calculated the population vector similarity scores across context by dividing the cells into deciles from the across context similarity distribution (Figure 3C). Figure S12A shows that cumulative distributions of normalized dot products from deciles 2 to 9 are more clustered to the right, whereas those corresponding to decile 1 are in the far left. Therefore, although there is graded distribution, which is unavoidable working with correlation scores, the greatest differences are observed between decile 1 and the rest of the deciles.

A.3) We also show that context prediction as a function of across context correlation threshold plateaus around 0.2-0.3 values (Figure S12B). These data demonstrate that context prediction is optimal when across context correlation ranges between 0-0.3, but steadily decreases as the threshold rises to incorporate more FI cells.

Figure S12. Complement of Figure 4. A) Population similarity across context represented by cumulative proportions. Average aligned maps recorded in Context A and B were stacked and the normalized dot product was computed between population vectors in each corresponding pixel across context. The cells were divided by deciles obtained from the average distribution of correlations across context shown in Figure 2C. Although the cumulative distributions of dot products display a graded pattern that increases with decile, the cumulative curve corresponding to decile 1 is markedly shifted to the left in comparison to the other distributions. This suggests that cells in decile 1 show lower population similarity across context than cells in the other deciles. B) Context prediction as a function of remapping threshold across context. Context prediction decreases as thresholds values increase. However, there is a plateau between 0.2-0.3 correlation values, and a substantial decline at higher correlation values.

7. This also highlights an additional measure that should be reported, which is the percentage of FS neurons that had place fields in both environments. This would help readers understand whether FS cells are doing something different from FI cells by their coherence with the population when firing or simply by restricting their firing to a single field/environment.

Of the 264 FS cells identified across 12 animals over 3 days, only 13 cells had place fields only in one context, amounting to 4.92% of the total sampled population.

8. Page 12/13: I am still concerned that the exact threshold for FS/FI cells is influencing some of the analyses in this figure. One explanation for the finding in Figure 6B is that because the average change in correlation is the same between groups (as seen in Figure S10), FS cells are more likely to become FI cells just because the mean of that group's across-environment correlation is closer to the threshold than that mean for the FI cells. This might be addressed by setting the FS/FI threshold at the 50th percentile of across-environment correlations, or in some other way account for the difference in group size/range of correlations.

We want to clarify that old Figure S10 (now Figure S12) shows rate changes in calcium data not the average change in correlation values. However, we do understand the critical point raised by the reviewer. He/she suggests that since FS are defined by a more constrained range of correlations than FI, this may have biased our estimation of identity stability. In other words, if a FS cell with a correlation across context of 0.25 on day 1 shows an increase of 0.15 on day 2, the across context correlation would be 0.40 (0.25+0.15), which would categorize this cell as FI on day 2. However, if a FI cell with a correlation across context of 0.40 on day 1 increases the same amount on day 2 (across context correlation of 0.55), it would still be considered FI on that day. This is an intriguing possibility that we did not consider before and we thank the reviewer for bringing this important point up. To address this, we calculated the difference in correlation values across days. We hypothesized that if FS cells display changes in identity modulated by learning, then the change in correlation values across days for cells in decile 1 should be significantly higher than other deciles. Conversely, if correlation differences across days were equal for all deciles, then we would not be able to rule out the possibility that changes in identity across days could be due to the fact that FS cells are defined by more constrained correlation values. We found that across days cells in decile 1 display more significant changes in correlations across context than other deciles (Figure S13B). These data provide confidence that FS changes in identity are due to their inherent properties rather than their constrained correlation range.

Figure S13B. B) Change in context similarity across days per decile. The absolute difference in correlations for all registered cells was calculated for each day pair by dividing the cells into the same deciles described in panel A. The data show that cells in decile 1 displayed more

significant differences in similarity scores across context than cells from other deciles (Day 1 to Day 2: $F(9,279) = 8.78$, $p < .0001$; Day 1 to Day 3: $F(9, 262) = 3.94$, $p < .0001$; Day 2 to Day 3: $F(9,226) = 3.93$, $p < .0001$, Pos Hoc multiple comparisons shown in the graph). All of the graphs correspond to Boxplots, in which the boxes indicate the upper and lower quartiles of the data and the whiskers (extending lines) the minimum and maximum outside the quartiles. The horizontal line indicates the median, and + indicates the presence of outliers. Asterisks represent $p \leq .05$ (*), $p \leq .01$ (**) or $p \leq .001$ (***), # indicates a trend approaching the significance level we set at .05.

9. Another issue that should be addressed is a comparison of the across-day registration quality and confidence of how well isolated the ROIs are of FS and FI cells; or, better yet, looking at quality and ROI isolation as a function of the across-environment correlation. The concern here is that if there are cases where the OASIS algorithm combines two neurons, or the method for registering neurons swaps two neurons, either event is more likely to turn a cell from FS to FI because there are many fewer FS cells and randomly adding firing events to a cell will spread out its event map, leading to higher likelihood of overlap across conditions compared.

We thank the reviewer for this important suggestion. The cell registration output provides a measure of footprint quality, which is obtained by correlating the spatial footprints across registered sessions (Sheintuch et al., 2017). This quality measure ranges from 0 to 1. We plotted the registration quality of all the cells that were present in at least 2 sessions as a function of average across context similarity per decile. Figure S13A shows that there were no differences in footprint quality across deciles [$F(9, 1317)=0.88$, $p=0.5417$].

Furthermore, we had previously evaluated the spatial footprints of cells on shape metrics (circularity, size, number of contiguous pixels) and trained a naïve Bayesian classifier against a subset of visually confirmed spatial footprints that were “cell-like”. Only cells that were classified as cell-like were included in subsequent analyses (Details in the Method section). This filter removed 1469 cells out of a total of 3771 cells that were identified across three days of recordings (this included artifacts and cells that could not be dissociated by shape). We would like to note that we always impose a shape metric filter to further validate the OASIS algorithm to avoid the possibility that OASIS combines two neurons or introduces artifacts. Finally, since there was no difference in spatial information content between FS and FI cell on any day of training (Figure S9), our data collectively indicate that the remapping properties of FS cells are not due to differences in registration or spatial quality.

Figure S13A. Registration quality per decile. To determine if the registration quality influenced the remapping properties across context, we used the registration score index (range = 0–1) developed by Sheintuch et al. (2017), which is provided by the CellReg algorithm. This value evaluates the registration certainty of all cell-pairs within the population of registered cells by computing the footprint spatial correlations for each cell across days. Spatial registration quality was evaluated for any cell present during at least 2 days of testing by dividing the average correlation across context distribution in deciles. There were no significant differences in footprint quality across deciles ($F(9, 1317)=0.88$, $p=.54$). This indicates that changes in stability across time are not due to differences in spatial registration quality.

Finally, we want to mention that other labs have found that unstable cells in CA1 may have important properties modulated by learning. McHugh and colleagues employed a genetic tagging approach to delineate the characteristics of place cells expressing cFos, an immediate early gene linked to learning, and those that do not. Consistent with our discoveries, their findings revealed two distinct populations of CA1 place cells with different remapping tendencies. The cFos-positive cells, which constituted a minority of all active place cells (as was the case of FS cells in our study), exhibited instability (i.e., remapping) in familiar environments. Conversely, the majority of active neurons, which were cFos-negative, displayed high stability in familiar settings and remapping in novel ones. Although using genetic markers facilitates distinguishing cell types in comparison to correlation scores, the similarity in cFos-positive cell remapping characteristics observed by McHugh's and our study suggests that differences in stability may reflect unique properties of CA1 neurons (Tanaka et al., 2018).

In summary, we have made a thorough attempt to select a remapping threshold using objective measures, demonstrating that this threshold does not separate cells based on poor registration or spatial quality. We also showed that FS cells produce maximal context discrimination using population similarity across context and serve to predict context across days. We believe that these analyses have strengthened our conclusions and thank the reviewer for asking us to explore alternative possibilities.

10. Page 13: the inclusion of occupancy metrics in S11A/B is appreciated; better yet (at the authors' discretion) would be a few examples trial trajectories as shown in the response letter. Additionally, it should be confirmed in the text (main or figure legends) that bins were only compared when there was sufficient occupancy of that bin on the trial or trials in the given comparison before smoothing the rate/occupancy maps. This is critical, since it cannot be assumed that bins with low/no occupancy had no cell activity and no analysis can be made of those bins for those trials in the absence of sufficient data.

We thank the reviewer for asking this clarification, which we now made clear in the caption. The reported occupancy maps shown in Figure S2B-C were calculated before smoothing. Path examples are shown in Figure S2 and below.

AK42

Day 1

Day 2

Day 3

MG1

References

- Keinath, A. T., Julian, J. B., Epstein, R. A., & Muzzio, I. A. (2017). Environmental Geometry Aligns the Hippocampal Map during Spatial Reorientation. *Curr Biol*, 27(3), 309-317. <https://doi.org/10.1016/j.cub.2016.11.046>
- Tanaka, K. Z., He, H., Tomar, A., Niisato, K., Huang, A. J. Y., & McHugh, T. J. (2018). The hippocampal engram maps experience but not place. *Science*, 361(6400), 392-397. <https://doi.org/10.1126/science.aat5397>

REVIEWERS' COMMENTS

Reviewer #2 (Remarks to the Author):

The authors have addressed all my comments. Nice work!

Reviewer #2 (Remarks on code availability):

The code and data are easily accessible, and I was able to install and run code from each of the three GitHub repositories to reproduce a figure panel from each. I did not exhaustively check and run the code for each figure panel, but I did notice that code for at least one panel (3E) appears to be missing. The authors should double-check to ensure that the relevant code is there for all figures.

Reviewer #3 (Remarks to the Author):

I am satisfied with the replies to my questions except for the justification of the 0.3 threshold to separate FS and FI cells. Many results rely on this threshold, and much attention is given to the distinction between FS and FI cells in the Discussion section. The degree to which these findings should be thought of as broad differences at the ends of a continuum vs. a highly specific discontinuity, and how closely other studies should expect similar results, depends on the robustness of this method.

The additional information provided in S5 and S6 is appreciated, but since this analysis is now being used to justify this threshold (as stated on page 9, lines 360-362 of the main text and S6/page 9 of the supplement, in contrast to what was present in the original submission), it is crucial that the typical reader follow the reasoning behind these methods. It still has not been explained why any of the details about this procedure are appropriate or if they are standard practice for setting a binary threshold on a continuous, non-multimodal distribution. The specific choice to divide the population of cells by their session average across-context correlation using the relationship between the average of pairwise trial correlations within and across contexts strays near circular reasoning/post-hoc justification: the use of this threshold is justified because it produces significant effects (authors' rebuttal letter, page 7, second half of paragraph beginning "We acknowledge that cells in decile 1..."). The correlations in S5 and S6 could just as easily have been part of the main results, so it is then an arbitrary decision to use those results to select the

threshold as opposed to relying on the differences observed in e.g. 5E to justify the threshold's location.

Here are some possible ideas for how to proceed:

1. Employ a method for setting a binary threshold on a continuous distribution that has been used widely.
2. Select equally sized FI and FS groups, for example from the top and bottom of the across-context correlation distribution leaving out some middle portion. This option avoids the need to make a claim about a specific threshold.
3. Soften the language of the discussion to position these results as broad ideas, specifically allowing that data from cells near the 0.3 across-context correlation threshold may not follow such a sharp cutoff.

REVIEWERS' COMMENTS

Reviewer #2 (Remarks to the Author):

The authors have addressed all my comments. Nice work!

Reviewer #2 (Remarks on code availability):

The code and data are easily accessible, and I was able to install and run code from each of the three GitHub repositories to reproduce a figure panel from each. I did not exhaustively check and run the code for each figure panel, but I did notice that code for at least one panel (3E) appears to be missing. The authors should double-check to ensure that the relevant code is there for all figures.

We thank the reviewer for his positive appraisal. We uploaded the code for panel 3E and checked that the code for all Figures is uploaded.

Reviewer #3 (Remarks to the Author):

I am satisfied with the replies to my questions except for the justification of the 0.3 threshold to separate FS and FI cells. Many results rely on this threshold, and much attention is given to the distinction between FS and FI cells in the Discussion section. The degree to which these findings should be thought of as broad differences at the ends of a continuum vs. a highly specific discontinuity, and how closely other studies should expect similar results, depends on the robustness of this method.

The additional information provided in S5 and S6 is appreciated, but since this analysis is now being used to justify this threshold (as stated on page 9, lines 360-362 of the main text and S6/page 9 of the supplement, in contrast to what was present in the original submission), it is crucial that the typical reader follow the reasoning behind these methods. It still has not been explained why any of the details about this procedure are appropriate or if they are standard practice for setting a binary threshold on a continuous, non-multimodal distribution. The specific choice to divide the population of cells by their session average across-context correlation using the relationship between the average of pairwise trial correlations within and across contexts strays near circular reasoning/post-hoc justification: the use of this threshold is justified because it produces significant effects (authors' rebuttal letter, page 7, second half of paragraph beginning "We acknowledge that cells in decile 1..."). The correlations in S5 and S6 could just as easily have been part of the main results, so it is then an arbitrary decision to use those results to select the threshold as opposed to relying on the differences observed in e.g. 5E to justify the threshold's location.

Here are some possible ideas for how to proceed:

1. Employ a method for setting a binary threshold on a continuous distribution that has been used widely.
2. Select equally sized FI and FS groups, for example from the top and bottom of the across-context correlation distribution leaving out some middle portion. This option avoids the need to make a claim about a specific threshold.
3. Soften the language of the discussion to position these results as broad ideas, specifically allowing that data from cells near the 0.3 across-context correlation threshold may not follow such a sharp cutoff.

We thank the reviewer for his suggestions. We would like to clarify that although the method we used to select the threshold may not have been widely used, it has been applied to a variety of experimental conditions to separate continuous data.

To address the validity of the method, we added to the results: “*Non-linear modeling has been applied to characterize the dynamics of other processes, such as learning curves*⁴¹, *psychophysics*⁴², and *biological processes*⁴³. *Importantly, similar modeling approaches have been used to characterize the behavioral relevance of distinct neuronal cell types*⁴⁴.” Pg. 9-10

However, the issue remains that certainty decreases as values approach the threshold value. To address this concern, we followed reviewer’s 3 suggestion (“Soften the language of the discussion to position these results as broad ideas, specifically allowing that data from cells near the 0.3 across-context correlation threshold may not follow such a sharp cutoff”) and added the following paragraph to the discussion:

*“The parallels between the Tanaka et al. study*⁵⁸ *and ours suggest that distinct subpopulations of CA1 neurons with inherent dynamic activity patterns are a critical feature of hippocampal processing. Interestingly, these similarities emerge even though in our study we separated cell types from a continuous distribution of similarity scores rather than using a selective molecular marker. Here, we used modeling to objectively select a separation threshold, a method previously used on continuous datasets*^{41,43}*. Furthermore, some of the model parameters (e.g., inflexion point) that we used to categorize FI and FS cells have been previously used to discriminate cell types associated with task-related behavior*⁴⁴*. However, it is worth noting that a potential caveat of this methodology is that categorization confidence decreases near the threshold. Despite this limitation, we show that the distinct properties of FI and FS cells do not reflect changes in registration quality or spatial information content.”* Pg. 17

In summary, we thank our 4 reviewers for their thoughtful review of our manuscript and thoughtful suggestions, which have served to strengthen our conclusions.

References

- 41 Aristizabal, J. A., Ramos-Alvarez, M. M., Callejas-Aguilera, J. E. & Rosas, J. M. Attention to irrelevant contexts decreases as training increases: Evidence from eye-fixations in a human predictive learning task. *Behav Processes* **124**, 66-73 (2016). <https://doi.org/10.1016/j.beproc.2015.12.008>
- 42 Martín-Guerrero, T. L., Rosas, J. M., Paredes-Olay, C. & Ramos-Alvarez, M. M. Psychophysical Curves for Tasting Based on A Dissociation Model. *J Sens Stud* **30**, 225-236 (2015). <https://doi.org/10.1111/joss.12153>
- 43 Panik, M. J. 1 online resource (455 pages) (John Wiley & Sons, Inc., Hoboken, New Jersey, 2014).
- 44 Ahn, J. R. & Lee, I. Neural Correlates of Both Perception and Memory for Objects in the Rodent Perirhinal Cortex. *Cereb Cortex* **27**, 3856-3868 (2017). <https://doi.org/10.1093/cercor/bhx093>
- 58 Tanaka, K. Z. & McHugh, T. J. The Hippocampal Engram as a Memory Index. *J Exp Neurosci* **12**, 1179069518815942 (2018). <https://doi.org/10.1177/1179069518815942>